# GRAVER: Generative Graph Vocabularies for Robust Graph Foundation Models Fine-tuning

**Haonan Yuan**[1], **Qingyun Sun**[1]*, **Junhua Shi**[1], **Xingcheng Fu**[2], **Bryan Hooi**[3],
**Jianxin Li**[1], **Philip S. Yu**[4]

[1]SKLCCSE, School of Computer Science and Engineering, Beihang University
[2]Key Lab of Education Blockchain and Intelligent Technology, Guangxi Normal University
[3]School of Computing, National University of Singapore
[4]Department of Computer Science, University of Illinois, Chicago
`{yuanhn,sunqy,shijunhua,lijx}@buaa.edu.cn`
`fuxc@gxnu.edu.cn, bhooi@comp.nus.edu.sg, psyu@uic.edu`

## Abstract

Inspired by the remarkable success of foundation models in language and vision, Graph Foundation Models (GFMs) hold significant promise for broad applicability across diverse graph tasks and domains. However, existing GFMs struggle with unstable few-shot fine-tuning, where both performance and adaptation efficiency exhibit significant fluctuations caused by the randomness in the support sample selection and structural discrepancies between the pre-trained and target graphs. How to fine-tune GFMs robustly and efficiently to enable trustworthy knowledge transfer across domains and tasks is the major challenge. In this paper, we propose **GRAVER**, a novel **G**enerative g**RA**ph **V**ocabulari**E**s for **R**obust GFM fine-tuning framework that tackles the aforementioned instability via generative augmentations. Specifically, to identify transferable units, we analyze and extract key class-specific subgraph patterns by ego-graph disentanglement and validate their transferability both theoretically and empirically. To enable effective pre-training across diverse domains, we leverage a universal task template based on ego-graph similarity and construct graph vocabularies via graphon-based generative experts. To facilitate robust and efficient prompt fine-tuning, we grave the support samples with in-context vocabularies, where the lightweight MoE-CoE network attentively routes knowledge from source domains. Extensive experiments demonstrate the superiority of GRAVER over *effectiveness*, *robustness*, and *efficiency* on downstream few-shot node and graph classification tasks compared with **15** state-of-the-art baselines.

## 1   Introduction

Graph-structured data is pervasive across a wide range of domains, including social networks [13, 127], recommendation systems [114, 100], and bioinformatics [50, 17]. Graphs are capable of modeling complex relationships and attributed interactions between entities, making them a powerful abstraction for solving real-world problems, such as fraud detection [60, 128], molecular property prediction [70, 135], and protein-protein interaction modeling [34, 15]. While foundation models such as GPTs [1] and Vision Transformers [38] have achieved impressive generalization capabilities in language and vision through large-scale pre-training, they struggle when applied directly to graphs due to their non-Euclidean nature, structural heterogeneity, and task diversity inherent in graph data [52, 46, 63, 37]. To address this gap, Graph Foundation Models (GFMs) [64, 75, 149] have emerged with the goal of enabling general-purpose learning on graph-structured data. Instead of

---

*Corresponding author.

39th Conference on Neural Information Processing Systems (NeurIPS 2025).

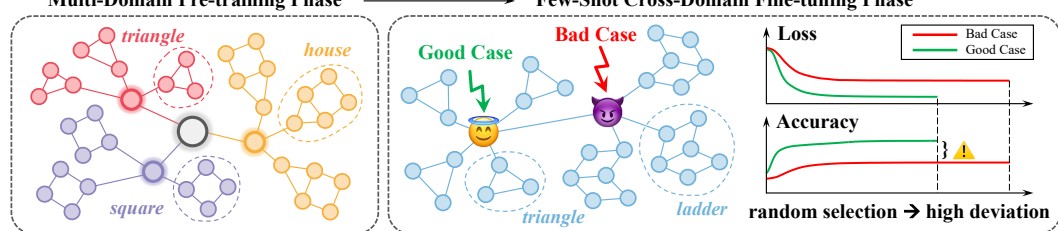

Figure 1: Case study of the fine-tuning instability. When support samples share structural patterns with pre-trained graphs (*e.g.*, triangle), fine-tuning is efficient and stable (Good Case). In contrast, mismatched patterns (*e.g.*, ladder) lead to poor convergence and lower accuracy (Bad Case). Loss and accuracy curves illustrate that random support selection causes high performance deviation, highlighting the need for structure-aware augmentation.

relying on task-specific deep graph models, ideal GFMs are trained once on diverse multi-domain graph datasets and then adapted with minimal supervision to a variety of downstream tasks [76, 5]. Such a *"pretrain-then-finetune"* paradigm [86, 23, 42] holds the promise of unifying graph learning across domains by providing a reusable model architecture that generalizes structural knowledge at scale [58, 83, 122, 136, 126].

Existing research on GFMs has addressed prominent challenges like architecture scalability [78, 42], domain alignment [47, 136], prompt design [83, 45], *etc.* Nevertheless, GFMs encounter a more fundamental and pervasive issue: **robustness** and **efficiency** of few-shot prompt fine-tuning. Specifically, the existing GFMs exhibit significant instability and inefficiency, whereby fine-tuning performance fluctuates heavily due to randomness in the selection of limited support samples and structural discrepancies between pre-trained and target graph domains. We conducted an illustrative case study in Figure 1 to highlight this issue. A synthetic cross-domain node classification setting is constructed with motif-based graphs. The findings reveal a consistent trend: when the randomly selected support nodes exhibit structural patterns similar to those in pre-training (*e.g.*, triangle △), fine-tuning converges faster and yields higher accuracy (Good Case). In contrast, the mismatched structures (*e.g.*, ladder ⋈) lead to slower convergence and degraded performance (Bad Case). This phenomenon is intuitive and commonly observed in practice. Such instability severely undermines the practical applicability and reliability of GFMs in real-world scenarios, where dependable knowledge transfer and rapid adaptation to new tasks and domains are essential.

**Research Question:** How can Graph Foundation Models be fine-tuned *robustly* and *efficiently* under random support sets to enable trustworthy knowledge transfer across diverse domains and tasks?

Recent studies attempt to improve GFM's adaptation ability. One line of work focuses on pre-training with transferable patterns [83, 7, 148, 97]. The most relevant GFT [98] proposes a computation-tree-based vocabulary to enable pattern reuse across domains. However, the proposed vocabulary is limited to tree-shaped structures, restricting transferability in multi-domain settings. Another line of research emphasizes generative augmentation [137, 84, 10, 96], which interpolates structures to simulate new training examples. Yet, they often generate task-agnostic structures that misalign with target domains and do not condition on the few-shot support samples, limiting their informativeness. A third group of works [122, 120, 119] introduces modular pre-training with weighted source tokens as prompts to facilitate domain-aware transfer. While effective in improving cross-domain adaptation, these strategies still fall short in scenarios where only a handful of labeled target samples are available. Notably, none of these aforementioned paradigms simultaneously address the dual challenges of generating transferable patterns and ensuring stable fine-tuning under arbitrary few-shot support sets.

To tackle this dilemma, we propose **GRAVER**, a novel **G**enerative g**RA**ph **V**ocabulari**E**s for **R**obust GFM fine-tuning framework. GRAVER directly addresses the instability inherent in few-shot fine-tuning through in-context subgraph augmentations. Specifically: ❶ We first analyze and validate the key transferable patterns from an ego-graph disentanglement perspective, rigorously exploring their transferability theoretical foundations and empirical utility across multiple domains. ❷ By leveraging this insight, we establish adaptive graph vocabularies through graphon-based generative experts in the multi-domain pre-training stage. ❸ At the final fine-tuning stage, GRAVER enhances randomly selected support samples by embedding context-specific generative vocabularies guided by a lightweight MoE-CoE network, which selectively routes pertinent structural and semantic knowledge from relevant source domains and prevents negative transfer. ***Our contributions are:***

- We propose a novel trustworthy GFM with robust and efficient prompt fine-tuning under multi-domain pre-training and cross-domain adaptation. To the best of our knowledge, this is the first GFM with generative graph vocabularies to in-context augment support sets for stable adaptation.
- We design graphon-based generative experts to synthesize structure-feature vocabularies aligned with transferable subgraphs, and introduce a MoE-CoE routing mechanism to dynamically assemble source-domain knowledge for context-aware augmentation during fine-tuning.
- Extensive experiments demonstrate its superior *effectiveness*, *robustness*, and *efficiency* on downstream few-shot node and graph classification compared with **15** state-of-the-art baselines.

## 2    Related Work

### 2.1    Multi-Domain Pre-trained Graph Foundation Models (Appendix G.1)

Graph foundation models have made rapid progress by using self-supervised pre-training to extract transferable patterns from large-scale graphs [57, 29, 104]. Pre-training tasks such as node masking, edge prediction, and contrastive learning have enabled methods like GCC [69], GPT-GNN [30], GPPT [82], L2P-GNN [61], and others [105, 36, 109, 8, 7, 113] to serve as the strong initializations for downstream tasks. However, most of these methods assume the structural or semantic similarity between pre-training and target domains [32, 135], making them sensitive to distribution shifts.

To address this, recent studies have investigated cross-domain graph learning, aiming to transfer knowledge between dissimilar domains. Models such as GCOPE [136], OMOG [54], PGPRec [112], UniAug [88], and UniGraph [23] focus on learning domain-invariant representations. While effective for single-source transfer, these methods often struggle to generalize across multiple heterogeneous domains and may rely on task-specific assumptions that limit scalability.

Toward multi-domain foundation models, recent work explores unified pre-training strategies across diverse domains. OFA [54] and HiGPT [87] use LLMs to encode node information as text, which improves alignment but restricts applicability to text-attributed graphs [95, 85]. GCOPE [136] introduces virtual domain bridges but lacks semantic alignment. MDGPT [123] incorporates domain tokens and SVD-based projection, though it does not ensure semantic consistency across domains.

### 2.2    Graph Foundation Models Fine-tuning with Prompt (Appendix G.2)

Fine-tuning the pre-trained graph foundation models is a widely adopted strategy to adapt general representations to downstream tasks [130, 84, 139, 108, 85, 115, 33, 35, 129]. Traditional methods typically update full model or selected components, which can be effective but often incur high costs.

To improve fine-tuning efficiency, parameter-efficient fine-tuning (PEFT) techniques update a small set of parameters [143, 89, 19, 106] or introduce lightweight modules [49, 147, 110]. These approaches preserve most pre-trained knowledge while reducing computational load. However, their generalization across domains remains limited due to the difficulty of handling domain shifts.

Prompt-based fine-tuning offers an alternative by conditioning the model on learned input prompts rather than altering parameters [55, 141, 140, 11]. In graph learning, prompt tuning leverages structured embeddings to guide the model toward task-specific patterns [82, 145, 77, 132], achieving better robustness and adaptability with minimal updates. Based on this paradigm, numerous graph prompt methods have been developed [82, 83, 58, 18, 145, 14, 31, 123, 118, 149], which incorporate learnable prompts to highlight structural or semantic cues, improving transferability across tasks.

### 2.3    GFM Pre-training with Transferable Patterns (Appendix G.3)

Graph Foundation Models aim to capture structural patterns that generalize across diverse domains and tasks [69, 142, 83, 7, 148, 97, 96]. A key challenge is defining transferable subgraph units for pre-training, since graphs do not contain obvious tokens like words or image patches.

One approach addresses this by identifying common substructures such as motifs or computation trees as a reusable graph vocabulary [142, 146, 47, 138, 144, 97, 117]. For example, GFT [98] encodes each node's computation tree and discretizes it through vector quantization. This enables the model to represent graphs in terms of recurring structural units, which improves generalization across domains and mitigates knowledge conflicts during transfer.

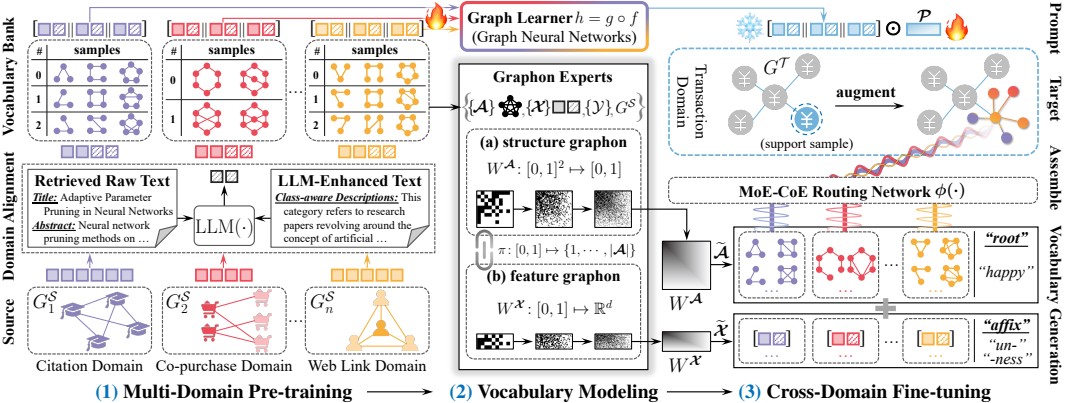

Figure 2: Framework of GRAVER. **(1)** aligns semantics via LLM-enhanced raw text, and disentangles ego-graph into transferable patterns. **(2)** models structure-feature tokens with graphon experts. **(3)** assembles class-aware vocabularies to augment support samples for robust cross-domain fine-tuning.

Another approach focuses on mining frequent motifs during pre-training [135, 4, 133]. MICRO-Graph [133] trains GNNs using motif-based contrastive objectives, while MGSSL [135] reconstructs motif segments to learn interpretable structural blocks, achieving strong performance in molecular prediction tasks. Other methods learn universal structural patterns from synthetic graphs [113].

## 2.4 GFM Fine-tuning with Generative Augmentations (Appendix G.4)

To improve the adaptability of pre-trained graph models, recent work explores generative augmentation techniques that synthesize new data aligned with the target graph distribution [30, 137, 7, 84, 115, 86, 111]. Unlike traditional perturbations on edges or features, these approaches aim to produce structurally meaningful examples to support robust fine-tuning.

Recent efforts have explored generative strategies to enhance the robustness and adaptability of graph foundation models. One line of work leverages graphons, continuous functions that define graph distributions [2, 68, 72, 12, 71, 84]. For instance, $\mathcal{G}$-Mixup [21] interpolates between class-wise graphons to produce realistic hybrid graphs that improve generalization and noise robustness. Another direction introduces generative fine-tuning objectives to mitigate the mismatch between pre-training and downstream distributions [48, 84, 9, 53], such as G-Tuning [84], which preserves the generative structure of downstream graphs to avoid overfitting. In addition, subgraph-level augmentations [3, 131, 41, 101, 24, 22] have been proposed to help models decouple subgraph semantics from global context; for example, [57] replaces subgraph environments during training to improve interpretability and robustness in molecular tasks. Complementing these efforts, mixture-of-experts architectures[27, 94, 59, 99, 20, 123] such as GraphMETRO [99] deploy multiple expert networks specialized on distinct structural regimes, with routing mechanisms to adaptively select experts for diverse input distributions.

## 3 Preliminary

**Notations.** A graph is denoted as $G = (\mathcal{V}, \mathcal{E})$, where $\mathcal{V}$ is the node set and $\mathcal{E}$ is the edge set. Let $\mathbf{A} \in \{0, 1\}^{N_i \times N_i}$ be the adjacency matrix and $\mathbf{X} \in \mathbb{R}^{N_i \times d_i}$ be the node feature matrix, where $N = |\mathcal{V}|$ is the number of nodes and $d_i$ denotes input feature dimension. $\mathbf{H}$ means hidden embeddings of $\mathbf{X}$.

**Pre-training and Few-shot Fine-tuning.** Given $n$ source graphs $\{G^{\mathcal{S}}\} \in \mathcal{G}^{\mathcal{S}}$ from multi-domains $\{D^{\mathcal{S}}\} \in \mathcal{D}^{\mathcal{S}}$ with their labels $\{Y^{\mathcal{S}}\} \in \mathcal{Y}^{\mathcal{S}}$, the pre-training goal is to train a graph learner $h = g \circ f$ on multi-domain graph datasets composed of a GNN-based graph encoder $f$ and a discriminator $g$, after which the pre-training parameter $\Theta^{\star}$ is frozen. During fine-tuning, given a set of target graphs $\{G^{\mathcal{T}}\} \in \mathcal{G}^{\mathcal{T}}$ from target domain $\{D^{\mathcal{T}}\} \in \mathcal{D}^{\mathcal{T}}$ (seen or unseen) with $m$ randomly selected labels $\{Y^{\mathcal{T}}\} \in \mathcal{Y}^{\mathcal{T}}$ (support set) under the $m$-shot setting ($m \ll \sum_{i=1}^{n} N_i$), the goal is to adapt the pre-trained GFM to predict the labels of the unlabeled samples (query set) correctly by learnable graph prompts.

# 4 Proposed GFM Framework: GRAVER

In this section, we elaborate on the proposed GRAVER with its framework illustrated in Figure 2.

## 4.1 Pre-training with Transferable Graph Vocabulary

A central challenge in constructing a transferable graph vocabulary for robust GFM fine-tuning lies in identifying subgraph patterns (graph vocabulary) that generalize across tasks and domains.

**Multi-Domain Alignment.** Graphs in different domains present heterogeneous features with inconsistent dimensions and semantics. To establish a unified representation space, we introduce a set of shared aligners that consequently unify dimension- and semantic-wise features. Given a graph $G_i^{\mathcal{S}} = (\mathcal{V}_i, \mathcal{E}_i)$ $(i \leqslant n)$ with feature matrix $\mathbf{X}_i^{\mathcal{S}} \in \mathbb{R}^{N_i \times d_i}$ from source domain, the aligners perform:

$$\widehat{\mathbf{X}}_i^{\mathcal{S}} \in \mathbb{R}^{N_i \times d} = \mathbf{W}_i^\top \big( \mathcal{F}\big(\mathbf{X}_i^{\mathcal{S}} \,\|\, \mathrm{LLM}\big(\mathbf{X}_i^{\mathcal{S}}\big)\big)\big), \quad \text{for all } G_i^{\mathcal{S}} \in \{G^{\mathcal{S}}\}, \tag{1}$$

where $\mathrm{LLM}(\cdot)$ is implemented by retrieving feature-related raw texts and enhancing class-aware semantics by large language models (GPT-4 and BertTokenizer) [47]. $\mathcal{F}$ implemented by truncated SVD [80] aligns feature dimensions, followed by a learnable projection $\mathbf{W}_i$ that normalizes semantics.

**Factor-aware Ego-Graph Disentanglement.** Motivated by the observation that localized graph patterns often emerge from multiple latent semantic factors [44], such as social affiliation, topic preference, structural role, *etc.*, each factor-specific subgraph forms a distinct semantic unit, analogous to a vocabulary in linguistics. Treating these units holistically during message-passing and aggregation may obscure the unique semantics of each vocabulary, significantly weakening the discriminability and transferability of pre-trained knowledge. To explicitly identify transferable patterns and prepare a "vocabulary bank" for modeling, we decompose the ego-graph $\mathbf{g}_u^{\mathcal{S}}$ of a given node $u$ into $K$ factor-aware subgraphs (vocabularies) with a differentiable neighbor soft-routing mechanism [62, 56].

Denote the multi-channel encoder as $f_{\boldsymbol{\Theta}}$: $\mathbf{g}_u^{\mathcal{S}} \mapsto \{\mathbf{h}_{u,k}^{\mathcal{S}} \in \mathbb{R}^d\}_{k=1}^K$, where each channel $k$ encodes a disentangled factor of the ego-neighbors. The $t$-th soft-routing iteration aims to find an attention:

$$\alpha_{v \to k}^{(t)} \propto \mathrm{Softmax}_k\big(\langle \mathbf{h}_{u,k}^{\mathcal{S}(t)}, \mathbf{h}_{v,k}^{\mathcal{S}(t)} \rangle / \tau\big), \quad \text{s.t. } \sum_{k=1}^K \alpha_{v \to k}^{(t)} = 1, \tag{2}$$

$$\text{and} \quad \mathbf{h}_{u,k}^{\mathcal{S}(0)} = \sigma\big(\mathbf{W}_k^\top \widehat{\mathbf{x}}_u^{\mathcal{S}} + \mathbf{b}_k\big)/\|\sigma\big(\mathbf{W}_k^\top \widehat{\mathbf{x}}_u^{\mathcal{S}} + \mathbf{b}_k\big)\|_2, \quad \text{iff } \|\sigma\big(\mathbf{W}_k^\top \widehat{\mathbf{x}}_u^{\mathcal{S}} + \mathbf{b}_k\big)\|_2 \geqslant \rho. \tag{3}$$

where $\tau$ is the temperature, $\mathbf{W}_k$ and $\mathbf{b}_k$ are projections. The routed neighbors form $K$ vocabularies:

$$\mathbf{g}_u^{\mathcal{S}(t)} = \big\{\mathbf{g}_{u,k}^{\mathcal{S}(t)}\big\}_{k=1}^K, \quad \mathbf{g}_{u,k}^{\mathcal{S}(t)} = \big(\mathcal{V}_{u,k}^{(t)}, \mathcal{E}_{u,k}^{(t)}\big), \quad \mathcal{V}_{u,k}^{(t)} = \big\{v \in \mathcal{N}(u) \mid \alpha_{v \to k}^{(t)}\big\}. \tag{4}$$

The updated $\mathbf{h}_u^{\mathcal{S}}$ is the concatenation of that from each disentangled ego-graph over $T$ iterations:

$$\mathbf{h}_u^{\mathcal{S}(T)} = \big\|_{k=1}^K \mathbf{h}_{u,k}^{\mathcal{S}(T)}, \quad \mathbf{h}_{u,k}^{\mathcal{S}(t+1)} := \mathbf{h}_{u,k}^{\mathcal{S}(t)} + \sum_{v \in \mathcal{V}_{u,k}^{(t)}} \big[\alpha_{v \to k}^{(t)} \mathbf{h}_{v,k}^{\mathcal{S}(t)}\big]. \tag{5}$$

**Semantic-Independence Promotion.** To ensure each channel captures distinct semantic factors, we derive a mutual information regularizer between each disentangled vocabulary, which we implemented by leveraging a contrastive-based variational upper bound with the InfoNCE estimator [67]:

$$\mathcal{R}_{\mathrm{MI}}^u = \sum_{i \neq j} I\big(\mathbf{h}_{u,i}^{\mathcal{S}}; \mathbf{h}_{u,j}^{\mathcal{S}}\big) \overset{\text{Proof C.1}}{\leqslant} \sum_{i \neq j} \mathbb{E}_{u \in \mathcal{B}} \Big[ -\log\Big( \mathrm{Softmax}_v \big(\langle \mathbf{h}_{u,i}^{\mathcal{S}}, \mathbf{h}_{v,j}^{\mathcal{S}} \rangle / \tau\big)\big|_{v=u}\Big)\Big], \tag{6}$$

where $\mathcal{B}$ is a batch of sampled nodes from source domains. Intuitively, $\mathcal{R}_{\mathrm{MI}}^u$ constrains vocabularies to be non-aligned across nodes, thus discouraging shared semantics and keeping distinguished attributes.

**Vocabulary Transferability Justification.** To investigate how different vocabulary combinations contribute to pre-training, we evaluate their transferability by measuring the stability of encoder $f$.

---

**Proposition 1 (Vocabulary Transferability).** *Given any nodes $u$, $v$, and assume $\|\widehat{\mathbf{x}}_u^{\mathcal{S}} - \widehat{\mathbf{x}}_v^{\mathcal{S}}\|_2 \leqslant \epsilon$. The semantic discrepancies $\Delta = \|f(\mathbf{g}_u^{\mathcal{S}}) - f(\mathbf{g}_v^{\mathcal{S}})\|_2$ is upper bounded by:*

$$\Delta \overset{[a]\ \text{Proof C.2}}{\leqslant} \Big[ \min_{\Pi \in S_K} \sum_{k=1}^K \big\|\mathbf{h}_{u,k}^{\mathcal{S}} - \mathbf{h}_{v,\Pi(k)}^{\mathcal{S}}\big\|_2^2 + \psi(\mathcal{R}_{\mathrm{MI}}^{u,v}) \Big]^{\frac{1}{2}} \overset{[b]\ \text{Proof C.3}}{\leqslant} \epsilon \sqrt{K} \left( \frac{C_\sigma L_{\mathbf{W}} L_s}{4 \rho \tau} \right)^T, \tag{7}$$

*where $\Pi$ denotes a permutation over channels, $\psi(\cdot)$ is a slack function controls bound tightness. $C_\sigma$, $L_{\mathbf{W}}$, and $L_s$ are Lipschitz constants for activation function $\sigma$, weight matrix $\mathbf{W}$, and cosine similarity $\langle \cdot, \cdot \rangle$ (inner product), respectively. $\rho$ is the lower bound of $L_2$-norm in Eq. (3).*

---

Proposition 1 states that the upper bound of semantic discrepancy is governed by a specific combination of vocabularies. This demonstrates that disentangled ego-graphs can function as independent and minimal semantic units, facilitating effective knowledge transfer and providing us with theoretical foundations for vocabulary-based generative augmentations.

**Objectives of Self-supervised Pre-training.** Inspired by prior studies [58], we adopt a fully self-supervised contrastive link prediction as the pre-training task. To address task heterogeneity, we build upon a universal task template based on ego-graph semantic similarity, which unifies both link prediction with node- or graph-level classification tasks under a shared framework.

Specifically, we construct a set of quadruples $(u, v^+, v^-, \mathbf{y}_u)$ sampled non-redundantly from $\{G^{\mathcal{S}}\}$, where $v^+, v^-$ are the positive and negative neighbor that link (do not link) to $u$. The binary label $\mathbf{y}_u$ denotes the link existence and is determined by the pairwise similarity discriminator $g = \mathrm{MLP}(\langle \cdot, \cdot \rangle)$. The objective is to encourage higher semantic similarity between $u$ and its positive neighbor $\mathbf{h}_{v^+}^{\mathcal{S}}$, while suppressing that with the negative neighbor $\mathbf{h}_{v^-}^{\mathcal{S}}$. The training objective is formalized as:

$$\mathcal{L}_{\mathrm{pre}}(\boldsymbol{\Theta}; \mathbf{H}^{\mathcal{S}}) = - \sum_{(u, v^+, v^-)} \left[ \log \frac{\exp(g(\mathbf{h}_u^{\mathcal{S}}, \mathbf{h}_{v^+}^{\mathcal{S}})/\tau)}{\exp(g(\mathbf{h}_u^{\mathcal{S}}, \mathbf{h}_{v^+}^{\mathcal{S}})/\tau) + \exp(g(\mathbf{h}_u^{\mathcal{S}}, \mathbf{h}_{v^-}^{\mathcal{S}})/\tau)} \right] + \lambda \cdot \sum_u \mathcal{R}_{\mathrm{MI}}^u, \quad (8)$$

where $\lambda$ is the trade-off hyperparameter. $\boldsymbol{\Theta}^{\star}$ is fixed once pre-training converges.

## 4.2 Vocabulary Modeling with Generative Graphon Experts

We denote each vocabulary extracted from the pre-training dataset $G^{\mathcal{S}}$ as a tuple $(\boldsymbol{\mathcal{A}}_i, \boldsymbol{\mathcal{X}}_i, \mathcal{Y}_i, G^{\mathcal{S}})$, where $\boldsymbol{\mathcal{A}}_i$ and $\boldsymbol{\mathcal{X}}_i$ represent its adjacency and node feature matrix, $\mathcal{Y}_i$ is class label. To enhance the generalization of disentangled vocabularies, enabling efficient and robust generative augmentation during fine-tuning, we establish a *"vocabulary bank"* equipped with two complementary graphon experts: $W^{\boldsymbol{\mathcal{A}}}$ dedicated to modeling vocabularies by structure tokens and $W^{\boldsymbol{\mathcal{X}}}$ to feature tokens.

**Structure Token Modeling.** For each class $c \in \{1, \cdots, \mathcal{C}\}$, we collect adjacencies $\{\boldsymbol{\mathcal{A}}_i^{(c)}\}_{i=1}^{N_c}$ for each disentangled vocabulary, and approximate the structure token via a nonparametric graphon:

$$W_c^{\boldsymbol{\mathcal{A}}} : [0,1]^2 \mapsto [0,1], \quad W_c^{\boldsymbol{\mathcal{A}}}(u, v) = \frac{1}{N_c} \sum_{i=1}^{N_c} \boldsymbol{\mathcal{A}}_i^{(c)}[\pi_i(u), \pi_i(v)], \quad (9)$$

where $\pi_i : [0,1] \mapsto \{1, \cdots, |\boldsymbol{\mathcal{A}}_i|\}$ is a measure-preserving mapping estimated via empirical node ordering (*e.g.*, node degree). $|\boldsymbol{\mathcal{A}}_i|$ denotes the number of nodes in this vocabulary structure token, and we align each vocabulary with the same number of nodes by utilizing node padding [21, 7].

**Feature Token Modeling.** Simultaneously, for each class $c$, we estimate a feature token graphon to model the distribution of node features conditioned on latent positions (coordinates):

$$W_c^{\boldsymbol{\mathcal{X}}} : [0,1] \mapsto \mathbb{R}^d, \quad W_c^{\boldsymbol{\mathcal{X}}}(u) = \frac{1}{N_c} \sum_{i=1}^{N_c} \boldsymbol{\mathcal{X}}_i^{(c)}[\pi_i(u), :], \quad (10)$$

where $\pi_i$ is shared with $W_c^{\boldsymbol{\mathcal{A}}}$. This joint token-level parameterization establishes a critical foundation for a hierarchical collaboration mechanism introduced in the next section. By coherently generating both structure and feature tokens, the unified modeling facilitates more stable knowledge transfer.

**Conditional Vocabulary Generation.** In linguistics, a word typically consists of a *root* and *affixes*, where the root conveys semantics, and the affixes determine its grammatical or functional properties. In a similar vein, a graph vocabulary follows a comparable *"root + affix"* paradigm: *root* (structures) determines its structural semantics (triad, grid, tree, *etc.*), and *affix* (features) represents its domain-specific properties. For instance, a triad can indicate a stable social relationship, but in molecular contexts, it can indicate chemical instability. Based on this understanding, we next illustrate how to generate a graph vocabulary conditioned on a given class within a specific domain.

For dataset $G^{\mathcal{S}}$, given a class label $c$, we can sample $(\widetilde{\boldsymbol{\mathcal{A}}}, \widetilde{\boldsymbol{\mathcal{X}}})$ as a new synthetic graph vocabulary by first drawing $n'$ latent positions: $\mathbf{u}_1, \cdots, \mathbf{u}_{n'} \sim \mathcal{U}[0,1]$, where $\mathcal{U}$ is the uniform distribution. Then:

$$\widetilde{\boldsymbol{\mathcal{A}}}[i, j] \sim \mathrm{Bern}\left(W_c^{\boldsymbol{\mathcal{A}}}(\mathbf{u}_i, \mathbf{u}_j)\right), \quad \widetilde{\boldsymbol{\mathcal{X}}}[i, :] = W_c^{\boldsymbol{\mathcal{X}}}(\mathbf{u}_i), \quad (11)$$

where $\mathrm{Bern}$ denotes the Bernoulli distribution. The generated vocabulary inherits both the structural priors and semantic traits of the specified class and domain. We depict this process in Figure 2 (right).

We theoretically demonstrate that the generated vocabularies converge in distribution to their empirical distribution, ensuring transferability and semantic consistency. We formulate this in Proposition 2.

**Proposition 2** (**Generation Distributional Convergence**). *Let* $\mathbf{g}_c^{\text{emp}}$ *be the empirical distribution over* $n'$*-node vocabulary subgraphs of class* $c$*, collected from the disentangled vocabulary bank:*

$$\mathbf{g}_c^{\text{emp}} := \frac{1}{N_c} \sum_{i=1}^{N_c} \delta_{(\mathcal{A}_i^{(c)}, \mathcal{X}_i^{(c)})}, \quad each \ (\mathcal{A}_i^{(c)}, \mathcal{X}_i^{(c)}) \in \{0,1\}^{n' \times n'} \times \mathbb{R}^{n' \times d}, \quad (12)$$

*where* $\delta_{(\cdot)}$ *denotes Dirac measure, representing a point mass probability distribution. Assume the true underlying structure and feature functions* $W_c^{\mathcal{A}\star}$, $W_c^{\mathcal{X}\star}$ *are bounded, Lipschitz, and satisfy permutation equivariance. If the estimators* $W_c^{\mathcal{A}}$, $W_c^{\mathcal{X}}$ *converge uniformly in* $L_\infty$ *norm, then the total variation (TV) distance between* $\mathbf{g}_c^{\text{gen}} = (\widetilde{\mathcal{A}}_c, \widetilde{\mathcal{X}}_c)$ *and* $\mathbf{g}_c^{\text{emp}}$ *vanishes as* $N_c \to \infty$:

$$\|\mathbf{g}_c^{\text{gen}} - \mathbf{g}_c^{\text{emp}}\|_{\text{TV}} \to 0. \quad (\text{Proof C.4}) \quad (13)$$

## 4.3 Fine-tuning with Augmented Support Samples

Given any sample (node or graph) $G_i^{\mathcal{T}} = (\mathcal{V}_i, \mathcal{E}_i)$ ($i \leqslant m$) with feature $\mathbf{X}_i^{\mathcal{T}} \in \mathbb{R}^{N_i \times d_i}$ from the target domain $D_j \in \{D^{\mathcal{T}}\}$, we utilize the shared LLM($\cdot$), aligners $\mathbf{W}_i$, $\mathcal{F}$ to obtain unified $\widehat{\mathbf{X}}_i^{\mathcal{T}} \in \mathbb{R}^{N_i \times d}$.

**MoE-CoE Network for Selective Augmentation.** Fine-tuning GFMs under few-shot settings is highly non-trivial. The scarcity of labeled samples makes it difficult to adapt large pre-trained models by tuning limited parameters, often resulting in unstable performance and overfitting. To alleviate this, we explicitly incorporate transferable knowledge from source domains while avoiding negative transfer, through a hierarchical routing mechanism termed as MoE-CoE. We propose a two-stage architecture consisting of a Mixture-of-Experts (MoE) layer and a Collaboration-of-Experts (CoE) layer, which together determine how pre-trained vocabulary tokens are reused for target adaptation.

MoE addresses the question of ***where to route*** by selecting relevant domains, where routing weights are assigned across $n$ source-specific vocabulary banks to enable coarse-level domain alignment. CoE addresses the question of ***how to compose*** by promoting collaboration among class-level token experts within each selected domain, which allows multiple tokens to contribute jointly, capturing complementary semantics and enhancing representation coverage. Specifically, assigned weights are:

$$\mathbf{S}_{\text{M}} \in \mathbb{R}^n = \text{Softmax}\left(\mathbf{W}_{\text{M}}^\top \cdot \phi(\widehat{\mathbf{X}}_i^{\mathcal{T}})\right), \quad \mathbf{S}_{\text{C}} \in \mathbb{R}^{\mathcal{C}} = \text{Softmax}\left(\mathbf{W}_{\text{C}}^\top \cdot (\phi(\widehat{\mathbf{X}}_i^{\mathcal{T}} \| \widetilde{\mathcal{X}})\right), \quad (14)$$

where $\mathbf{W}_{\text{M}}$ and $\mathbf{W}_{\text{C}}$ are projections, $\phi(\cdot)$ is a learnable network. The composed vocabulary is:

$$\widetilde{\mathbf{g}}_i^{\text{gen}} \stackrel{\text{def}}{=} (\widetilde{\mathcal{A}}_i^{\text{gen}}, \widetilde{\mathcal{X}}_i^{\text{gen}}) = \sum_{i=1}^n \mathbf{S}_{\text{M},i}^\top \left(\sum_{c=1}^{\mathcal{C}} \mathbf{S}_{\text{C},c}^\top \cdot \mathbf{g}_c^{\text{gen}}\right), \quad \widetilde{G}_i^{\mathcal{T}} = G_i^{\mathcal{T}} \oplus \widetilde{\mathbf{g}}_i^{\text{gen}}, \quad (15)$$

where $\oplus$ denotes augmenting the support sample $G_i$ with the generated vocabulary by overlapping on the max-degree node for aggregation-enhanced alignment. To mitigate load imbalance and encourage sparse activation in the MoE-CoE routing architecture, the weight assignment vectors $\mathbf{S}_{\text{M}}, \mathbf{S}_{\text{C}}$ are optimized in a supervised manner by minimizing their selective uncertainty:

$$\mathcal{L}_{\text{MoE-CoE}}(\mathbf{S}_{\text{M}}, \mathbf{S}_{\text{C}}) = -\sum_{i=1}^n \mathbf{S}_{\text{M},i}^\top \log(\mathbf{S}_{\text{M},i}) - n \sum_{c=1}^{\mathcal{C}} \mathbf{S}_{\text{C},c}^\top \log(\mathbf{S}_{\text{C},c}). \quad (16)$$

**Objectives of Few-shot Fine-tuning.** Motivated by prompt learning [6], we introduce learnable graph prompts as few-shot adapters to retrieve relevant knowledge:

$$\mathbf{H}_i^{\mathcal{T}} = f_{\Theta}^\star\left(\mathcal{P}_{\Omega} \odot \widetilde{G}_i^{\mathcal{T}}\right), \quad (17)$$

where $\mathcal{P}$ is a graph prompt initialized with tunable parameter $\Omega$, optimized over $m$-shot augmented support samples. The graph prompt serves as a lightweight adapter, selectively injecting transferable semantics while preserving domain specificity. To support both node- and graph-level classification, we adopt a unified task template across pre-training and fine-tuning based on ego-graph similarity.

Given $m$ ($m \ll \sum_{i=1}^n N_i$) vocabulary-augmented support samples $\{(\widetilde{G}_i^{\mathcal{T}}, Y_i^{\mathcal{T}})\}_{i=1}^m$, where $\mathbf{H}^{\mathcal{T}}$ denotes the node (for node task) or ego-graph embeddings (for graph task), the fine-tuning objective is transformed into determining the similarity between the query sample and class prototype embedding:

$$\mathcal{L}_{\text{cls}}(\Omega; \mathbf{H}^{\mathcal{T}}) = -\sum_{(\widetilde{G}_i^{\mathcal{T}}, Y_i^{\mathcal{T}})} \left[ \log \frac{\exp\left(g(\mathbf{H}_i^{\mathcal{T}}, \overline{\mathbf{H}}_{Y_i}^{\mathcal{T}})/\tau\right)}{\sum_{Y_j \in \{Y^{\mathcal{T}}\}} \exp\left(g(\mathbf{H}_i^{\mathcal{T}}, \overline{\mathbf{H}}_{Y_j}^{\mathcal{T}})/\tau\right)} \right], \quad (18)$$

where $\overline{\mathbf{H}}_{Y_i}^{\mathcal{T}}$ is the class prototype for samples in class $Y_i$. The overall fine-tuning objective is:

$$\mathcal{L}_{\text{ftn}} = \mathcal{L}_{\text{cls}}(\Omega; \mathbf{H}^{\mathcal{T}}) + \mu \cdot \mathcal{L}_{\text{MoE-CoE}}(\mathbf{S}_{\text{M}}, \mathbf{S}_{\text{C}}), \quad (19)$$

where $\mu$ is the trade-off hyperparameter.

# 5 Experiments

In this section, we conduct extensive experiments to evaluate *effectiveness*, *efficiency*, and *robustness* of the proposed GRAVER[2]. We focus on the following research questions:

- **RQ1:** How effective and robust is the few-shot classification fine-tuning? ($\triangleright$ Section 5.2)
- **RQ2:** Which module contributes most to the classification performance? ($\triangleright$ Section 5.3)
- **RQ3:** How time-efficient in the few-shot fine-tuning phase? ($\triangleright$ Section 5.4)
- **RQ4:** How LLMs enhance domain alignment to enable zero-shot transferability? ($\triangleright$ Section 5.5)
- **RQ5:** How sensitive is the performance to hyperparameter fluctuations? ($\triangleright$ Section 5.6)

## 5.1 Experimental Settings

**Datasets.** To highlight the multi-domain pre-training capability of GFMs, we select **seven** benchmark graph datasets from **three** distinct domains. It is worth noting that this differs from conventional experimental settings, which typically treat a single dataset as a domain. Specifically:

- **Citation Domain:** `Cora` [65], `CiteSeer` [16], `PubMed` [73], and the large-scale `ogbn-arXiv` [28].
- **Co-purchase Domain:** `ogbn-Tech` and `ogbn-Home` from large-scale co-purchase network [28].
- **Web Link Domain:** `Wiki-CS` [66], a hyperlink network constructed from a subset of Wikipedia.

**Baselines.** We compare GRAVER with **15** state-of-the-art baselines from **four** primary categories.

- **Vanilla Graph Neural Networks:** including `GCN` [40] and `GAT` [91] without pre-training.
- **Self-Supervised Graph Pre-training:** `GCC` [69], `DGI` [92], `InfoGraph` [81], `GraphCL` [116], `DSSL` [102], and `GraphACL` [103], which utilize self-supervised objectives to pre-train and finetune.
- **Prompt-based Graph Fine-tuning:** we select pioneering prompt-based fine-tuning baselines including `GPPT` [82], `GraphPrompt` [58], `GPF` [14], and `ProNoG` [121].
- **Multi-Domain Graph Pre-training:** we select the most closely related baselines on multi-domain graph pre-training and finetuning, including `GCOPE` [136], `MDGPT` [123], and `SAMGPT` [119].

**Multi-Domain Pre-training Settings.** We mainly focus on the challenging task of adapting GFMs to unseen datasets and domains in downstream applications. To better characterize the setting, we explicitly distinguish between *cross-dataset* and *cross-domain* adaptation:

- **Cross-Dataset:** pre-training and fine-tuning on *different* datasets, which are from the *same* domain. Specifically, take **Citation**, **Co-purchase**, and **Web Link** as the pre-training domains. When each dataset is used as the target alternatively, the remainings are collectively used as the source datasets.
- **Cross-Domain:** extends to a more challenging scenario, where the target dataset belongs to an *unseen* domain different from the source domain. Specifically, we alternately set **Citation**, **Co-purchase**, and **Web Link** as the target domain, while the remaining two domains are the source.

**Few-shot Fine-tuning Settings.** We evaluate node and graph classification under the $m$-shot setting, where $m$ labeled samples per class are randomly selected. As each dataset contains a single large graph, we follow prior work [61, 118, 123] to extract ego-graphs centered on target nodes for graph classification, assigning labels based on the central nodes. Accuracy (Acc.) is used for evaluation.

## 5.2 RQ1: Few-Shot Node and Graph Classification

**One-Shot Classification** (Table 1). ❶ Overall, GRAVER achieves the best accuracy, demonstrating a clear superiority. Specifically, it outperforms the runner-up with average gains of 2.8% (node) and 3.2% (graph), which are consistent across both dataset and domain transfer scenarios, indicating effective source knowledge utilization even under extreme one-shot conditions. ❷ Compared to strong baselines like `SAMGPT` and `MDGPT`, GRAVER achieves significantly higher performance by over 4% in both tasks. The performance gap widens further in harder domains like `ogbn-arXiv` and `Wiki-CS`, demonstrating its broad applicability. Notably, GRAVER still leads with large margins in the more challenging cross-domain settings due to domain shift. ❸ On robustness, GRAVER consistently exhibits an obviously smaller standard deviation compared to baselines with an average relative decrease of 54.0% (node) and 54.6% (graph). This suggests GRAVER maintains stable performance regardless of the random support set selection, thanks to the generative vocabulary augmentations, and is beneficial for real-world deployment. ❹ Additional results under the **five-shot** setting, presented in Appendix F.1, consistently exhibit the same performance trend as observed in the one-shot scenario.

---

[2]Codes are available at: `https://github.com/RingBDStack/GRAVER`.

Table 1: Accuracy (% ± std for 20 runs) of **one-shot classification**. Best results are presented in **bold** and the runner-ups are underlined. `CR` = Cora, `CS` = CiteSeer, `PM` = PubMed, `arXiv` = ogbn-arXiv, `Tech` = ogbn-tech, `Home` = ogbn-home, `Wiki` = Wiki-CS. Color denotes domain.

| Source | Cross-Dataset | | | | Cross-Domain | | |
|---|---|---|---|---|---|---|---|
| | `CS` `PM` `Home` `Wiki` | `CR` `PM` `Home` `Wiki` | `CR` `CS` `Home` `Wiki` | `CR` `CS` `PM` `Home` `Wiki` | `CR` `CS` `PM` `Wiki` | `CR` `CS` `PM` `Home` | `Home` `Tech` `Wiki` |
| **Model / Target** | CR | CS | PM | Tech | Home | Wiki | arXiv |
| **Node Classification** | | | | | | | |
| GCN (bb.) [40] | 28.40 ± 4.62 | 29.25 ± 3.39 | 40.33 ± 6.90 | 61.59 ± 5.13 | 53.89 ± 3.35 | 36.74 ± 2.53 | 28.58 ± 5.39 |
| GAT [91] | 29.72 ± 5.17 | 29.31 ± 3.47 | 40.51 ± 3.95 | 61.98 ± 5.23 | 51.70 ± 3.96 | 36.24 ± 4.19 | 29.03 ± 5.75 |
| GCC [69] | 32.47 ± 4.55 | 32.78 ± 3.85 | 41.66 ± 3.27 | 64.74 ± 3.78 | 55.36 ± 5.86 | 37.66 ± 3.80 | 30.42 ± 5.54 |
| DGI [92] | 30.77 ± 3.92 | 31.41 ± 4.11 | 39.97 ± 5.90 | 63.76 ± 3.77 | 53.31 ± 2.58 | 39.20 ± 5.67 | 32.15 ± 4.91 |
| GraphCL [116] | 33.64 ± 5.75 | 28.20 ± 3.13 | 39.03 ± 8.67 | 62.44 ± 6.55 | 51.55 ± 8.20 | 38.05 ± 3.30 | 31.81 ± 5.04 |
| DSSL [102] | 29.76 ± 4.55 | 30.84 ± 7.62 | 39.99 ± 6.29 | 61.27 ± 5.21 | 51.90 ± 6.52 | 37.58 ± 6.78 | 28.14 ± 5.17 |
| GraphACL [103] | 35.92 ± 5.61 | 33.88 ± 6.69 | 42.73 ± 5.26 | 66.70 ± 4.29 | 58.01 ± 6.32 | 40.94 ± 6.65 | 35.57 ± 4.29 |
| GPPT [82] | 32.38 ± 5.74 | 31.78 ± 4.61 | 41.97 ± 5.48 | 65.81 ± 6.56 | 57.81 ± 5.36 | 40.97 ± 5.41 | 34.22 ± 6.51 |
| GraphPrompt [58] | 38.02 ± 6.52 | 34.57 ± 6.34 | 46.19 ± 7.63 | 70.11 ± 6.62 | 60.99 ± 7.19 | 44.02 ± 6.74 | 40.74 ± 6.56 |
| GPF [14] | 40.01 ± 8.21 | 40.07 ± 7.64 | 47.58 ± 5.83 | 70.07 ± 6.55 | 59.05 ± 5.85 | 44.62 ± 8.68 | 40.13 ± 5.07 |
| ProNoG [121] | 43.67 ± 6.33 | 40.34 ± 7.11 | 51.35 ± 7.16 | 73.49 ± 7.34 | 62.77 ± 6.10 | 45.65 ± 6.44 | 41.15 ± 4.21 |
| GCOPE [136] | 36.27 ± 3.93 | 40.42 ± 4.64 | 44.75 ± 4.67 | 71.26 ± 5.95 | 60.39 ± 6.08 | 41.80 ± 4.96 | 40.00 ± 6.53 |
| MDGPT [123] | 42.55 ± 6.84 | 37.92 ± 7.18 | 51.03 ± 8.99 | 72.10 ± 7.12 | 62.69 ± 7.29 | 45.93 ± 6.24 | 43.33 ± 5.75 |
| SAMGPT [119] | 46.79 ± 6.54 | 38.65 ± 6.35 | 51.92 ± 9.50 | 73.60 ± 7.55 | 64.32 ± 7.02 | 46.03 ± 6.98 | 45.27 ± 5.05 |
| **GRAVER (ours)** | **48.00 ± 2.52** | **41.13 ± 3.00** | **55.30 ± 3.03** | **77.62 ± 2.94** | **67.12 ± 3.18** | **49.33 ± 2.44** | **49.25 ± 3.43** |
| **Graph Classification** | | | | | | | |
| GCN (bb.) [40] | 40.07 ± 4.78 | 29.53 ± 5.74 | 45.25 ± 7.32 | 81.16 ± 4.56 | 58.82 ± 2.48 | 38.46 ± 4.47 | 46.05 ± 4.42 |
| GAT [91] | 36.01 ± 5.09 | 25.98 ± 7.84 | 41.03 ± 5.77 | 77.75 ± 4.92 | 59.12 ± 5.97 | 38.49 ± 5.11 | 40.80 ± 6.87 |
| InfoGraph [81] | 42.18 ± 5.17 | 30.18 ± 4.11 | 49.10 ± 5.42 | 81.62 ± 8.35 | 63.31 ± 3.14 | 40.98 ± 4.24 | 43.96 ± 6.48 |
| GraphCL [116] | 39.56 ± 5.79 | 32.61 ± 6.54 | 47.71 ± 7.04 | 82.27 ± 4.46 | 63.77 ± 4.66 | 46.14 ± 3.10 | 47.61 ± 5.02 |
| DSSL [102] | 41.46 ± 7.91 | 32.92 ± 5.24 | 47.28 ± 7.71 | 83.66 ± 4.99 | 60.56 ± 5.39 | 41.09 ± 5.18 | 47.50 ± 5.20 |
| GraphACL [103] | 40.60 ± 5.00 | 33.17 ± 7.53 | 50.48 ± 5.77 | 80.98 ± 3.78 | 62.38 ± 8.05 | 45.01 ± 5.62 | 46.43 ± 6.32 |
| GraphPrompt [58] | 41.71 ± 3.45 | 38.76 ± 7.06 | 50.61 ± 6.39 | 84.23 ± 2.43 | 61.37 ± 5.64 | 43.98 ± 8.22 | 47.98 ± 5.40 |
| GPF [14] | 41.25 ± 9.41 | 38.18 ± 8.22 | 47.83 ± 8.17 | 83.72 ± 5.42 | 67.63 ± 4.72 | 47.16 ± 5.08 | 47.27 ± 4.16 |
| ProNoG [121] | 51.72 ± 8.20 | 39.87 ± 5.89 | 54.13 ± 8.93 | 82.55 ± 3.19 | 68.95 ± 4.20 | 47.20 ± 4.83 | 52.11 ± 5.58 |
| GCOPE [136] | 55.91 ± 7.37 | 40.99 ± 9.01 | 54.36 ± 8.63 | 82.58 ± 2.39 | 67.45 ± 4.39 | 49.26 ± 5.25 | 62.90 ± 3.64 |
| MDGPT [123] | 52.77 ± 6.71 | 41.01 ± 9.72 | 55.47 ± 8.27 | 83.91 ± 4.06 | 70.46 ± 5.21 | 50.24 ± 7.49 | 59.29 ± 4.25 |
| SAMGPT [119] | 53.25 ± 4.28 | 42.39 ± 7.28 | 57.66 ± 6.34 | 85.08 ± 6.39 | 73.69 ± 5.06 | 52.10 ± 3.65 | 62.28 ± 4.05 |
| **GRAVER (ours)** | **59.20 ± 2.38** | **45.50 ± 2.00** | **61.70 ± 2.58** | **87.33 ± 1.84** | **75.46 ± 2.24** | **55.90 ± 2.33** | **67.10 ± 3.26** |

$m$**-Shot Classification** (Figure 3). To assess performance under varying $m$, we compare with three strongest baselines from Table 1: `GCOPE`, `MDGPT`, and `SAMGPT`. Results show: ❶ GRAVER consistently outperforms three baselines across nearly all $m$-shot settings. ❷ The advantage is most pronounced under low-shot conditions ($m \leqslant 4$), where the generative vocabulary introduces semantically diverse and transferable patterns to enhance adaptation. As $m$ increases, the relative gain narrows due to reduced augmentation benefits and increasing noise. ❸ Similar trends are observed in both cross-dataset and cross-domain scenarios, confirming GRAVER's generalizability across transfer settings. ❹ Additional results of **graph classification** in Appendix F.1 further support these findings.

**Robustness against Random Noise** (Figure 4). To evaluate robustness under perturbations, we add Gaussian feature noise and random edge modifications to support sets, controlled by parameters $\lambda_f$ and $\lambda_s$. Results show: ❶ GRAVER maintains superior performance under both feature and structure noise, showing robustness to both support sample variability and direct corruption. ❷ While baselines degrade more rapidly with increasing noise, especially in structure, GRAVER shows only gradual decline, or even slight increase, highlighting the stabilizing effect of vocabulary-guided augmentation. ❸ The widening performance gap under higher noise levels indicates that the MoE-CoE architecture effectively filters out irrelevant patterns and reinforces useful knowledge for robust fine-tuning.

## 5.3 RQ2: Ablation Study

We conduct ablation studies on **three** core components of GRAVER: the semantic-independence promotion (*SIP*), the vocabulary augmentation (*VA*), and the MoE-CoE network (*MC*). Results (Figure 5) show that: ❶ Removing *SIP* leads to a minor yet consistent drop, indicating that encouraging semantic diversity across vocabulary channels is beneficial, though not dominant. ❷ Disabling *VA* causes a major decline, highlighting the importance of generative subgraphs in enhancing support quality under low-shot conditions. ❸ Replacing *MC* with uniform fusion results in significant degradation, confirming that selective routing is crucial for injecting relevant knowledge while suppressing noise. ❹ Additional results on other datasets in Appendix F.2 demonstrate similar conclusions.

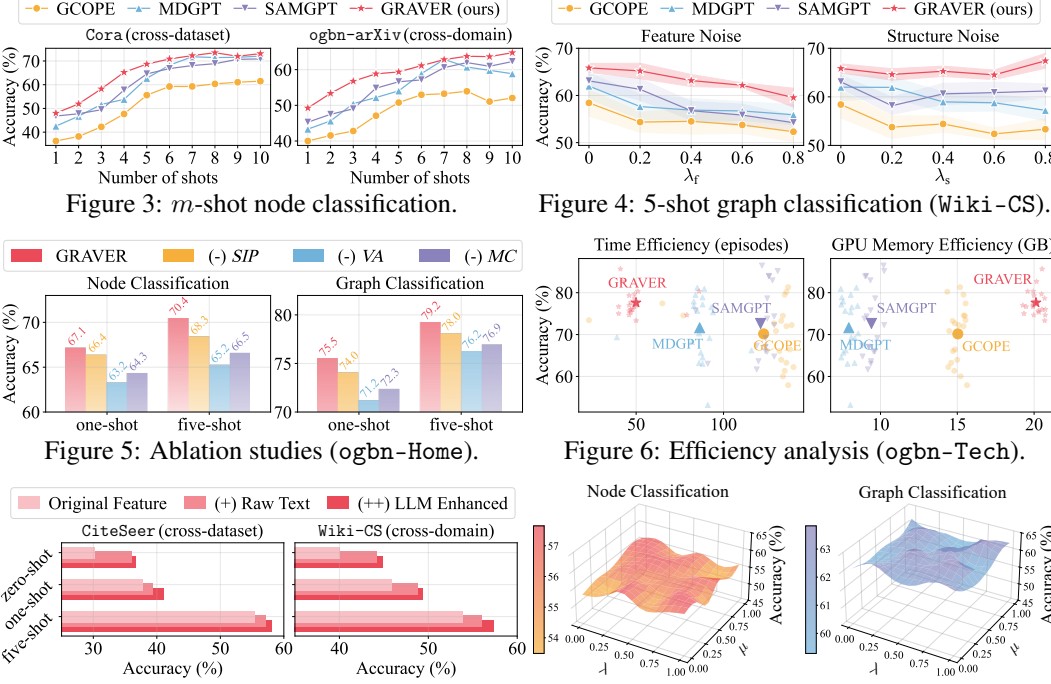

Figure 3: $m$-shot node classification.

Figure 4: 5-shot graph classification (`Wiki-CS`).

Figure 5: Ablation studies (`ogbn-Home`).

Figure 6: Efficiency analysis (`ogbn-Tech`).

Figure 7: Analysis on LLM (node classification).   Figure 8: Hyperparameter sensitivity (`PubMed`).

## 5.4   RQ3: Efficiency Evaluation for Fine-tuning

In addition to the theoretical complexity analysis presented in Appendix B, we empirically analyze the time and memory efficiency of one-shot node classification, where the results (Figure 6) show that: ❶ GRAVER converges in nearly half the episodes compared to baselines while achieving higher accuracy, demonstrating the efficiency of its generative vocabulary and lightweight MoE-CoE routing design. ❷ Though GRAVER consumes more GPU memory due to maintaining and routing richer vocabulary tokens, the trade-off is acceptable given its superior performance and faster convergence.

## 5.5   RQ4: LLM-Enhanced Multi-Domain Alignment

We enhance node features by incorporating raw text and LLM-generated class-aware descriptions. Results (Figure 7) indicate that, raw text alone brings substantial improvement in zero-shot settings, as it serves as a domain-agnostic semantic bridge when no labeled data is available. This highlights the crucial role of textual information in aligning feature spaces across domains purely through inherent semantics. Building on this, LLM-enhanced features further boost low-shot performance by injecting class-discriminative signals. These improvements are consistent across both cross-dataset and cross-domain tasks, underscoring the effectiveness of text-based feature enhancement.

## 5.6   RQ5: Hyperparameter Sensitivity Investigation

We evaluate the sensitivity of GRAVER to two important hyperparameters: $\lambda$ (for $\mathcal{R}_{\mathrm{MI}}$ in Eq. (8)) and $\mu$ (for $\mathcal{L}_{\mathrm{MoE\text{-}CoE}}$ in Eq. (19)). Results (Figure 8) show that accuracy reveals only mild fluctuations within a broad range of values for both node and graph classification tasks. This indicates GRAVER is robust to hyperparameter combinations. We provide detailed hyperparameter settings in Appendix E.

## 6   Conclusion

We propose **GRAVER**, a trustworthy graph foundation model that enables robust and efficient fine-tuning under multi-domain pre-training and cross-domain adaptation. To our knowledge, this is the first framework to leverage generative graph vocabularies for in-context support set augmentation to enhance adaptation stability. GRAVER introduces graphon-based generative experts to synthesize structure-feature vocabularies aligned with transferable subgraphs, and employs a hierarchical MoE-CoE to dynamically assemble source-domain knowledge for context-aware augmentation. Extensive experiments demonstrate its multi-task superiority over 15 state-of-the-art baselines.

## Acknowledgments

The corresponding author is Qingyun Sun. This work is supported in part by NSFC under grants No.623B2010, No.62225202, and No.62302023, by NSF under grants III-2106758 and POSE-2346158, by the Fundamental Research Funds for the Central Universities, and by the Academic Excellence Foundation of BUAA for PhD Students. We extend our sincere thanks to all authors for their valuable contributions.

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

# Appendix

## Table of Contents

# A   Notations

| Notations | Descriptions |
| --- | --- |
| $G = (\mathcal{V}, \mathcal{E})$ | Graph $G$ with node set $\mathcal{V}$ and edge set $\mathcal{E}$. |
| $\{G^{\mathcal{S}}\} \in \mathcal{G}^{\mathcal{S}}, \{G^{\mathcal{T}}\} \in \mathcal{G}^{\mathcal{T}}$ | Graphs sampled from source domain ($\mathcal{S}$) and target domains ($\mathcal{T}$). |
| $\{Y^{\mathcal{S}}\} \in \mathcal{Y}^{\mathcal{S}}, \{Y^{\mathcal{T}}\} \in \mathcal{Y}^{\mathcal{T}}$ | Corresponding graph labels in source domain ($\mathcal{S}$) and target domains ($\mathcal{T}$). |
| $\{D^{\mathcal{S}}\} \in \mathcal{D}^{\mathcal{S}}, \{D^{\mathcal{T}}\} \in \mathcal{D}^{\mathcal{T}}$ | Source domains and targeted domains. |
| $\mathbf{A} \in \{0,1\}^{N_i \times N_i}$ | Adjacency matrix $\mathbf{A}$, where $N_i$ is the number of nodes. |
| $\mathbf{X} \in \mathbb{R}^{N_i \times d_i}$ | Node feature matrix $\mathbf{X}$, with $N_i$ nodes and $d_i$ input feature dimensions. |
| $\mathcal{A}, \widetilde{\mathcal{A}} \in \mathbb{R}^{n' \times n'}$ | Adjacency matrix of disentangled and generated vocabulary with $n'$ nodes. |
| $\mathcal{X}, \widetilde{\mathcal{X}} \in \mathbb{R}^{n' \times h}$ | Node feature matrix of disentangled and generated vocabulary with $h$ dim. |
| $\mathbf{X}^{\mathcal{S}}, \widehat{\mathbf{X}}^{\mathcal{S}}, \widehat{\mathbf{x}}_i^{\mathcal{S}} \in \mathbb{R}^{- \times d}$ | Source domain graph features, aligned features, and single node features. |
| $\mathbf{H}^{\mathcal{S}}, \mathbf{h}^{\mathcal{S}} \in \mathbb{R}^{K \cdot h}$ | Encoded source domain graph features and node features. |
| $\mathbf{X}^{\mathcal{T}}, \widehat{\mathbf{X}}^{\mathcal{T}}, \widehat{\mathbf{x}}_i^{\mathcal{T}} \in \mathbb{R}^{- \times d}$ | Target domain graph features, aligned features, and single node features. |
| $\mathbf{H}^{\mathcal{T}}, \mathbf{h}^{\mathcal{T}} \in \mathbb{R}^{K \cdot h}$ | Encoded target domain graph features and node features. |
| $\mathbf{g}_u = \{\mathbf{g}_{u,k}\}_{k=1}^{K}$ | Ego-graph and its $K$ disentangled vocabularies of the central node $u$. |
| $\mathbf{g}^{\text{emp}}, \mathbf{g}^{\text{gen}}, \widetilde{\mathbf{g}}$ | Observed, generated, and weighted mixed generated vocabularies. |
| $f_{\boldsymbol{\Theta}}(\cdot): \mathbb{R}^d \mapsto \mathbb{R}^{K \cdot h}$ | Graph vocabulary encoder with learnable parameter $\boldsymbol{\Theta}$. |
| $g(\cdot): \mathbb{R}^{K \cdot h} \times \mathbb{R}^{K \cdot h} \mapsto \mathbb{R}^2$ | Graph discriminator composed of an inner product $\langle \cdot, \cdot \rangle$ and a binary MLP. |
| $h(\cdot) = g \circ f$ | The graph learner compound of encoder $f(\cdot)$ and binary discriminator $g(\cdot)$ |
| $\mathcal{F}(\cdot): \mathbb{R}^{d_i} \mapsto \mathbb{R}^d$ | Domain feature alignment function from $d_i$ dimension to $d$. |
| $\sigma(\cdot)$ | Non-linear activation function (PReLU). |
| $\phi(\cdot)$ | Learnable routing network in the MoE-CoE network. |
| $\psi(\cdot)$ | The slack function controls bound tightness. |
| $\Pi(\cdot)$ | The set of permutation functions (disentangle channel mappings). |
| $\pi(\cdot)$ | The measure-preserving mapping estimated via empirical node ordering. |
| $I(\cdot; \cdot)$ | The Shannon Mutual Information term. |
| $\mathcal{N}(v)$ | The neighbor nodes of $v$ (1-hop by default). |
| $\mathcal{D}_{\text{KL}}$ | The Kullback-Leibler divergence. |
| $\mathcal{D}_{\text{match}}$ | The matching distance between disentangled channels. |
| $\mathcal{D}_{\text{TV}}$ | The Total Variation (TV) distance between vocabularies. |
| $W_c^{\mathcal{A}}: [0,1]^2 \mapsto [0,1]$ | The nonparametric estimated structure token graphon. |
| $W_c^{\mathcal{X}}: [0,1]^2 \mapsto \mathbb{R}^d$ | The nonparametric estimated feature token graphon. |
| $\mathbf{S}_{\text{M}}, \mathbf{S}_{\text{C}}$ | Assignment weight vector for the MoE and CoE network, respectively. |
| $\mathbf{W}, \mathbf{W}_{\text{M}}, \mathbf{W}_{\text{C}}$ | Learnable weight matrix for non-linear projection. |
| $\mathbf{b}$ | The non-linear projection bias. |
| $\boldsymbol{\mathcal{P}}_{\boldsymbol{\Omega}}$ | Graph prompt with learnable parameter $\boldsymbol{\Omega}$. |
| $\alpha_{i \to k}$ | The routing attention (weights) from node $i$ to channel (factor) $k$. |
| $\oplus$ | The augmentation operations by overlapping with the generated vocabulary. |
| $T$ | The number of disentanglement routing iterations. |
| $C_{\sigma}$ | The Lipschitz constant for the activation function $\sigma$. |
| $L_{\mathbf{W}}$ | The Lipschitz constant for the weight matrix $\mathbf{W}$. |
| $L_s$ | The Lipschitz constant for the cosine similarity $\langle \cdot, \cdot \rangle$. |
| $\Delta$ | The semantic discrepancies of $\|f(\mathbf{g}_u^{\mathcal{S}}) - f(\mathbf{g}_v^{\mathcal{S}})\|_2$. |
| $\delta$ | The Dirac measure, representing a point mass probability distribution. |
| $\epsilon$ | The upper bound of $\|\widetilde{\mathbf{x}}_u^{\mathcal{S}} - \widetilde{\mathbf{x}}_v^{\mathcal{S}}\|_2$. |
| $\rho$ | The lower bound of $L_2$-norm in Eq. (3). |
| $\tau$ | The temperature parameter. |
| $\lambda$ | The trade-off hyperparameter for $\mathcal{R}_{\text{MI}}$ in Eq. (8). |
| $\mu$ | The trade-off hyperparameter for $\mathcal{L}_{\text{MoE-CoE}}$ in Eq. (19). |
| $\mathcal{R}_{\text{MI}}$ | The mutual information regularizer between each disentangled vocabulary. |
| $\mathcal{L}_{\text{pre}}$ | The pre-training loss function. |
| $\mathcal{L}_{\text{MoE-CoE}}$ | Loss function for the MoE-CoE network. |
| $\mathcal{L}_{\text{cls}}$ | Loss function for the downstream classification. |
| $\mathcal{L}_{\text{ftn}}$ | Loss function for the fine-tuning phase few-shot training. |

# B   Algorithm and Complexity Analysis

The pre-training pipeline of GRAVER is illustrated in Algorithm 1, and the fine-tuning pipeline is shown in Algorithm 2. Next, we theoretically analyze their computational complexity, respectively.

---

**Algorithm 1:** Pre-training pipeline of GRAVER.

---

**Input:** $n$ source graphs $\{G^{\mathcal{S}}\} \in \mathcal{G}^{\mathcal{S}}$ from multi-domain $\{D^{\mathcal{S}}\} \in \mathcal{D}^{\mathcal{S}}$, where each of the graph $G_i^{\mathcal{S}}$ is associate with its feature matrix $\mathbf{X}_i \in \mathbb{R}^{N_i \times d_i}$ and adjacency matrix $\mathbf{A} \in \mathbb{R}^{N_i \times N_i}$; Feature dimension aligner $\mathcal{F}(\cdot)$; Domain semantic aligners $\mathbf{W}_i$; Learnable weight matrix $\mathbf{W}_k$ for disentangle channel projection; Number of the latent factors (channels) $K$; Number of the routing iterations $T$; Number of the pre-training epochs $E_1$; Temperature parameter $\tau$; Hyperparameter $\lambda$ for trading-off the mutual information regularization term $\mathcal{R}_{\text{MI}}$.

**Output:** Pre-trained graph encoder and discriminator $g(f_{\Theta}^{\star}(\cdot))$; Structure token graphon $W_c^{\mathcal{A}}$ and feature token graphon $W_c^{\mathcal{X}}$ for each source domain $G^{\mathcal{S}}$ and class $c$.

**1** Initialize all parameters randomly;

**2 for** $e = 1, 2, \cdots, E_1$ **do**

**3**     // domain feature alignment

**4**     Aligned graph feature: $\widehat{\mathbf{X}}_i^{\mathcal{S}} \leftarrow$ Eq. (1) for each $G_i^{\mathcal{S}} \in \{G^{\mathcal{S}}\}$ with $\mathcal{F}$ and $\mathbf{W}_i$;

**5**     // ego-graph disentanglement

**6**     **for** $t = 1, 2, \cdots, T$ and $k = 1, 2, \cdots, K$ **do**

**7**        Initialize projected embedding $\mathbf{h}_{u,k}^{\mathcal{S}(0)} \leftarrow$ Eq. (3) for each node $u$;

**8**        Routing attention for neighbor node $v$ to $k$-th channel (factor): $\alpha_{v \to k}^{(t)} \leftarrow$ Eq. (2);

**9**        Disentangled ego-graphs (vocabularies): $\mathbf{g}_u^{\mathcal{S}(t)} \leftarrow$ Eq. (4) for each node $u$;

**10**        Updated node disentangled embeddings: $\mathbf{h}_u^{\mathcal{S}(t)} \leftarrow$ Eq. (5) for each node $u$;

**11**        Calculate the mutual information regularizer: $\mathcal{R}_{\text{MI}}^u \leftarrow$ Eq. (6) for each node $u$;

**12**     **end**

**13**     // self-supervised pre-training

**14**     Calculate the overall pre-training loss: $\mathcal{L}_{\text{pre}} \leftarrow$ Eq. (8);

**15**     Update $\Theta$ by minimizing $\mathcal{L}_{\text{pre}}$ and back-propagation;

**16**     **if** $e = E_1$ **then**

**17**        // establish the vocabulary bank

**18**        Modeling structure token graphon with observed vocabularies: $W_c^{\mathcal{A}} \leftarrow$ Eq. (9);

**19**        Modeling feature token graphon with observed vocabularies: $W_c^{\mathcal{X}} \leftarrow$ Eq. (10);

**20**     **end**

**21 end**

---

**Complexity Analysis.** We analyze the complexity of each part in Algorithm 1 as follows.

- **Dimension-wise feature alignment:** We implement the aligner function $\mathcal{F} : \mathbb{R}^{d_i} \mapsto \mathbb{R}^d$ by the truncated singular value decomposition (SVD) [80]. We omit this computational complexity as it can be pre-processed without increasing the runtime computational burdens.

- **Semantic-wise feature alignment:** We implemented each aligner $\mathbf{W}_i \in \mathbb{R}^{d \times d}$ by a randomly initialized learnable non-linear projection matrix, where the complexity takes $\mathcal{O}(nd^2)$.

- **Ego-graph disentanglement:** We first project the aligned feature of each node to $K$ embedding space, which takes the complexity of $\mathcal{O}(n|\mathcal{V}_i|Kdh)$, where $|\mathcal{V}_i|$ is the average number of nodes in each of the pre-training graphs. Next, the neighbor routing iterations take the complexity of $\mathcal{O}(n|\mathcal{V}_i|T|\mathcal{N}(u)|Kh)$, where $|\mathcal{N}(u)|$ is the number of neighbor nodes of of $u$. Then, the node disentangled embedding updating takes the complexity of the same $\mathcal{O}(n|\mathcal{V}_i|T|\mathcal{N}(u)|Kh)$. For brevity, denote the average of $|\mathcal{V}_i|$ as $n_1$, and the average of $|\mathcal{N}(u)|$ as $n_2$. Thus, the total complexity of the ego-graph disentanglement takes $\mathcal{O}(nn_1Kh(d + Tn_2))$.

- **Pair-wise mutual information regularizer:** For a pair of factors $(i, j)$, the inner product takes the complexity of $\mathcal{O}(|\mathcal{B}|h)$. For each node, we can enumerate $|\mathcal{B}|K(K-1)$ factor pairs. Then, the total complexity accounting on each node $u$ takes the complexity of $\mathcal{O}(nn_1|\mathcal{B}|^2K(K-1)h)$.

- **Pre-training loss calculation:** Suppose there are $n_3$ direct node neighbors, $n_4$ indirect neighbors (nodes that are connected) averaged for each node in the source domain graphs, we can sample $n \cdot n_3 \cdot n_4$ quadruples, which is utilized to formalize the self-supervised pre-training contrastive loss. Its computational complexity takes $\mathcal{O}(nn_1(n_3 + n_4)Kh)$.

- **Graphon experts estimation:** We implement $\pi_i(\cdot)$ by ordering nodes with degree, which takes the complexity of $\mathcal{O}(N_c n' \log n')$ for each graph $G^{\mathcal{S}}$ and class $c$. For each coordinate $(u, v) \in \{1, \cdots, n'\}$, the estimation takes the complexity of $\mathcal{O}(N_c)$ for $n'^2$ coordinate pairs. Thus, the total complexity takes $\mathcal{O}(n\mathcal{C}(N_c n' \log n' + N_c n'^2))$, where $\mathcal{C}$ is the average number of classes.

The overall computational complexity for the pre-training phase is:

$$\mathcal{O}(nd^2) + \mathcal{O}(nn_1 Kh(d + Tn_2)) + \mathcal{O}(nn_1|\mathcal{B}|^2 K(K-1)h) + \mathcal{O}(nn_1(n_3 + n_4)Kh)$$
$$+ \mathcal{O}(n\mathcal{C}(N_c n' \log n' + N_c n'^2)). \quad \text{(B.1)}$$

As $n_1 \gg n$, $n_1 \gg n_2$, $n_2 \doteq n_3$, $n_1 \doteq n_4$, and $K$, $T$, $|\mathcal{B}|$, $\mathcal{C}$, $N_c$, $d$, $h$, $n'$ are relatively small constants, the overall computational complexity is approximately reduced to *quadratic* with respect to the number of pre-training nodes, which demonstrates the proposed GRAVER is on par with other state-of-the-art graph multi-domain pre-training GFMs.

---

**Algorithm 2:** Fine-tuning pipeline of GRAVER.

**Input:** $m$ nodes class-balanced sampled from graph $G^{\mathcal{T}} \in \mathcal{G}^{\mathcal{T}}$ (for node classification) or $m$ graphs $\{G^{\mathcal{T}}\} \in \mathcal{G}^{\mathcal{T}}$ class-balanced sampled from target domain $\{D^{\mathcal{T}}\} \in \mathcal{D}^{\mathcal{T}}$ (for graph classification), where each of the graph $G_i^{\mathcal{T}}$ is associate with feature matrix $\mathbf{X}_i \in \mathbb{R}^{N_i \times d_i}$ and adjacency matrix $\mathbf{A} \in \mathbb{R}^{N_i \times N_i}$; Pre-trained graph encoder and discriminator $g(f_{\hat{\Theta}}^{\star}(\cdot))$; Structure token graphon $W_c^{\mathcal{A}}$ and feature token graphon $W_c^{\mathcal{X}}$ for each source domain $G^{\mathcal{S}}$ and class $c$; Learnable weight matrix $\mathbf{W}_M$ and $\mathbf{W}_C$; Weight assignment vectors $\mathbf{S}_M$ and $\mathbf{S}_C$; Learnable graph prompts $\mathcal{P}_{\Omega}$; Number of the maximum fine-tuning episodes $E_2$; Temperature parameter $\tau$; Hyperparameter $\mu$ for trading-off the MoE-CoE loss term.

**Output:** Fine-tuned graph encoder and discriminator $g(f_{\hat{\Theta}}^{\star}(\cdot))$ with $\Omega^{\star}$.

1  Initialize all parameters randomly;
2  **for** $e = 1, 2, \cdots, E_2$ **do**
3     // domain feature alignment
4     Aligned graph feature: $\widehat{\mathbf{X}}^{\mathcal{T}} \leftarrow$ Eq. (1) for the given $G^{\mathcal{T}}$ (node classification) or $\widehat{\mathbf{X}}_i^{\mathcal{T}} \leftarrow$ Eq. (1) for each $G_i^{\mathcal{T}} \in \{G^{\mathcal{T}}\}$ (graph classification);
5     // MoE-CoE network initialization and optimization
6     Initialize assigned routing weights: $\mathbf{S}_C \leftarrow$ Eq. (14), $\mathbf{S}_M \leftarrow$ Eq. (14);
7     Calculate the MoE-CoE loss: $\mathcal{L}_{\text{MoE-CoE}} \leftarrow$ Eq. (16);
8     // support samples augmentation
9     Vocabulary generation: $\mathbf{g}_c^{\text{gen}} \leftarrow$ Eq. (11) for each source dataset and class;
10    Vocabulary composition: $\widetilde{\mathbf{g}}_i^{\text{gen}} \leftarrow$ Eq. (15) for each supporting samples;
11    Vocabulary overlapping: $\widetilde{G}_i^{\mathcal{T}} \leftarrow$ Eq. (15) for each supporting samples;
12    Encode target domain node or graph with graph prompts: $\mathbf{H}_i^{\mathcal{T}} \leftarrow$ Eq. (17);
13    Calculate downstream classification loss: $\mathcal{L}_{\text{cls}}(\Omega; \mathbf{H}^{\mathcal{T}}) \leftarrow$ Eq. (18);
14    // optimization
15    Calculate overall fune-tuning loss: $\mathcal{L}_{\text{ftn}} \leftarrow$ Eq. (19);
16    Update $\Omega$ by minimizing $\mathcal{L}_{\text{ftn}}$ and back-propagation.
17 **end**

---

**Complexity Analysis.** We analyze the complexity of each part in Algorithm 2 as follows.

- **Dimension-wise feature alignment:** We apply the same shared aligner function $\mathcal{F}: \mathbb{R}^{d_i} \mapsto \mathbb{R}^d$ which is implemented by the truncated singular value decomposition (SVD) [80]. Similarly, we omit its complexity as it can be preprocessed without increasing the runtime computational burdens.

- **Semantic-wise feature alignment:** We implemented each aligner $\mathbf{W}_i \in \mathbb{R}^{d \times d}$ by a randomly initialized learnable non-linear projection matrix, where the complexity takes $\mathcal{O}(|\mathcal{V}_i|d^2)$ (node clas-

sification) and $\mathcal{O}(m|\mathcal{V}_i|d^2)$ (graph classification). Denote the average of $|\mathcal{V}_i|$ as $n_1$, the complexity for this step takes $\mathcal{O}(n_1 d^2)$ for node classification, and $\mathcal{O}(mn_1 d^2)$ for graph classification.

- **MoE-CoE weights initialization:** For $\mathbf{S}_M$, the matrix multiplication and Softmax operation takes the complexity of $\mathcal{O}(nn_1 d)$ for each support samples, while $\mathbf{S}_C$ takes the complexity of $\mathcal{O}(nn_1(d + h)\mathcal{C})$ for each support samples. Thus, the total complexity takes $\mathcal{O}(mnn_1 d(\mathcal{C} + 1) + mnn_1 h\mathcal{C})$.

- **MoE-CoE self-supervised optimization:** Calculating the uncertainty of $\mathbf{S}_M$ takes the complexity of $\mathcal{O}(n^2 n_1)$, and that of $\mathbf{S}_C$ takes $\mathcal{O}(nn_1 \mathcal{C}^2)$. Thus, the total complexity takes $\mathcal{O}(nn_1(n + \mathcal{C}^2))$.

- **Support samples augmentation:** For each source dataset and class, the vocabulary generation takes the complexity of $\mathcal{O}(mn\mathcal{C}(n'^2 + n'h))$. Then, the vocabulary composition takes the complexity of $\mathcal{O}(mnh(\mathcal{C} + 1))$. Next, the vocabulary overlapping takes the complexity of $\mathcal{O}(mn' \log n')$. Thus, the total complexity of this step takes $\mathcal{O}(mn\mathcal{C}(n'^2 + n'h) + mnh(\mathcal{C} + 1) + mn' \log n')$.

- **Graph prompt insertion:** We initialize graph prompt with learnable parameter $\boldsymbol{\Omega}$ over the aligned features, which takes the complexity of $\mathcal{O}(n_1 d)$ for node classification and $\mathcal{O}(mn_1 d)$ for graph classification.

- **Augmented support sample encoding:** Similar to the pre-training phase, we first project the aligned feature of each node to $K$ embedding space, which takes the complexity of $\mathcal{O}((n_1 + mn')Kdh)$ for node classification, and $\mathcal{O}(m(n_1 + n')Kdh)$ for graph classification. Next, the neighbor routing iterations take the complexity of $\mathcal{O}((n_1 n_2 + mn')TKh)$ for node classification, and $\mathcal{O}((nn_1 n_2 + mn')TKh)$ for graph classification, where $n_2$ denotes the average number of node neighbors. Then, the node disentangled embedding updating takes the same complexity as the last step. The total complexity takes $\mathcal{O}((n_1 + mn')Kdh + (n_1 n_2 + mn')TKh)$ for node classification, and $\mathcal{O}(m(n_1 + n')Kdh + (nn_1 n_2 + mn')TKh)$ for graph classification.

- **Fine-tuning classification loss:** Suppose there are $n_c$ classes in total, the contrastive loss takes the computational complexity of $\mathcal{O}(mn_c Kh)$ for both node and graph classification.

The overall computational complexity for the fine-tuning phase of node classification takes:

$$
\mathcal{O}(n_1 d^2) + \mathcal{O}(mnn_1 d(\mathcal{C} + 1) + mnn_1 h\mathcal{C}) + \mathcal{O}(nn_1(n + \mathcal{C}^2)) + \mathcal{O}(mn\mathcal{C}(n'^2 + n'h)
$$
$$
+ mnh(\mathcal{C} + 1) + mn' \log n') + \mathcal{O}(n_1 d) + \mathcal{O}((n_1 + mn')Kdh + (n_1 n_2 + mn')TKh)
$$
$$
+ \mathcal{O}(mn_c Kh). \quad \text{(B.2)}
$$

The overall computational complexity for the fine-tuning phase of graph classification takes:

$$
\mathcal{O}(mn_1 d^2) + \mathcal{O}(mnn_1 d(\mathcal{C} + 1) + mnn_1 h\mathcal{C}) + \mathcal{O}(nn_1(n + \mathcal{C}^2)) + \mathcal{O}(mn\mathcal{C}(n'^2 + n'h)
$$
$$
+ mnh(\mathcal{C} + 1) + mn' \log n') + \mathcal{O}(mn_1 d) + \mathcal{O}(m(n_1 + n')Kdh
$$
$$
+ (nn_1 n_2 + mn')TKh) + \mathcal{O}(mn_c Kh). \quad \text{(B.3)}
$$

As $n_1 \gg n$, $m$, $n'$, and $K$, $T$, $\mathcal{C}$, $d$, $h$, $n_c$ are relatively small constants, the overall complexity is reduced to *quasi-linear* complexity, scaling as $\mathcal{O}(n \log n)$ with the number of nodes in target graphs.

It is worth mentioning that we temporarily defer the theoretical analysis of space complexity, and instead assess the empirical computational cost through efficiency evaluation. As shown in our experiments (see Appendix D for setup details), the pre-training phase of GRAVER exhibits comparable time and memory consumption to strong baselines. Although the fine-tuning phase incurs moderately higher overhead, this can be effectively alleviated via optimized training strategies, enabling a favorable trade-off between performance and efficiency.

To summarize, the proposed GRAVER achieves an efficient training paradigm through a carefully structured pipeline. The pre-training phase, though involving multiple steps such as disentangled representation construction and graphon expert estimation, scales approximately quadratically with the number of nodes in the source graphs. In contrast, the fine-tuning phase benefits from a streamlined and modular design that leverages learned graph vocabularies and MoE-CoE routing, thereby reducing the overall complexity to quasi-linear time with respect to the target graph size. This efficiency, together with our empirical results, demonstrates that the proposed GRAVER maintains strong scalability and practicality for large-scale multi-domain graph learning tasks as a scalable GFM.

# C   Proofs

## C.1   Proof: the Upper Bound of $\mathcal{R}_{\mathrm{MI}}^u$

We first restate $\mathcal{R}_{\mathrm{MI}}^u$ for reference.

---

**Eq. (6):**

$$\mathcal{R}_{\mathrm{MI}}^u = \sum_{1 \leqslant i < j \leqslant K} I\big(\mathbf{h}_{u,i}^{\mathcal{S}}; \mathbf{h}_{u,j}^{\mathcal{S}}\big) \leqslant \sum_{1 \leqslant i < j \leqslant K} \mathbb{E}_{u \in \mathcal{B}}\Big[ -\log\Big( \mathrm{Softmax}_v\left(\langle \mathbf{h}_{u,i}^{\mathcal{S}}, \mathbf{h}_{v,j}^{\mathcal{S}}\rangle/\tau\right)\Big|_{v=u}\Big)\Big].$$

---

*Proof.* We provide a detailed derivation showing how the mutual information between two disentangled representations can be upper bounded using the InfoNCE loss.

**Step-1: From Mutual Information to InfoNCE.** The mutual information between two random variables $\mathbf{X}$ and $\mathbf{Y}$ is defined as the Kullback-Leibler ($\mathcal{D}_{\mathrm{KL}}$) divergence between their joint distribution $p(\mathbf{x}, \mathbf{y})$ and the product of marginals $p(\mathbf{x})p(\mathbf{y})$:

$$I(\mathbf{X}; \mathbf{Y}) = \mathcal{D}_{\mathrm{KL}}(p(\mathbf{x}, \mathbf{y}) \| p(\mathbf{x})p(\mathbf{y})) = \mathbb{E}_{p(\mathbf{x},\mathbf{y})}\left[\log \frac{p(\mathbf{x}, \mathbf{y})}{p(\mathbf{x})p(\mathbf{y})}\right]. \tag{C.1}$$

In practice, Eq. (C.1) is intractable to compute for continuous high-dimensional variables because it requires access to the true densities $p(\mathbf{x}, \mathbf{y})$ and $p(\mathbf{x})p(\mathbf{y})$. The InfoNCE estimator [67] provides a tractable lower bound for $I(\mathbf{X}; \mathbf{Y})$, based on a noise-contrastive learning setup that distinguishes true positive pairs from negative samples.

**Step-2: InfoNCE Objective.** Assume we sample $|\mathcal{B}|$ examples $\{(\mathbf{x}_i, \mathbf{y}_i)\}_{i=1}^{|\mathcal{B}|}$ where each $(\mathbf{x}_i, \mathbf{y}_i) \sim p(\mathbf{x}, \mathbf{y})$ is drawn from the true joint distribution. One positive pair is drawn from $p(\mathbf{x}, \mathbf{y})$, and $|\mathcal{B}| - 1$ negative pairs are sampled independently from the marginal $p(\mathbf{x})p(\mathbf{y})$. The InfoNCE objective is defined as:

$$\mathcal{L}_{\mathrm{InfoNCE}} = \mathbb{E}_{(\mathbf{x},\mathbf{y}),\{\mathbf{y}_i^-\}}\left[ -\log \frac{s(\mathbf{x}, \mathbf{y})}{s(\mathbf{x}, \mathbf{y}) + \sum_{i=1}^{|\mathcal{B}|-1} s(\mathbf{x}, \mathbf{y}_i^-)}\right], \tag{C.2}$$

where $s(\mathbf{x}, \mathbf{y})$ is a positive scoring function, typically chosen as $\exp(\langle \mathbf{x}, \mathbf{y}\rangle/\tau)$ with temperature $\tau$.

**Step-3: Mutual Information Upper Bound.** Assume that the positive sample $(\mathbf{x}, \mathbf{y}) \sim p(\mathbf{x}, \mathbf{y})$, and that the negatives $\{\mathbf{y}_i^-\} \sim p(\mathbf{y})$ are *i.i.d.*. From [67], we can rewrite the InfoNCE loss in expectation form over the true joint distribution and the sampling distribution:

$$\mathcal{L}_{\mathrm{InfoNCE}}' = \mathbb{E}_{p(\mathbf{x},\mathbf{y})}\left[ -\log \frac{s(\mathbf{x}, \mathbf{y})}{\sum_{\mathbf{y}'} s(\mathbf{x}, \mathbf{y}')}\right], \tag{C.3}$$

where the sum in the denominator includes the positive $\mathbf{y}$ and $|\mathcal{B}| - 1$ negative samples $\mathbf{y}' \sim p(\mathbf{y})$. The optimal discriminator in this noise contrastive setup yields the bound:

$$I(\mathbf{X}; \mathbf{Y}) \leqslant \log |\mathcal{B}| - \mathcal{L}_{\mathrm{InfoNCE}}'. \tag{C.4}$$

This bound becomes tight as $|\mathcal{B}| \to \infty$, then InfoNCE provides a tractable surrogate for minimizing mutual information by replacing $\mathbf{x}$ with $\mathbf{h}_{u,i}^{\mathcal{S}}$, and $\mathbf{y}$ with $\mathbf{h}_{u,j}^{\mathcal{S}}$, respectively.

**Step-4: Batch-Based InfoNCE and Independence Regularization.** In our setting, we consider the batch $\mathcal{B}$ of size $|\mathcal{B}|$, and two latent factor-specific representations $\mathbf{h}_{u,i}^{\mathcal{S}}$, $\mathbf{h}_{u,j}^{\mathcal{S}}$ over the same node $u$. The positive pair is $(\mathbf{h}_{u,i}^{\mathcal{S}}, \mathbf{h}_{u,j}^{\mathcal{S}})$, while negative samples are the cross-channel representations $\{\mathbf{h}_{v,j}^{\mathcal{S}}\}_{v \neq u}$ from other nodes in the batch. Then, the InfoNCE objective becomes:

$$\mathcal{L}_{\mathrm{InfoNCE}}^{(i,j)} = \mathbb{E}_{u \in \mathcal{B}}\left[ -\log \frac{\exp(\langle \mathbf{h}_{u,i}^{\mathcal{S}}, \mathbf{h}_{u,j}^{\mathcal{S}}\rangle/\tau)}{\sum_{v \in \mathcal{B}} \exp(\langle \mathbf{h}_{u,i}^{\mathcal{S}}, \mathbf{h}_{v,j}^{\mathcal{S}}\rangle/\tau)}\right] \tag{C.5}$$

$$\triangleq \mathbb{E}_{u \in \mathcal{B}}\Big[ -\log\Big( \mathrm{Softmax}_v\left(\langle \mathbf{h}_{u,i}^{\mathcal{S}}, \mathbf{h}_{v,j}^{\mathcal{S}}\rangle/\tau\right)\Big|_{v=u}\Big)\Big]. \tag{C.6}$$

Hence, by the inequality derived above:

$$I\big(\mathbf{h}_{u,i}^{\mathcal{S}}; \mathbf{h}_{u,j}^{\mathcal{S}}\big) \leqslant \log |\mathcal{B}| - \mathcal{L}_{\mathrm{InfoNCE}}^{(i,j)}. \tag{C.7}$$

By discarding the constant $\log |\mathcal{B}|$, we obtain a looser but practical upper bound:

$$I\big(\mathbf{h}_{u,i}^{\mathcal{S}}; \mathbf{h}_{u,j}^{\mathcal{S}}\big) \leqslant \mathbb{E}_{u \in \mathcal{B}}\Big[ -\log \Big( \mathrm{Softmax}_v \left. \left( \langle \mathbf{h}_{u,i}^{\mathcal{S}}, \mathbf{h}_{v,j}^{\mathcal{S}} \rangle / \tau \right) \right|_{v=u} \Big) \Big], \tag{C.8}$$

$$\Rightarrow \sum_{1 \leqslant i < j \leqslant K} I\big(\mathbf{h}_{u,i}^{\mathcal{S}}; \mathbf{h}_{u,j}^{\mathcal{S}}\big) \leqslant \sum_{1 \leqslant i < j \leqslant K} \mathbb{E}_{u \in \mathcal{B}}\Big[ -\log \Big( \mathrm{Softmax}_v \left. \left( \langle \mathbf{h}_{u,i}^{\mathcal{S}}, \mathbf{h}_{v,j}^{\mathcal{S}} \rangle / \tau \right) \right|_{v=u} \Big) \Big]. \tag{C.9}$$

This concludes the proof. $\qquad\square$

## C.2 Proof: Inequity [a] in Proposition 1

We first restate the inequity [a] in Proposition 1 for reference.

> **Proposition 1 (Inequity [a]):**
> $$\Delta = \big\| f\big(\mathbf{g}_u^{\mathcal{S}}\big) - f\big(\mathbf{g}_v^{\mathcal{S}}\big) \big\|_2 \leqslant \left[ \min_{\Pi \in S_K} \sum_{k=1}^{K} \Big\| \mathbf{h}_{u,k}^{\mathcal{S}} - \mathbf{h}_{v,\Pi(k)}^{\mathcal{S}} \Big\|_2^2 + \psi(\mathcal{R}_{\mathrm{MI}}^{u,v}) \right]^{\frac{1}{2}}.$$

*Proof.* We next provide a proof of the semantic discrepancies upper bound via subgraph matching and mutual information constraints.

**Step-1: Minimal Matching Distance.** Since the per-channel routing is independently conducted for each node, the semantic order of channels may not be aligned across nodes. Therefore, we consider the minimal matching distance between their vocabularies (subgraph embeddings):

$$\mathcal{D}_{\mathrm{match}} = \min_{\Pi \in S_k} \sum_{k=1}^{K} \big\| \mathbf{h}_{u,k}^{\mathcal{S}} - \mathbf{h}_{v,\Pi(k)}^{\mathcal{S}} \big\|_2^2, \tag{C.10}$$

where $\Pi \in S_k$ denotes a permutation over channels.

**Step-2: Information-theoretic Slack.** To encourage disentanglement, we introduce an information-theoretic regularization $\mathcal{R}_{\mathrm{MI}}$ during training to minimize inter-channel mutual information. Ideally, a small $\mathcal{R}_{\mathrm{MI}}$ implies that each channel captures a distinct and independent semantic factor. However, due to optimization limitations, residual dependencies across channels may still exist. These dependencies introduce cross-channel interaction terms in the final embedding space, making the overall semantic discrepancies larger than the matching term alone. To formally capture this effect, we introduce a slack function $\psi(\mathcal{R}_{\mathrm{MI}})$, upper bounding the contribution from such interactions.

To obtain a tractable analytical form, we adopt a linear upper bound of the form:

$$\psi(\mathcal{R}_{\mathrm{MI}}) \leqslant \beta \cdot \mathcal{R}_{\mathrm{MI}}, \tag{C.11}$$

where $\beta > 0$ is a model-dependent constant that reflects how sensitive the final semantic discrepancies are to cross-channel redundancy. This linear approximation is justified under mild distributional assumptions. Specifically, when the per-channel embeddings approximately follow Gaussian or smooth distributions, the single mutual information term between two channels can be bounded in terms of their correlation:

$$I\big(\mathbf{h}_{u,i}^{\mathcal{S}}; \mathbf{h}_{u,j}^{\mathcal{S}}\big) \geqslant \frac{1}{2} \log \frac{1}{1 - \rho^2}, \tag{C.12}$$

where $\rho$ is the Pearson correlation coefficient. The equality can be satisfied only if $\big(\mathbf{h}_{u,i}^{\mathcal{S}}; \mathbf{h}_{u,j}^{\mathcal{S}}\big) \sim \mathcal{N}$ is jointly Gaussian with variances $\sigma_i^2, \sigma_j^2$ and covariance $\rho\sigma_i\sigma_j$. This inequality is commonly used in the literature as a conservative estimate of mutual information [93, 79], since direct computation of $I(\mathbf{h}_{u,i}^{\mathcal{S}}; \mathbf{h}_{u,j}^{\mathcal{S}})$ is intractable for arbitrary distributions. We omit its proof for simplicity.

Conversely, high mutual information implies strong coupling (large covariance) between channels, which directly contributes to the cross terms in the semantic discrepancies calculation. Thus, using a linear upper bound allows us to quantify this slack without explicitly modeling the full joint statistics.

**Step-3: Upper Bound on Semantic Discrepancy.** Combining the matching difference with the information-theoretic slack, we obtain the following upper bound:

$$\Delta = \left\| f\!\left(\mathbf{g}_u^{\mathcal{S}}\right) - f\!\left(\mathbf{g}_v^{\mathcal{S}}\right) \right\|_2 \leqslant \left[ \mathcal{D}_{\text{match}} + \psi(\mathcal{R}_{\text{MI}}^{u,v}) \right]^{\frac{1}{2}} \tag{C.13}$$

$$= \left[ \min_{\Pi \in S_K} \sum_{k=1}^{K} \left\| \mathbf{h}_{u,k}^{\mathcal{S}} - \mathbf{h}_{v,\Pi(k)}^{\mathcal{S}} \right\|_2^2 + \psi(\mathcal{R}_{\text{MI}}^{u,v}) \right]^{\frac{1}{2}}. \tag{C.14}$$

This concludes the proof. □

## C.3    Proof: Inequity [b] in Proposition 1

We first restate the inequity [b] in Proposition 1 for reference.

---

**Proposition 1 (Inequity [b]):**

$$\left[ \min_{\Pi \in S_K} \sum_{k=1}^{K} \left\| \mathbf{h}_{u,k}^{\mathcal{S}} - \mathbf{h}_{v,\Pi(k)}^{\mathcal{S}} \right\|_2^2 + \psi(\mathcal{R}_{\text{MI}}^{u,v}) \right]^{\frac{1}{2}} \leqslant \epsilon \sqrt{K} \left( \frac{C_\sigma L_{\mathbf{W}} L_s}{4\rho\tau} \right)^{T}.$$

---

*Proof.* We achieve the proof first by further relaxing the conditions in Proof C.2.

**Step-1: Further Relaxation of the inequity [a].** From Eq. (C.10), we can define the optimal channel matching can be derived with the optimal permutation function as:

$$\Pi^\star = \arg \min_{\Pi \in S_k} \sum_{k=1}^{K} \left\| \mathbf{h}_{u,k}^{\mathcal{S}} - \mathbf{h}_{v,\Pi(k)}^{\mathcal{S}} \right\|_2^2. \tag{C.15}$$

Then, we can reach a more relaxed upper bound:

$$\sum_{k=1}^{K} \left\| \mathbf{h}_{u,k}^{\mathcal{S}} - \mathbf{h}_{v,\Pi^\star(k)}^{\mathcal{S}} \right\|_2^2 \leqslant \sum_{k=1}^{K} \left\| \mathbf{h}_{u,k}^{\mathcal{S}} - \mathbf{h}_{v,k}^{\mathcal{S}} \right\|_2^2. \tag{C.16}$$

In this way, we can rewrite the inequality [a] in Proposition C.2 as:

$$\left[ \min_{\Pi \in S_K} \sum_{k=1}^{K} \left\| \mathbf{h}_{u,k}^{\mathcal{S}} - \mathbf{h}_{v,\Pi(k)}^{\mathcal{S}} \right\|_2^2 + \psi(\mathcal{R}_{\text{MI}}^{u,v}) \right]^{\frac{1}{2}} \leqslant \left[ \sum_{k=1}^{K} \left\| \mathbf{h}_{u,k}^{\mathcal{S}} - \mathbf{h}_{v,k}^{\mathcal{S}} \right\|_2^2 + \psi(\mathcal{R}_{\text{MI}}^{u,v}) \right]^{\frac{1}{2}}. \tag{C.17}$$

Next, we continue proof with respect to the RHS of Eq. (C.17).

**Step-2: Semantic Discrepancy of Single Channel.** Assuming $\|\widehat{\mathbf{x}}_u^{\mathcal{S}} - \widehat{\mathbf{x}}_v^{\mathcal{S}}\|_2 \leqslant \epsilon$. The activation function $\sigma$, weight matrix $\mathbf{W}$, and cosine similarity $\langle \cdot, \cdot \rangle$ (inner product) are Lipschitz continuous with constants $C_\sigma$, $L_{\mathbf{W}}$, and $L_s$, respectively. $\rho$ is the lower bound of $L_2$-norm in Eq. (3). Then, for each single channel, the semantic discrepancies in projected features satisfy:

$$\left\| \mathbf{h}_{u,k}^{\mathcal{S}} - \mathbf{h}_{v,k}^{\mathcal{S}} \right\|_2 \leqslant \frac{C_\sigma L_{\mathbf{W}}}{\rho} \left\| \widehat{\mathbf{x}}_u^{\mathcal{S}} - \widehat{\mathbf{x}}_v^{\mathcal{S}} \right\|_2 \leqslant \frac{C_\sigma L_{\mathbf{W}}}{\rho} \cdot \epsilon, \tag{C.18}$$

where the normalization lower bound ensures numerical stability by preventing the denominator from approaching 0. Next, we discuss the discrepancy in the channel routing probability.

The routing attention $\alpha_{v \to k}$ is determined by the cosine similarity $\langle \mathbf{h}_{u,k}^{\mathcal{S}}, \mathbf{h}_{v,k}^{\mathcal{S}} \rangle$:

$$\alpha_{v \to k} \propto \frac{\exp\left( \langle \mathbf{h}_{u,k}^{\mathcal{S}}, \mathbf{h}_{v,k}^{\mathcal{S}} \rangle / \tau \right)}{\sum_{k'} \exp\left( \langle \mathbf{h}_{u,k}^{\mathcal{S}}, \mathbf{h}_{v,k'}^{\mathcal{S}} \rangle / \tau \right)}. \quad \text{(Eq. (2))} \tag{C.19}$$

Differentiating with respect to $\langle \mathbf{h}_{u,k}^{\mathcal{S}}, \mathbf{h}_{v,k}^{\mathcal{S}} \rangle$, we obtain the sensitivity of routing attention:

$$\frac{\partial \alpha_{v \to k}}{\partial \langle \mathbf{h}_{u,k}^{\mathcal{S}}, \mathbf{h}_{v,k}^{\mathcal{S}} \rangle} = \frac{\alpha_{v \to k}(1 - \alpha_{v \to k})}{\tau}. \tag{C.20}$$

Then, the discrepancy in routing attention is bounded by:

$$|\alpha_{v \to k} - \alpha_{u \to k}| \leqslant \frac{L_s}{4\tau} \left\| \mathbf{h}_{u,k}^{\mathcal{S}} - \mathbf{h}_{v,k}^{\mathcal{S}} \right\|_2 . \quad \text{(since } \alpha_{v \to k}(1 - \alpha_{v \to k}) \leqslant \frac{1}{4}) \qquad \text{(C.21)}$$

Thus, the semantic discrepancy after routing is:

$$\left\| \mathbf{h}_{u,k}^{\mathcal{S}(T)} - \mathbf{h}_{v,k}^{\mathcal{S}(T)} \right\|_2 \leqslant \epsilon \left( \frac{C_\sigma L_{\mathbf{W}} L_s}{4\rho\tau} \right)^T . \qquad \text{(C.22)}$$

**Step-3: Semantic Discrepancy for $K$ Channels.** Integrating Eq. (C.22) into RHS of Eq. (C.17):

$$\left[ \sum_{k=1}^{K} \left\| \mathbf{h}_{u,k}^{\mathcal{S}} - \mathbf{h}_{v,k}^{\mathcal{S}} \right\|_2^2 + \psi(\mathcal{R}_{\mathrm{MI}}^{u,v}) \right]^{\frac{1}{2}} \leqslant \left[ \sum_{k}^{K} \epsilon \left( \frac{C_\sigma L_{\mathbf{W}} L_s}{4\rho\tau} \right)^T + \psi(\mathcal{R}_{\mathrm{MI}}^{u,v}) \right]^{\frac{1}{2}} \qquad \text{(C.23)}$$

$$= \epsilon\sqrt{K} \left( \frac{C_\sigma L_{\mathbf{W}} L_s}{4\rho\tau} \right)^T + \sqrt{\psi(\mathcal{R}_{\mathrm{MI}}^{u,v})} . \qquad \text{(C.24)}$$

Considering $\rho \to 0$ in Eq. (C.12), which implies the variables are linearly uncorrelated with no shared information. Then, the mutual information regularization term $\mathcal{R}_{\mathrm{MI}}^{u,v} \to 0$. We reach:

$$\left[ \sum_{k=1}^{K} \left\| \mathbf{h}_{u,k}^{\mathcal{S}} - \mathbf{h}_{v,k}^{\mathcal{S}} \right\|_2^2 + \psi(\mathcal{R}_{\mathrm{MI}}^{u,v}) \right]^{\frac{1}{2}} \leqslant \epsilon\sqrt{K} \left( \frac{C_\sigma L_{\mathbf{W}} L_s}{4\rho\tau} \right)^T . \qquad \text{(C.25)}$$

By combining Eq. (C.17) and Eq. (C.25), we conclude the proof. $\qquad \square$

## C.4   Proof: Generation Distributional Convergence (Proposition 2)

We first introduce a lemma on the graph estimator consistency.

> **Lemma 1** (**Consistency of Vocabulary Graphon Estimators**). *Assume that the graph vocabulary samples are drawn i.i.d. from a latent generative model with the true underlying structure graphon function $W_c^{\mathcal{A}\star} \colon [0,1]^2 \mapsto [0,1]$ and the feature graphon function $W_c^{\mathcal{X}\star} \colon [0,1] \mapsto \mathbb{R}^d$, both of which are bounded and Lipschitz continuous. Further, assume that the alignment mappings $\pi_i$ are consistent with the true latent positions up to vanishing approximation error. Then, as $N_c \to \infty$, the estimators satisfy uniform convergence:*
>
> $$\left\| W_c^{\mathcal{A}} - W_c^{\mathcal{A}\star} \right\|_\infty \to 0, \quad \left\| W_c^{\mathcal{X}} - W_c^{\mathcal{X}\star} \right\|_\infty \to 0. \qquad \text{(C.26)}$$

*Proof.* Let $u, v \in [0, 1]$ be two arbitrary latent positions. We consider the estimation error:

$$\left| W_c^{\mathcal{A}}(u,v) - W_c^{\mathcal{A}\star}(u,v) \right| = \left| \frac{1}{N_c} \sum_{i=1}^{N_c} \left( \mathcal{A}_i^{(c)}[\pi_i(u), \pi_i(v)] - W_c^{\mathcal{A}\star}(u,v) \right) \right| . \qquad \text{(C.27)}$$

We decompose this difference into bias and variance terms. The random variable $\mathcal{A}_i^{(c)}[\pi_i(u), \pi_i(v)]$ is a Bernoulli variable with success probability $W_c^{\mathcal{A}\star}(u,v) + \nu_i(u,v)$, where $\nu_i(u,v)$ captures the alignment approximation error and latent variability. Under the alignment consistency assumption, we have $|\nu_i(u,v)| \leqslant \nu_{n'}$ for all $i$, with $\nu_{n'} \to 0$ as the number of nodes in a vocabulary $n' \to \infty$.

Applying Hoeffding's inequality [25], we bound the deviation for fixed $(u,v)$:

$$p\left( \left| W_c^{\mathcal{A}}(u,v) - \mathbb{E}\left[ W_c^{\mathcal{A}}(u,v) \right] \right| > \omega \right) \leqslant 2 \exp\left( -2N_c \omega^2 \right). \qquad \text{(C.28)}$$

Since the domain $[0,1]^2$ is compact, and $W_c^{\mathcal{A}\star}$ is Lipschitz, we cover the domain with a finite $\omega$-net of size $\mathcal{O}(1/\omega^2)$, and apply the union bound. Then, for any $\eta > 0$, with probability at least $1 - \eta$:

$$\sup_{u,v} \left| W_c^{\mathcal{A}}(u,v) - W_c^{\mathcal{A}\star}(u,v) \right| \leqslant \nu_{n'} + \mathcal{O}\left( \sqrt{\frac{\log(1/\eta)}{N_c}} \right), \qquad \text{(C.29)}$$

where $\nu_{n'}$ accounts for alignment error. Therefore, as $N_c \to \infty$, we conclude that:

$$\left\| W_c^{\mathcal{A}} - W_c^{\mathcal{A}\star} \right\|_\infty \to 0. \tag{C.30}$$

The same argument applies to $W_c^{\mathcal{X}}$. Since the feature graphon is vector-valued and assumed Lipschitz, and the feature vectors are bounded, we apply Azuma-Hoeffding inequality [74] coordinate-wise to each dimension of $\mathbb{R}^d$, and take a union bound. Thus:

$$\left\| W_c^{\mathcal{X}} - W_c^{\mathcal{X}\star} \right\|_\infty \to 0, \tag{C.31}$$

as $N_c \to \infty$. This completes the proof. $\qquad\square$

Based on Lemma 1, we then prove Proposition 2. We first restate it for reference.

---

**Proposition 2** (**Generation Distributional Convergence**) *Let $\mathbf{g}_c^{\text{emp}}$ be the empirical distribution over $n'$-node vocabulary subgraphs of class $c$, collected from the disentangled vocabulary bank:*

$$\mathbf{g}_c^{\text{emp}} := \frac{1}{N_c} \sum_{i=1}^{N_c} \delta_{(\mathcal{A}_i^{(c)}, \mathcal{X}_i^{(c)})}, \quad \text{each } (\mathcal{A}_i^{(c)}, \mathcal{X}_i^{(c)}) \in \{0,1\}^{n' \times n'} \times \mathbb{R}^{n' \times d},$$

*where $\delta_{(\cdot)}$ denotes Dirac measure, representing a point mass probability distribution. Assume the true underlying structure and feature functions $W_c^{\mathcal{A}\star}$, $W_c^{\mathcal{X}\star}$ are bounded, Lipschitz, and satisfy permutation equivariance. If the estimators $W_c^{\mathcal{A}}$, $W_c^{\mathcal{X}}$ converge uniformly in $L_\infty$ norm, then the total variation (TV) distance between $\mathbf{g}_c^{\text{gen}} = (\widetilde{\mathcal{A}}_c, \widetilde{\mathcal{X}}_c)$ and $\mathbf{g}_c^{\text{emp}}$ vanishes as $N_c \to \infty$:*

$$\| \mathbf{g}_c^{\text{gen}} - \mathbf{g}_c^{\text{emp}} \|_{\text{TV}} \to 0.$$

---

*Proof.* Let $\mathbf{u}_1, \cdots, \mathbf{u}_{n'}$ be *i.i.d.* latent positions drawn from $\mathcal{U}[0,1]$. Denote latent vector as $\mathbf{u} = (\mathbf{u}_1, \cdots, \mathbf{u}_{n'}) \in [0,1]^{n'}$. We define the vocabulary generation process via the mapping: $\Phi \colon \mathbf{u} \mapsto (\widetilde{\mathcal{A}}, \widetilde{\mathcal{X}})$. The convergence assumption on $W_c^{\mathcal{A}}$ and $W_c^{\mathcal{X}}$ are formally established in Lemma 1.

**Step-1: Total Variation Distance Decomposition.** To study the convergence of $\mathbf{g}_c^{\text{gen}}$ to the empirical distribution $\mathbf{g}_c^{\text{emp}}$, we introduce the limiting (true) distribution $\mathbf{g}_c^\star$ induced by the true graphons $W_c^{\mathcal{A}\star}$, $W_c^{\mathcal{X}\star}$. We then decompose the total variation distance based on the triangle inequality as:

$$\| \mathbf{g}_c^{\text{gen}} - \mathbf{g}_c^{\text{emp}} \|_{\text{TV}} \leqslant \| \mathbf{g}_c^{\text{gen}} - \mathbf{g}_c^\star \|_{\text{TV}} + \| \mathbf{g}_c^\star - \mathbf{g}_c^{\text{emp}} \|_{\text{TV}}. \tag{C.32}$$

We prove that both terms vanish as $N_c \to \infty$, using functional convergence of the graphon estimators.

**Step-2: Convergence of $\mathbf{g}_c^{\text{gen}} \to \mathbf{g}_c^\star$.** We consider the pushforward measure of latent positions $\mathbf{u}$ through the mapping $\Phi^{(W)}$ that uses estimated graphons $W_c^{\mathcal{A}}$, $W_c^{\mathcal{X}}$, versus the mapping $\Phi^{(\star)}$ that uses the true graphons $W_c^{\mathcal{A}\star}$, $W_c^{\mathcal{X}\star}$. Since $\widetilde{\mathcal{A}}[i,j]$ are conditionally independent Bernoulli trials, and the Bernoulli distribution is Lipschitz with respect to its parameter, we have for every fixed $\mathbf{u}$:

$$\mathcal{D}_{\text{TV}}\left( \varsigma_{n'}^{\mathcal{A}}(\cdot \mid \mathbf{u}), \varsigma_{n'}^{\mathcal{A}\star}(\cdot \mid \mathbf{u}) \right) \leqslant \sum_{i<j} \left| W_c^{\mathcal{A}}(\mathbf{u}_i, \mathbf{u}_j) - W_c^{\mathcal{A}\star}(\mathbf{u}_i, \mathbf{u}_j) \right|, \tag{C.33}$$

where $\varsigma_{n'}^{\mathcal{A}}$ be the induced measure over adjacency matrices. By uniform convergence of $W_c^{\mathcal{A}} \to W_c^{\mathcal{A}\star}$, this sum is bounded by $\mathcal{O}(n'^2 \cdot \Delta_{n'}^{\mathcal{A}})$ where $\Delta_{n'}^{\mathcal{A}} = \| W_c^{\mathcal{A}} - W_c^{\mathcal{A}\star} \|_\infty \to 0$. Similarly, for features:

$$\left\| \widetilde{\mathcal{X}}_i - \mathcal{X}_i^\star \right\|_2 = \left\| W_c^{\mathcal{X}}(\mathbf{u}_i) - W_c^{\mathcal{X}^*}(\mathbf{u}_i) \right\|_2 \leqslant \left\| W_c^{\mathcal{X}} - W_c^{\mathcal{X}\star} \right\|_\infty. \tag{C.34}$$

Hence, via coupling over $\mathbf{u}$, we construct a measure-preserving map such that:

$$\| \mathbf{g}_c^{\text{gen}} - \mathbf{g}_c^\star \|_{\text{TV}} \leqslant C \left( n'^2 \Delta_{n'}^{\mathcal{A}} + n' \Delta_{n'}^{\mathcal{X}} \right) \to 0, \tag{C.35}$$

where $\Delta_{n'}^{\mathcal{A}} = \| W_c^{\mathcal{A}} - W_c^{\mathcal{A}\star} \|_\infty$, $\Delta_{n'}^{\mathcal{X}} = \| W_c^{\mathcal{X}} - W_c^{\mathcal{X}\star} \|_\infty$, and $C > 0$ is a constant.

**Step-3: Convergence of $\mathbf{g}_c^{\text{emp}} \to \mathbf{g}_c^\star$.** By assumption, the vocabulary bank $\{ (\mathcal{A}_i^{(c)}, \mathcal{X}_i^{(c)}) \}$ are *i.i.d.* samples from $\mathbf{g}_c^\star$. Since the sample space $\mathcal{Z}_{n'} \in \{0,1\}^{n' \times n'} \times \mathbb{R}^{n' \times d}$ is a Polish space (product of compact discrete and Euclidean space), the Glivenko-Cantelli theorem [90] applies. Therefore:

$$\| \mathbf{g}_c^{\text{emp}} - \mathbf{g}_c^\star \|_{\text{TV}} \to 0. \tag{C.36}$$

This completes the proof. $\qquad\square$

# D Experiment Details

In this section, we present experimental details. The dataset statistics are summarized in Table D.1.

Table D.1: Statistics of the multi-domain graph dataset.

| Dataset | Domain | # Nodes | # Edges | # Feature Dimensions | # Classes | Avg. # Deg. |
|---------|--------|---------|---------|----------------------|-----------|-------------|
| Cora [65] | Citation | 2,708 | 5,429 | 1,433 | 7 | 4.00 |
| CiteSeer [16] | Citation | 3,186 | 4,277 | 3,703 | 6 | 2.57 |
| PubMed [73] | Citation | 19,717 | 44,338 | 500 | 3 | 4.50 |
| ogbn-arXiv [28] | Citation | 169,343 | 1,166,243 | 128 | 40 | 13.77 |
| ogbn-Tech [28] | Co-purchase | 47,428 | 2,077,241 | 100 | 3 | 87.60 |
| ogbn-Home [28] | Co-purchase | 9,790 | 131,841 | 100 | 5 | 26.93 |
| Wiki-CS [66] | Web Link | 11,701 | 216,123 | 300 | 10 | 36.94 |

## D.1 Datasets Details

To emphasize multi-domain pre-training, we select **seven** benchmark graph datasets across **three** distinct domains, in contrast to conventional settings where each dataset is treated as an independent domain. These datasets vary in structure, scale, and feature distribution, providing a diverse and challenging testbed for evaluating pre-trained graph models.

**Citation Domain.** This domain comprises **four** widely used citation network benchmarks. Each dataset consists of a single graph, where nodes represent academic papers, edges denote citation relationships, and node labels correspond to the category or research topic of each paper.

- Cora [65]: [CC BY 4.0] This dataset contains 2,708 machine learning-related papers from 7 research fields. Each paper is represented by a 1,433-dimensional bag-of-words feature vector, where each dimension indicates a specific term. The graph includes 5,429 citations, resulting in a moderately sparse structure. Due to its small size and clear class boundaries, Cora is widely used for semi-supervised node classification benchmarks.

- CiteSeer [16]: [CC BY-NC-SA 3.0] This dataset comprises 3,186 publications categorized into 6 fields. CiteSeer represents each node with a 3,703-dimensional sparse feature vector capturing word occurrences. The graph contains 4,277 citations and exhibits a sparser structure with fewer average node degrees compared to Cora. The dataset is considered more challenging due to its noisy labels and lower graph connectivity.

- PubMed [73]: [License Unspecified] This biomedical citation network consists of 19,717 research articles on diabetes, divided into 3 classes. Each node is described by a dense 500-dimensional feature vector based on TF-IDF statistics of medical terms. With 44,338 citations, PubMed features a significantly denser graph structure. Its larger scale and domain specificity make it a common benchmark for evaluating scalable graph learning methods.

- ogbn-arXiv [28]: [ODC-BY] A large-scale citation network comprising 169,343 computer science papers from arXiv. Each node is associated with a 128-dimensional feature vector, generated by averaging word2vec embeddings of the paper's title and abstract. The graph contains 1,166,243 citations, and node labels span 40 subject areas. ogbn-arXiv is widely used for benchmarking scalable graph learning methods on real-world data.

**Co-purchase Domain.** This domain includes **two** networks derived from the ogbn-Product [28], capturing relationships between items frequently co-purchased on Amazon. These datasets are widely used for evaluating recommendation and product classification tasks. Nodes represent products, edges denote co-purchase relationships, and node labels correspond to product categories.

- ogbn-Tech [28]: [Amazon License] A co-purchase sub-network of ogbn-Product [28] related to technology with 47,428 nodes, 2,077,241 edges, and 3 categories. Each node represents a product, edges indicate co-purchase relationships, and node labels correspond to product categories. ogbn-Tech is commonly used for testing recommendation algorithms in the tech product domain.

- `ogbn-Home` [28]: [Amazon License] A co-purchase sub-network of `ogbn-Product` [28] related to home usage consisting of 9,790 nodes, 131,841 edges, and 6 categories. Each node represents a product, edges denote co-purchase relationships, and node labels correspond to product categories. `ogbn-Home` is useful for evaluating recommendation systems in the home goods domain.

**Web Link Domain.** This domain includes `Wiki-CS` [66][MIT License], a comprehensive collection of Wikipedia entries with 11,701 nodes and 216,123 web links, which is categorized into 10 distinct areas of computer science. The graph contains nodes representing articles, with edges denoting citation relationships between them. Each node is associated with a feature vector generated from the article's textual content. `Wiki-CS` is widely used for evaluating graph learning methods, particularly for tasks that involve knowledge construction and domain-specific knowledge extraction.

## D.2  Baseline Details

We benchmark GRAVER against **15** state-of-the-art methods spanning **four** major categories: vanilla graph neural networks, self-supervised graph pre-training, prompt-based graph fine-tuning, and multi-domain pre-training frameworks. This comprehensive comparison highlights the effectiveness and robustness of our approach across diverse paradigms.

**Vanilla Graph Neural Networks.** This category comprises standard graph neural networks trained from scratch on downstream tasks without any pre-training. These models serve as fundamental baselines for evaluating representation learning performance.

- `GCN` [40] (backbone): A spectral-based method that captures local graph structures via layer-wise neighborhood aggregation. We adopt `GCN` as a backbone for both node and graph classification.
- `GAT` [91]: A graph neural network that leverages attention mechanisms to assign adaptive weights to neighbors, enabling selective aggregation of relevant information. We apply `GAT` to both node and graph classification tasks.

**Self-Supervised Graph Pre-training.** This category includes baselines that learn transferable representations by optimizing auxiliary tasks without annotations. By capturing intrinsic structural and semantic patterns during pre-training, these models significantly enhance downstream performance.

- `GCC` [69]: It learns transferable structural representations through subgraph instance discrimination across multiple graphs. By leveraging contrastive learning, `GCC` captures domain-invariant topological patterns, enhancing generalization to unseen graphs and improving performance in out-of-domain scenarios. It is applied to node classifications.
- `DGI` [92]: It maximizes mutual information between local patch and global summary, encouraging node representations to retain graph-wide context. Unlike relying on random walks, it is well-suited for both transductive and inductive settings. It is applied to node classifications.
- `InfoGraph` [81]: It focuses on learning graph-level representations by maximizing mutual information between entire graphs and hierarchical substructures. This design enables effective encoding of both structural and semantic features. It is applied to graph classifications.
- `GraphCL` [116]: It introduces contrastive learning with a suite of graph augmentations, such as node dropping, edge perturbation, and subgraph sampling, *etc*. These transformations promote the learning of robust and transferable representations under various supervision regimes. It is applied to both node and graph classifications.
- `DSSL` [102]: It addresses the limitations of homophily assumptions by introducing a latent variable model that decouples semantic content from graph structure. This enables the capture of diverse neighborhood patterns, leading to improved adaptability across homophilic and heterophilic graphs. It is applied to both node and graph classifications.
- `GraphACL` [103]: It proposes an asymmetric contrastive learning method tailored for heterophilic graphs. By removing the reliance on handcrafted augmentations and homophily priors, it effectively models both local interactions and monophily-based similarities. It is applied to both node and graph classifications.

**Prompt-based Graph Fine-tuning.** This category includes baselines that inject learnable prompts into pre-trained graph models to steer task-specific adaptation. By introducing lightweight, tunable instructions, prompt-based fine-tuning enhances the model's flexibility and improves knowledge transfer to diverse downstream tasks.

- `GPPT` [82]: It bridges pre-training and downstream adaptation by introducing token-pair prompts that recast node classification into edge prediction. It employs masked edge prediction as a pre-training task and maintains this objective format during fine-tuning, thereby reducing the task discrepancy and enabling effective prompt-guided transfer. It is applied to node classifications.
- `GraphPrompt` [58]: It formulates both pre-training and downstream tasks within a unified prompting framework, driven by learnable instructions. This design narrows the objective gap while enhancing label efficiency and transferability. The method provides a general-purpose interface for applying prompts to various graph tasks. It is applied to both node and graph classifications.
- `GPF` [14]: It presents a universal prompt-based tuning approach compatible with arbitrary pre-trained GNNs. By injecting prompts directly into input feature space, it enables flexible downstream adaptation without architectural constraints. `GPF` consistently outperforms full-model fine-tuning and tailored prompt strategies. It is applied to both node and graph classifications.
- `ProNoG` [121]: It targets the challenge of non-homophilic graphs by introducing a conditional network that dynamically adjusts prompt representations based on node-level structural cues. This allows the model to capture both homophilic and heterophilic patterns, achieving strong generalization across real-world graph scenarios. It is applied to both node and graph classifications.

**Multi-Domain Graph Pre-training.** This category includes the most directly related methods to ours, aiming to learn graph representations that generalize across heterogeneous domains. By exploiting shared structural and semantic patterns, these methods enhance the transferability and robustness of pre-trained models in multi-domain settings.

- `GCOPE` [136]: It proposes a unified graph pre-training framework that aggregates multiple datasets to distill common structural patterns across domains. By explicitly mitigating negative transfer, which is a common challenge in cross-domain scenarios, it facilitates more effective few-shot adaptation. This method advances the development of general-purpose graph foundation models tailored for multi-domain settings. It is applied to both node and graph classifications.
- `MDGPT` [123]: It introduces a text-independent multi-domain pre-training framework that aligns heterogeneous graph domains through learnable domain tokens. It further incorporates a dual-prompt strategy, which comprises unifying and mixing prompts to adaptively reconcile domain gaps during fine-tuning. This design enables effective transfer by minimizing semantic conflicts and enhancing the utility of source knowledge. It is applied to both node and graph classifications.
- `SAMGPT` [119]: It tackles cross-domain adaptation in text-free graphs by aligning structural patterns across domains, in contrast to `MDGPT`, which focuses on feature alignment using domain tokens. It introduces structure tokens to unify graph aggregation during pre-training, and employs dual prompts to transfer both generalizable and domain-specific structural knowledge. It is applied to both node and graph classifications.

### D.3 Experimental Setting Details

### D.3.1 Detailed Settings for Section 5.2 (RQ1)

This section examines the effectiveness and robustness of GRAVER in transferring knowledge from multiple source domains to unseen target datasets under the few-shot learning settings, *i.e.*, ($C$-way) $m$-shot scenarios (where $1 \leqslant m \leqslant 10$). GRAVER is first pre-trained on source-domain graphs and then fine-tuned with only a limited number of labeled samples from the target domain. To mitigate the instability of fine-tuning and build a trustworthy GFM, we propose a generative augmentation strategy that enriches support samples based on graph vocabularies. This approach ensures both high quality and stability of GFM fine-tuning under any random sample selection.

$m$-**shot Fine-tuning Settings.** In each experiment, we randomly sample $m$ labeled instances (nodes or graphs) per class to form the support set used for fine-tuning, while the remaining labeled samples serve as the query set for evaluation. To evaluate robustness and mitigate sampling bias, we perform 20 independent runs with different random seeds, reporting their mean accuracy and standard deviation. We evaluate GRAVER on both node and graph classification tasks. For graph classification, we construct graph-level datasets by extracting 2-hop ego-graphs centered on each node. It is worth noting that `GCC`, `DGI`, and `GPPT` are designed exclusively for node-level classification and are thus only included in node-level comparisons, whereas `InfoGraph`, an extension of `DGI`, is evaluated solely on graph-level classification.

Table D.2: Statistics of the multi-domain graph dataset.

| Model | Pre-training | | | | Fine-tuning and Downstream Task | | | | | |
|---|---|---|---|---|---|---|---|---|---|---|
| | multi-domain | transferable pattern modeling | LLM-enhanced feature | self-supervised pre-training | cross-domain | multi-task | zero-shot learning | few-shot learning | in-context learning | data augmentation |
| GCN (bb.) [40] | – | – | – | – | – | – | – | ✓ | – | – |
| GAT [91] | – | – | – | – | – | – | – | ✓ | – | – |
| GCC [69] | – | – | – | ✓ | ✓ | ✓ | – | – | – | – |
| DGI [92] | – | – | – | ✓ | ✓ | – | – | – | – | – |
| InfoGraph [81] | – | – | – | ✓ | – | – | – | – | – | – |
| GraphCL [116] | – | – | – | ✓ | – | ✓ | – | – | – | – |
| DSSL [102] | – | – | – | ✓ | – | ✓ | – | – | – | – |
| GraphACL [103] | – | – | – | ✓ | – | ✓ | – | – | – | – |
| GPPT [82] | – | – | – | ✓ | – | ✓ | – | ✓ | ✓ | – |
| GraphPrompt [58] | – | – | – | ✓ | – | ✓ | – | ✓ | ✓ | – |
| GPF [14] | – | – | – | ✓ | – | ✓ | – | ✓ | ✓ | – |
| ProNoG [121] | – | – | – | ✓ | ✓ | ✓ | – | ✓ | ✓ | – |
| GCOPE [136] | ✓ | – | – | ✓ | ✓ | – | – | ✓ | ✓ | – |
| MDGPT [123] | ✓ | ✓ | – | ✓ | ✓ | ✓ | – | ✓ | ✓ | – |
| SAMGPT [119] | ✓ | ✓ | – | ✓ | ✓ | ✓ | – | ✓ | ✓ | – |
| **GRAVER** (ours) | ✓ | ✓ | ✓ | ✓ | ✓ | ✓ | ✓ | ✓ | ✓ | ✓ |

**Robustness against Random Noise Settings.** To further evaluate the robustness of GRAVER under perturbations during fine-tuning, we introduce random noise into the support sets from two complementary aspects: *feature noise* and *structure noise*. For *feature noise*, we add random Gaussian perturbations to each dimension of node features for all support samples, following the formulation $\lambda_f \cdot r \cdot \epsilon_f$, where $r$ denotes the reference amplitude of the original features, $\epsilon_f \sim \mathcal{N}(0, 1)$ is standard Gaussian noise, and $\lambda_f \in \{0.2, 0.4, 0.6, 0.8\}$ controls the noise intensity. For *structure noise*, we randomly modify the support graph structure by simultaneously deleting and adding $\lambda_s$ (%) of the edges, with the same range of $\lambda_f$ values. These controlled perturbations allow us to systematically assess the stability and noise tolerance of GRAVER in few-shot adaptation scenarios.

**Cross-Dataset and Cross-Domain Settings.** We consider two types of domain adaptation scenarios. For *cross-dataset* transfer, the target dataset is unseen during pre-training but comes from the same domain as the source datasets. For *cross-domain* transfer, the target dataset originates from a different domain altogether. It is worth noting that, while some prior works on cross-domain graph pre-training and fine-tuning [136, 123, 119] treat each dataset as a separate domain, we argue that such treatment is overly coarse. Instead, we define domains based on broader semantic or application categories. As a result, our *cross-dataset* setting corresponds to what previous literature often calls *cross-domain*, whereas our *cross-domain* setting reflects a higher-level shift across distinct fields.

### D.3.2 Detailed Settings for Section 5.3 (RQ2)

We investigate contributions of GRAVER's three key components

**GRAVER (*w/o SIP*).** This variant removes the semantic independence promotion (*SIP*) module, which regularizes mutual information minimization between disentangled vocabulary channels. Specifically, we eliminate the mutual information regularizer $\mathcal{R}_{MI}$ in Eq. 8 (set $\lambda = 0$), thus removing the constraint that encourages each factor-specific ego-subgraph to capture distinct semantics. This variant evaluates the necessity of encouraging vocabulary disentanglement in the pre-training stage.

**GRAVER (*w/o VA*).** This variant disables the vocabulary augmentation (*VA*) mechanism during fine-tuning. Specifically, we remove the generative vocabulary $\widetilde{\mathbf{g}}_i^{gen}$ in Eq. (15) and skip the support augmentation step $G_i^{\mathcal{T}} \oplus \widetilde{\mathbf{g}}_i^{gen}$. Fine-tuning is conducted directly on raw support graphs $G_i^{\mathcal{T}}$ without in-place augmentation. This ablation tests the utility of injecting class-conditioned generative subgraphs to enrich the few-shot supervision.

**GRAVER (*w/o MC*).** This variant replaces the hierarchical MoE-CoE routing network (*MC*) with a uniform composition mechanism. Specifically, we set the MoE routing weights $\mathbf{S}_M$ and CoE routing weights $\mathbf{S}_C$ in Eq. (14) to uniform distributions, *i.e.*, $\mathbf{S}_M = [1/n, \cdots, 1/n]$ and $\mathbf{S}_C = [1/N_c, \cdots, N_c]$. This variant examines whether learned adaptive routing is necessary, or whether naive equal-weight fusion suffices for knowledge reuse.

All variants are evaluated on the `ogbn-Home` dataset (cross-domain) under one-shot and five-shot settings for both node and graph classification. We report the accuracy (%) averaged over 20 runs, along with the standard deviation to measure robustness. Other settings, such as pre-trained weights, support-query splits, and training schedules, are kept identical to ensure fair comparison.

### D.3.3  Detailed Settings for Section 5.4 (RQ3)

**Convergence Efficiency.** We measure the number of training episodes required for the model to converge under the one-shot node classification setting. The convergence is defined as no accuracy increase for 50 consecutive episodes. All models are initialized with the same pre-trained weights and run on the same query set split for fairness.

**Memory Usage.** We monitor the peak GPU memory consumption (GB) during fine-tuning using NVIDIA's `nvidia-smi` tool. Measurements are taken over 20 repeated runs and averaged. Analysis includes memory used by prompt embeddings, vocabulary augmentation, routing networks, and model forward/backward passes.

**Target Dataset.** We perform one-shot node classification on the `ogbn-Tech` dataset for all efficiency evaluations due to its large graph size and complex feature space. The setting reflects real-world high-load scenarios, making it suitable for assessing practical efficiency.

### D.3.4  Detailed Settings for Section 5.5 (RQ4)

**(+) Raw Text.** For each node in graphs, we retrieve associated raw textual metadata (*e.g.*, paper titles, product names, or page summaries, depending on the specific domain) following [51, 47, 54, 9]. We tokenize the raw text using a pre-trained `BERT-Base-Uncased` tokenizer, and transform it into a dense vector by taking the mean-pooling over all output token embeddings. This vector is then concatenated with the original node features to form an enriched representation.

**(++) LLM Enhanced.** Building on the raw text, we further incorporate class-aware textual information. Specifically, for each node, we append a short class-level description (*e.g.*, a node in the "Graph Neural Networks" class may be appended with "This paper belongs to the topic of graph neural networks.") to its raw text. These descriptions are generated manually or retrieved from pre-defined templates, and are consistent within each class. The concatenated text is then encoded with the same `BERT-Base-Uncased` tokenizer as above. The resulting embedding is concatenated with the original features, similarly to the raw text setting.

**Evaluation Protocol and Datasets.** In both (+) and (++) settings, the obtained textual embeddings are directly concatenated to the original node feature vectors before alignment. This results in an extended feature space that blends structural and semantic cues. We compare three settings: original structural features, raw text-enhanced features (+), and LLM-enhanced features (++). For all settings, the model architecture and routing modules remain unchanged. Especially, in the ***zero-shot*** setting, no fine-tuning is performed, and predictions are made directly using the pre-trained encoder on the target domain. In the one-shot and five-shot settings, textual features are used alongside support samples for fine-tuning. We report results on `CiteSeer` (cross-dataset) and `Wiki-CS` (cross-domain), which are representative benchmarks with rich textual content. The zero-shot setting is evaluated by removing all supervision in the target domain and relying solely on aligned textual semantics.

### D.3.5  Detailed Settings for Section 5.6 (RQ5)

**Hyperparameter Ranges.** For both $\lambda$ and $\mu$, we sweep over the values $\{0, 0.2, 0.4, 0.6, 0.8\}$, forming a 5×5 grid in the hyperparameter space. This results in 25 configurations, each evaluated independently. To provide a smooth and interpretable view of sensitivity trends, we apply bilinear interpolation over the 5×5 accuracy matrix to obtain a denser and continuous accuracy surface. Figure 8 clearly illustrates how model performance varies with respect to hyperparameter changes.

**Evaluation Protocol.** For each configuration, we perform five-shot classification on the `PubMed` dataset for both node and graph tasks. We report the accuracy averaged over 5 random support-query splits per configuration to ensure statistical reliability. The results suggest that GRAVER maintains stable performance over a broad range of $\lambda$ and $\mu$ values, with only mild fluctuations in accuracy. This indicates that the model is relatively robust to hyperparameter tuning.

# E Implementation Details

In this section, we provide implementation details of GRAVER and baselines.

## E.1 Implementation Details of GRAVER

**Pre-training.** We pre-train GRAVER for up to 10,000 epochs, with early stopping applied if training loss does not decrease for 50 consecutive epochs to ensure efficiency without compromising performance. The aligned feature dimension is set to 64, and the hidden dimension of the encoder is 256. The number of disentangled channels (factors) $K$ is set to 4, the number of routing iterations $T$ is set to 3, and the number of convolution layers is set to 2. The trade-off hyperparameter $\lambda$ is tuned within the range of 0 to 1. For optimization, we adopt the Adam optimizer [39], with the learning rate and weight decay selected from the range of 1e-5 to 1e-2 via grid search on the validation set. All model parameters are randomly initialized to eliminate potential bias from pretrained weights.

**Fine-tuning.** We fine-tune GRAVER for up to 1,000 epochs, using early stopping based on training accuracy with a patience of 50 epochs. All architectural and optimization hyperparameters remain consistent with pre-training. During vocabulary-based augmentation, we generate 15 nodes per class to overlap with the support set. The trade-off hyperparameter $\mu$ is tuned within [0, 1] to balance the CoE-MoE objectives. Model parameters are randomly initialized to ensure fair adaptation.

## E.2 Implementation Details of Baselines

We provide the baseline implementations with their respective licenses as follows.

- GCN [40]: [MIT License] https://github.com/pyg-team/pytorch_geometric.
- GAT [91]: [MIT License] https://github.com/pyg-team/pytorch_geometric.
- GCC [69]: [MIT License] https://github.com/THUDM/GCC.
- DGI [92]: [MIT License] https://github.com/PetarV-/DGI.
- InfoGraph [81]: [License Unspecified] https://github.com/sunfanyunn/InfoGraph.
- GraphCL [116]: [MIT License] https://github.com/Shen-Lab/GraphCL.
- DSSL [102]: [Apache-2.0 License]
  https://github.com/BUPT-GAMMA/OpenHGNN/blob/main/openhgnn/models/DSSL.py.
- GraphACL [103]: [License Unspecified] https://github.com/tengxiao1/GraphACL.
- GPPT [82]: [License Unspecified] https://github.com/MingChen-Sun/GPPT.
- GraphPrompt [58]: [License Unspecified] https://github.com/Starlien95/GraphPrompt.
- GPF [14]: [MIT License] https://github.com/zjunet/GPF.
- ProNoG [121]: [License Unspecified] https://github.com/Jaygagaga/ProNoG.
- GCOPE [136]: [License Unspecified] https://github.com/cshhzhao/GCOPE.
- MDGPT [123]: [License Unspecified] https://anonymous.4open.science/r/MDGPT.
- SAMGPT [119]: [License Unspecified] https://github.com/blue-soda/SAMGPT.

For all baselines, we follow the recommended hyperparameters from the original papers or official implementations. If unavailable or suboptimal, we apply careful tuning for best performance. To ensure fairness, we standardize key settings (*e.g.*, hidden dimensions, layers) to match our method, so that performance differences reflect model design rather than external factors.

## E.3 Hardware and Software Configurations

We conduct the experiments with the following hardware and software configurations.

- Operating System: Ubuntu 20.04 LTS.
- CPU: Intel(R) Xeon(R) Platinum 8358 CPU@2.60GHz with 1TB DDR4 of Memory.
- GPU: NVIDIA Tesla A100 SMX4 with 80GB of Memory.
- Software: CUDA 10.1, Python 3.8.12, PyTorch[3] 1.9.1, PyTorch Geometric[4] 2.0.1.

---

[3] https://github.com/pytorch/pytorch
[4] https://github.com/pyg-team/pytorch_geometric

Table F.1: Accuracy (% ± std for 20 runs) of **five-shot node classification**. Best results are presented in **bold** and the runner-ups are underlined. `CR` = Cora, `CS` = CiteSeer, `PM` = PubMed, `arXiv` = ogbn-arXiv, `Tech` = ogbn-tech, `Home` = ogbn-home, `Wiki` = Wiki-CS. Color denotes domain.

| Source | Cross-Dataset | | | | Cross-Domain | | |
|---|---|---|---|---|---|---|---|
| | CS PM Home Wiki | CR PM Home Wiki | CR CS Home Wiki | CR CS PM Home Wiki | CR CS PM Wiki | CR CS PM Home | Home Tech Wiki |
| **Model / Target** | CR | CS | PM | Tech | Home | Wiki | arXiv |
| GCN (bb.) [40] | 50.19 ± 4.86 | 45.89 ± 5.42 | 50.64 ± 7.46 | 63.44 ± 5.18 | 58.24 ± 4.27 | 47.37 ± 6.06 | 35.79 ± 5.06 |
| GAT [91] | 49.03 ± 7.91 | 46.11 ± 5.12 | 52.19 ± 6.29 | 64.06 ± 6.34 | 57.40 ± 5.40 | 47.10 ± 5.42 | 36.74 ± 4.82 |
| GCC [69] | 49.81 ± 8.88 | 45.36 ± 6.40 | 53.03 ± 6.53 | 63.77 ± 5.62 | 57.94 ± 4.02 | 45.25 ± 6.92 | 36.88 ± 3.72 |
| DGI [92] | 49.89 ± 6.64 | 46.52 ± 7.10 | 53.63 ± 7.12 | 65.75 ± 6.90 | 57.07 ± 6.49 | 46.27 ± 4.02 | 38.28 ± 5.41 |
| GraphCL [116] | 53.17 ± 5.44 | 48.76 ± 7.65 | 54.65 ± 4.36 | 69.71 ± 4.44 | 56.20 ± 2.51 | 47.05 ± 6.04 | 40.41 ± 4.57 |
| DSSL [102] | 48.36 ± 6.99 | 48.48 ± 7.53 | 54.44 ± 8.08 | 67.12 ± 5.83 | 56.05 ± 5.05 | 49.36 ± 4.80 | 38.67 ± 3.67 |
| GraphACL [103] | 54.50 ± 7.33 | 50.41 ± 6.72 | 55.13 ± 4.77 | 67.12 ± 5.45 | 59.38 ± 4.98 | 51.62 ± 4.62 | 44.22 ± 5.96 |
| GPPT [82] | 51.50 ± 5.24 | 48.48 ± 6.02 | 55.22 ± 7.85 | 64.36 ± 4.98 | 58.34 ± 2.59 | 50.75 ± 3.32 | 42.06 ± 4.33 |
| GraphPrompt [58] | 54.94 ± 6.73 | 52.53 ± 5.03 | 58.03 ± 7.26 | 69.22 ± 3.98 | 61.59 ± 1.72 | 53.50 ± 3.99 | 49.18 ± 4.58 |
| GPF [14] | 54.21 ± 8.39 | 52.89 ± 7.38 | 56.38 ± 5.47 | 70.75 ± 6.38 | 61.41 ± 5.57 | 53.43 ± 5.29 | 49.17 ± 4.99 |
| ProNoG [121] | 62.20 ± 6.18 | 54.93 ± 6.48 | 57.83 ± 7.71 | 77.70 ± 6.01 | 64.98 ± 7.47 | 53.24 ± 4.39 | 54.27 ± 3.91 |
| GCOPE [136] | 55.61 ± 6.40 | 56.92 ± 5.77 | 53.64 ± 8.58 | 74.96 ± 4.17 | 62.88 ± 6.93 | 54.03 ± 5.04 | 50.77 ± 6.51 |
| MDGPT [123] | 62.71 ± 5.98 | 55.85 ± 3.27 | 58.72 ± 6.23 | 77.15 ± 5.10 | 66.30 ± 5.67 | 55.08 ± 4.71 | 54.05 ± 4.06 |
| SAMGPT [119] | 64.55 ± 6.72 | 56.43 ± 4.69 | 59.05 ± 5.97 | 79.11 ± 4.27 | 68.96 ± 4.90 | 55.95 ± 3.34 | 56.60 ± 5.02 |
| **GRAVER** (ours) | **68.68 ± 2.82** | **58.10 ± 2.34** | **61.63 ± 2.54** | **81.63 ± 2.21** | **70.43 ± 2.30** | **57.37 ± 2.21** | **59.37 ± 2.68** |

Table F.2: Accuracy (% ± std for 20 runs) of **five-shot graph classification**. Best results are presented in **bold** and the runner-ups are underlined. `CR` = Cora, `CS` = CiteSeer, `PM` = PubMed, `arXiv` = ogbn-arXiv, `Tech` = ogbn-tech, `Home` = ogbn-home, `Wiki` = Wiki-CS. Color denotes domain.

| Source | Cross-Dataset | | | | Cross-Domain | | |
|---|---|---|---|---|---|---|---|
| | CS PM Home Wiki | CR PM Home Wiki | CR CS Home Wiki | CR CS PM Home Wiki | CR CS PM Wiki | CR CS PM Home | Home Tech Wiki |
| **Model / Target** | CR | CS | PM | Tech | Home | Wiki | arXiv |
| GCN (bb.) [40] | 52.93 ± 4.08 | 43.91 ± 5.86 | 55.36 ± 5.26 | 77.52 ± 6.08 | 61.78 ± 5.88 | 40.17 ± 8.30 | 41.84 ± 5.57 |
| GAT [91] | 49.62 ± 5.09 | 45.25 ± 7.28 | 54.49 ± 7.30 | 79.77 ± 6.41 | 61.45 ± 6.83 | 39.44 ± 5.28 | 34.64 ± 4.07 |
| InfoGraph [81] | 54.65 ± 4.93 | 47.21 ± 5.16 | 59.70 ± 7.12 | 78.85 ± 5.88 | 61.94 ± 3.99 | 43.01 ± 4.32 | 40.05 ± 5.83 |
| GraphCL [116] | 55.22 ± 5.87 | 46.41 ± 3.80 | 59.94 ± 5.44 | 81.38 ± 4.67 | 62.75 ± 6.53 | 44.78 ± 5.90 | 40.98 ± 5.99 |
| DSSL [102] | 53.92 ± 5.05 | 47.49 ± 5.18 | 62.43 ± 3.61 | 78.14 ± 7.23 | 62.07 ± 7.17 | 51.67 ± 6.68 | 46.67 ± 3.55 |
| GraphACL [103] | 56.19 ± 7.08 | 45.18 ± 6.14 | 60.10 ± 7.08 | 81.50 ± 5.82 | 69.69 ± 2.53 | 48.92 ± 7.40 | 46.93 ± 7.40 |
| GraphPrompt [58] | 53.83 ± 5.19 | 50.47 ± 4.21 | 64.38 ± 6.40 | 83.31 ± 5.31 | 73.77 ± 4.53 | 55.64 ± 6.13 | 56.31 ± 4.81 |
| GPF [14] | 54.79 ± 4.73 | 53.90 ± 5.81 | 63.08 ± 6.36 | 84.79 ± 5.44 | 73.13 ± 7.38 | 56.54 ± 4.89 | 55.43 ± 5.76 |
| ProNoG [121] | 62.09 ± 5.09 | 57.64 ± 7.57 | 66.44 ± 6.17 | 84.17 ± 7.35 | 73.17 ± 6.16 | 59.52 ± 4.65 | 58.08 ± 3.68 |
| GCOPE [136] | 63.86 ± 4.75 | 58.20 ± 5.78 | 66.39 ± 3.71 | 85.99 ± 4.17 | 72.10 ± 5.36 | 58.42 ± 5.56 | 58.14 ± 3.20 |
| MDGPT [123] | 65.10 ± 4.18 | 59.32 ± 6.03 | 67.60 ± 4.64 | 84.85 ± 3.39 | 74.30 ± 7.18 | 61.99 ± 5.51 | 61.15 ± 3.24 |
| SAMGPT [119] | 68.29 ± 3.36 | 62.08 ± 5.66 | 67.94 ± 4.56 | 85.27 ± 4.62 | 75.83 ± 4.56 | 63.09 ± 2.69 | 63.82 ± 3.41 |
| **GRAVER** (ours) | **71.07 ± 2.51** | **65.45 ± 2.14** | **71.33 ± 2.39** | **89.32 ± 2.57** | **79.18 ± 2.92** | **65.85 ± 1.88** | **65.28 ± 2.58** |

# F  Additional Results

In this section, we provide additional experiment results.

## F.1  Additional Results: Few-Shot Node and Graph Classification

**Five-Shot Classification** (Table F.1 and Table F.2). We further evaluate GRAVER under the *five-shot* settings. Results consistently demonstrate that GRAVER maintains superior performance across both node and graph classification tasks in cross-dataset and cross-domain scenarios, extending the positive trends that are observed in the one-shot setting. Specifically, GRAVER achieves significant performance advantages over baselines, marked by notably higher accuracy and substantially reduced variance, emphasizing its robustness against the randomness inherent in few-shot support set selection. While the performance gap narrows slightly owing to improved baseline performance with increased labeled data, GRAVER continues to outperform competitors. These findings highlight its consistent effectiveness in leveraging generative graph vocabularies to robustly and efficiently adapt to limited labeled data. Overall, the additional results provided here reinforce the reliability and superiority of GRAVER in few-shot learning scenarios.

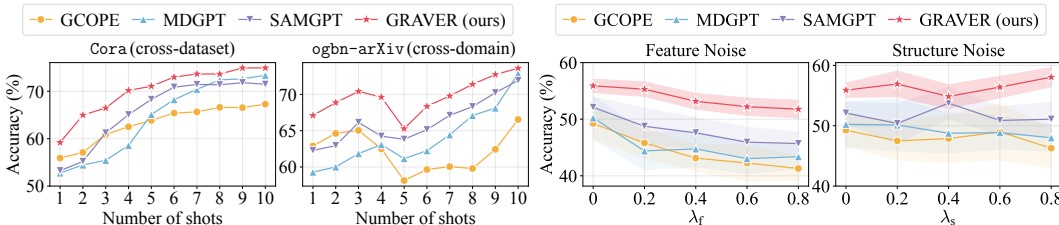

Figure F.1: $m$-shot graph classification.

Figure F.2: 1-shot graph classification (`Wiki-CS`).

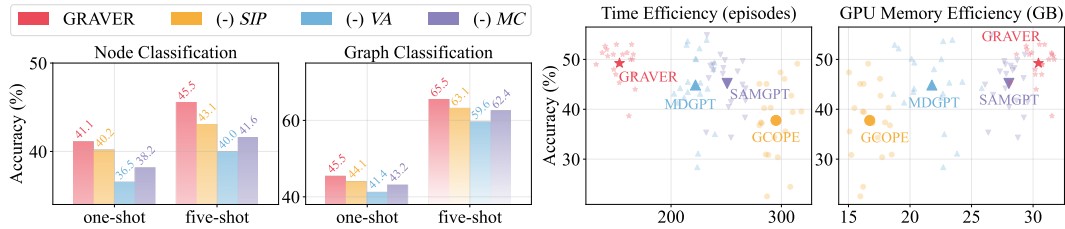

Figure F.3: Ablation studies (`CiteSeer`).

Figure F.4: Efficiency analysis (`ogbn-arXiv`).

$m$-**Shot Classification** (Figure F.1). We additionally analyze the performance under varying shot settings for graph classification tasks. Results further corroborate the conclusions drawn from Figure F.1. Specifically, the performance margin remains evident in the low-shot regime ($m \leqslant 5$), underscoring GRAVER's ability to inject meaningful generative vocabularies even with minimal supervision. Notably, in the cross-domain case, where domain shifts pose a significant challenge, GRAVER's margin remains prominent throughout, highlighting its superior generalization and robustness. These findings reinforce our core claim: generative vocabulary augmentation effectively enhances few-shot adaptation in both low-resource and more relaxed settings.

**Robustness against Random Noise** (Figure F.2). Compared to the five-shot setting, the one-shot setting reveals a more pronounced performance drop in the presence of noise for baseline methods. In contrast, GRAVER exhibits consistently high accuracy and reduced degradation across all noise levels. Under feature noise, GRAVER shows graceful performance decay, while other models experience sharp declines. Under structure noise, GRAVER not only maintains stability but occasionally improves, likely benefiting from structural diversity introduced by generative augmentation. This contrast highlights GRAVER's robustness advantage, especially in the extremely low-shot regime.

## F.2 Additional Results: Ablation Study

Figure F.3 extends the ablation study to the `CiteSeer` (cross-dataset) setting. Compared to results on `ogbn-Home` (cross-domain), the relative contributions of the three core components remain consistent. Specifically, removing *VA* causes the most substantial performance drop in both node and graph classification, underscoring its centrality to effective adaptation under extreme supervision scarcity. The degradation from removing *MC* is also evident, indicating the importance of selective knowledge routing. Notably, the performance gaps between full GRAVER and its ablated variants are ***larger*** on `CiteSeer` than on `ogbn-Home`, particularly in the five-shot setting. This suggests that GRAVER's mechanisms are especially effective when dealing with smaller, more homogeneous graphs.

## F.3 Additional Results: Efficiency Evaluation for Fine-tuning

Figure F.4 supplements the efficiency analysis on the large-scale `ogbn-arXiv` dataset. Compared to the earlier findings on `ogbn-Tech`, GRAVER continues to demonstrate strong efficiency-performance trade-offs. It consistently converges in fewer episodes and achieves higher accuracy with less variance than baselines. Although GRAVER incurs higher GPU memory usage, its performance gain per memory unit remains the highest among all methods. Notably, the gap in time efficiency is even more pronounced on `ogbn-arXiv`, suggesting that GRAVER's generative vocabulary and routing design not only stabilize training but also accelerate convergence on large and heterogeneous graphs.

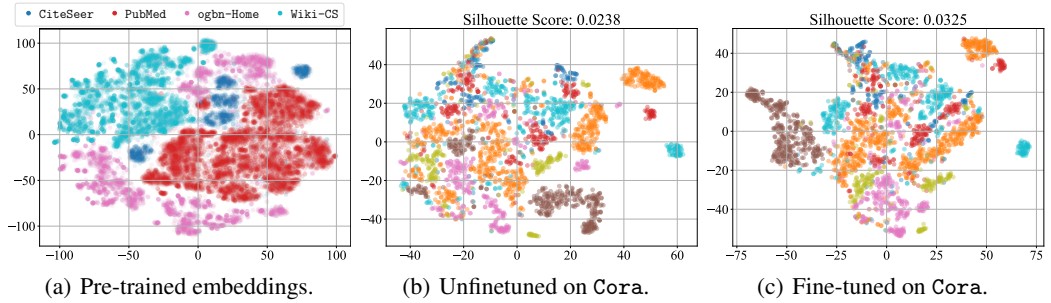

(a) Pre-trained embeddings.  (b) Unfinetuned on `Cora`.  (c) Fine-tuned on `Cora`.

Figure F.5: Visualization of node embeddings for 1-shot node classification (`Cora`, cross-dataset).

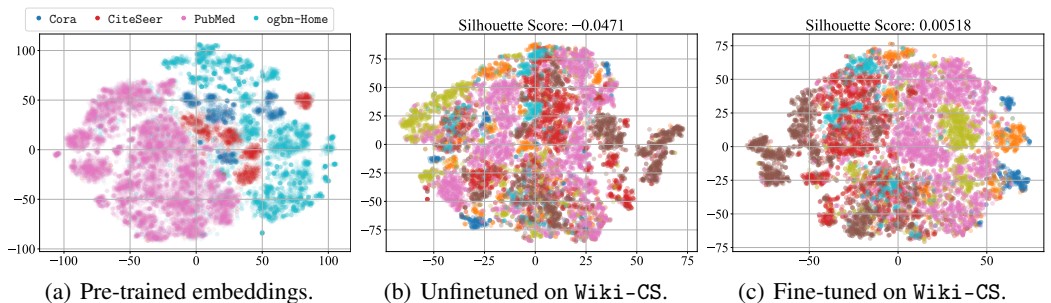

(a) Pre-trained embeddings.  (b) Unfinetuned on `Wiki-CS`.  (c) Fine-tuned on `Wiki-CS`.

Figure F.6: Visualization of node embeddings for 1-shot node classification (`Wiki-CS`, cross-domain).

## F.4  Additional Results: Node Embedding Visualization

Figure F.5 and Figure F.6 provide a qualitative comparison of node embeddings for one-shot node classification under cross-dataset (`Cora`) and cross-domain (`Wiki-CS`) settings, respectively. Across both tasks, we observe consistent improvements in the clustering structure of the embeddings after fine-tuning with GRAVER.

In both settings, the pre-trained embeddings exhibit clear separation among nodes from different source domains, which supports the effectiveness of GRAVER's multi-domain pre-training design. Specifically, the use of LLM-enhanced semantic alignment and shared dimension-wise projection enables graphs with heterogeneous features to be embedded into a unified latent space. Furthermore, the ego-graph disentanglement ensures that factor-specific patterns are preserved across domains. As a result, nodes from different graphs are aligned without sacrificing their structural diversity.

However, despite strong domain-level alignment, the pre-trained embeddings lack strong class-level separability, as evidenced by the relatively low silhouette scores (*e.g.*, 0.0238 on `Cora` and –0.0471 on `Wiki-CS`). Silhouette scores quantitatively measure the class separability of embeddings, where higher values indicate tighter intra-class cohesion and clearer inter-class boundaries. After applying one-shot fine-tuning, silhouette scores increase in both scenarios (*e.g.*, to 0.0325 on `Cora` and 0.00518 on `Wiki-CS`), showing that GRAVER is able to adapt pre-trained embeddings into more class-discriminative structures with minimal supervision. The improvement is more pronounced in the cross-dataset case (`Cora`), while still non-trivial under the harder cross-domain setting (`Wiki-CS`), where semantic and structural gaps are larger.

In summary, GRAVER's pre-training effectively aligns nodes across domains by combining LLM-based semantic projection and ego-graph disentanglement, leading to structured but class-agnostic embeddings. The low silhouette scores reflect poor class separability at this stage. After fine-tuning, the introduction of class-aware vocabulary improves silhouette scores, indicating enhanced class discriminability on top of the well-aligned multi-domain space.

# G Additional Related Work

## G.1 Multi-Domain Pre-trained Graph Foundation Models

Graph foundation models have rapidly progressed by leveraging self-supervised pre-training to capture universal patterns in graph-structured data [57, 29, 104, 124]. These models aim to distill transferable knowledge from large-scale graphs through tasks such as node feature masking, edge reconstruction, or contrastive instance discrimination. Notable examples include GCC [69], GPT-GNN [30], GPPT [82], L2P-GNN [61], and others [105, 36, 109, 8, 7, 113, 20]. After pre-training, these foundation models are fine-tuned or prompted for specific downstream tasks. However, most existing models are developed under the assumption that the pre-training and downstream domains share similar structural or semantic distributions, often involving homogeneous subgraphs [32] or graphs within a narrow domain [135]. As such, these approaches are vulnerable to domain shift and tend to underperform when directly transferred to out-of-distribution graphs.

To overcome this limitation, recent efforts have explored cross-domain graph learning, where the goal is to adapt knowledge from one domain to another dissimilar domain. Representative models in this direction include GCOPE [136], OMOG [54], PGPRec [112], UniAug [88], and UniGraph [23], which focus on extracting domain-invariant representations to reduce the distribution gap. Despite their effectiveness in single-source transfer settings, these models often lack the capacity to generalize across multiple diverse domains simultaneously. Furthermore, many are designed with task-specific or domain-specific assumptions, limiting their scalability as universal graph foundation models.

To advance toward multi-domain graph foundation models, several recent works explore pre-training strategies that can encode graph knowledge across heterogeneous domains. OFA [54] and HiGPT [87], for example, introduce LLM-guided frameworks that convert node information into natural language for alignment. While promising, these methods are restricted to text-attributed graphs [95, 85] and cannot handle general-purpose structural graphs. Other approaches, such as GCOPE [136], insert virtual nodes to serve as domain bridges, yet these do not achieve semantic feature alignment. MDGPT [123] proposes the use of domain tokens to align multi-domain representations, but it employs SVD-based alignment that overlooks semantic consistency across domains.

## G.2 Graph Foundation Models Fine-tuning with Prompt

Fine-tuning pre-trained graph foundation models is a dominant strategy for adapting general-purpose representations to downstream applications. Conventional fine-tuning methods typically update either the full parameter set or selected components of the model to optimize for specific tasks [130, 84, 139, 108, 85, 115, 33, 35, 129, 125]. While such approaches can yield strong task-specific performance, they are often resource-intensive and prone to overfitting.

To address these limitations, parameter-efficient fine-tuning (PEFT) methods have been proposed. Instead of retraining the entire model, PEFT techniques update only a small subset of parameters [143, 89, 19, 106] or incorporate lightweight modules [49, 147, 110] that can be trained with significantly fewer resources. These strategies retain much of the model's pre-trained knowledge while improving adaptability and reducing computational overhead. Despite their advantages, PEFT still faces difficulties in generalizing across domains, where task variability and domain shift may compromise their effectiveness due to the inability to handle inter-domain discrepancies.

In parallel, prompt-based fine-tuning is a promising alternative. Originating from natural language processing [55, 141, 140, 11], prompt learning reframes task adaptation as conditioning the model via learned input tokens or templates, rather than altering model parameters. For graph learning, this paradigm has been extended through the design of graph prompts, which are structured inputs or embeddings that guide the behavior of pre-trained models toward downstream objectives [82, 145, 77, 132]. Prompt tuning enables efficient task adaptation with minimal parameter updates, thus mitigating overfitting and supporting better generalization, especially in few-shot settings.

Building upon this foundation, a variety of prompt learning methods for graphs have been proposed. These include GPPT [82], All in One [83], GraphPrompt [58], Self-Pro [18], SGL-PT [145], GPF-Plus [14], PRODIGY [31], MDGPT [123], HGPROMPT [118], and ProG [149], among others. These models introduce learnable prompts into architectures, allowing them to emphasize salient topological patterns or semantic features relevant to specific tasks, which has shown promise in improving cross-task transferability and enhancing model robustness on structurally complex graphs.

Nevertheless, current graph prompt methods often exhibit limited scalability in multi-domain settings. Many of them are tailored to single-domain tasks or require manual domain-specific prompt designs, restricting their flexibility when faced with diverse or evolving graph distributions. Moreover, the field still lacks a formal understanding of prompt learning in the context of graph structures, in which most methods are empirically driven, without theoretical guarantees or generalization analysis.

## G.3  GFM Pre-training with Transferable Patterns

Graph Foundation Models aim to capture structural patterns generalizable across diverse domains and tasks [69, 142, 83, 7, 148, 97, 96]. A core challenge is defining transferable subgraph patterns for pre-training since graphs lack obvious "tokens" like words or patches.

One line of work addresses this by identifying common substructures (motifs, computation trees, *etc.*) as a graph vocabulary that can be learned and reused [142, 146, 47, 138, 144, 97, 117]. For example, Wang et al. [98] propose GFT, which treats each node's computation tree (the tree unrolled from one node's message-passing neighborhood) as a token. By vector-quantizing these computation trees into a discrete vocabulary, GFT's pre-training encourages the model to represent graphs in terms of recurring structural patterns. This approach improved cross-domain generalization and reduced negative knowledge transfer in experiments, illustrating that encoding graphs via transferable pattern "tokens" can serve as a promising pre-training strategy.

Another perspective directly learns frequent subgraph motifs during pre-training [135, 4, 133, 107]. Zhang et al. [133] introduce a motif-driven contrastive framework MICRO-Graph that automatically mines significant subgraph patterns from large graphs. MICRO-Graph clusters similar subgraphs into motif slots and trains a GNN via graph-to-subgraph contrastive learning. By pulling together representations of graphs sharing a motif and pushing apart those without, the GNN learns motif-aware embeddings. This motif-based pre-training yielded improved transfer to downstream tasks like molecular property prediction, as motifs (*e.g.*, functional groups in molecules) carry semantic patterns that generalize across graphs. Similarly, MGSSL [135] incorporates a motif generation objective: a GNN is trained to reconstruct masked substructures and generate likely motif segments within graphs. By capturing the "building blocks" of graphs, MGSSL's multi-level pre-training (at both atom and motif levels) significantly boosted downstream molecular classification performance.

Beyond motifs, researchers have explored universal structural pattern learning for cross-task transfer [43, 142, 148, 134]. For instance, Yin et al. [113] present $\pi$-GNN, which is pre-trained on synthetic graphs with known explanatory subgraphs. $\pi$-GNN uses a dedicated structural pattern learning module to extract diverse, universal subgraph patterns that occur across graph types, which are then integrated as a comprehensive graph representation. The pre-trained $\pi$-GNN can be fine-tuned on real datasets, where it not only improves accuracy but also generalizes its interpretability to new tasks. Notably, a single $\pi$-GNN pre-trained on graph-level tasks achieved state-of-the-art explanation quality even on node classification tasks, indicating that the "universal structural patterns" it learned are transferable across task formats.

Early self-supervised graph pre-training methods (*e.g.*, context prediction or graph-level contrastive learning) also implicitly encourage the learning of generalizable structure [26]. However, these often rely on dataset-specific augmentations or predictive tasks. In contrast, the recent trend is to explicitly encode structural commonalities, whether via a subgraph vocabulary [98], motif discovery [133], or synthetic pattern pre-training [113], to build graph foundation models with stronger out-of-distribution robustness. These approaches collectively demonstrate that transferable graph patterns (from small motifs up to rooted computation trees) form a powerful basis for general-purpose graph learning.

## G.4  GFM Fine-tuning with Generative Augmentations

After pre-training, adapting graph models to specific downstream tasks can be further enhanced by generative data augmentation techniques [30, 137, 7, 84, 115, 86, 111]. Rather than only tuning on the limited observed graph data, recent works use graph generators to synthesize additional training examples or to adjust the model to the target domain's underlying graph distribution. Such generative augmentations go beyond traditional edge or node perturbations by creating new realistic graph structures, often in a context-aware manner.

One prominent approach is to leverage graphons, which are continuous graph generation functions [2, 68, 72, 12, 71, 84]. Han et al. [21] propose $\mathcal{G}$-Mixup, which adapts Mixup to graphs by interpolating between the underlying graphons of different classes. By augmenting with inter-class hybrid graphs, $\mathcal{G}$-Mixup substantially improved GNN generalization and robustness against noisy labels. The use of a generative graph model is key: it produces realistic graphs that capture smooth transitions between structural patterns of different classes, something unattainable by random edge swaps.

While $\mathcal{G}$-Mixup creates new input graphs, other work focuses on generative fine-tuning objectives to better fit a pre-trained model to the target graph data [48, 84, 9, 53]. A recent example is G-Tuning by Sun et al. [84], which addresses the issue of structural mismatch between pre-training and downstream graphs. They identify that a pre-trained GNN might overfit to patterns from its pre-training graphs and struggle to capture a new graph's structure if the generative processes differ. G-Tuning tackles this by explicitly preserving the "generative patterns" of the downstream graph during fine-tuning. It shows that treating fine-tuning as fitting a generative graph model helps retain broad generalization ability while adapting to new structures.

Another category of generative augmentation focuses on subgraph generation and replacement [3, 131, 41, 101, 24, 22]. These methods generate new sub-components of graphs to enrich training data or to regularize model explanations. For instance, Liu et al. [57] introduce an environment replacement as part of a graph rationalization framework. By training the GNN on both original and these augmented examples, the model learns to identify the subgraph's influence independent of a specific context. Such context-aware subgraph augmentations improve model robustness and interpretability, as the GNN sees multiple generated environments for the same substructure. The success in molecular property prediction tasks demonstrates that generatively augmenting graph contexts can help the model generalize the learned subgraph patterns beyond the limited surroundings.

More broadly, mixture-of-experts (MoE) have been applied to handle diverse generative regimes [27, 94, 59, 99, 20, 123]. Rather than a single generator, multiple generative "experts" can be trained, each specializing in a certain structural pattern or data distribution. Wu et al. [99] develop GraphMETRO, an MoE-based GNN that implicitly performs augmentation by routing each graph through an expert tuned to its distributional subdomain. Each expert produces a representation focusing on a particular structural shift, and a gating network combines their outputs. This idea could be extended by having each expert generate a slight variant of the input and then ensembling the predictions.

In summary, generative augmentations for graph fine-tuning go beyond traditional techniques like edge drops or feature noise by synthesizing structurally meaningful examples or aligning with target distributions. This provides richer training signals that better capture task-specific graph patterns. Empirical studies show that approaches such as graphon-based sampling [21], subgraph replacement [57], and generative alignment [84] consistently improve robustness and generalization. These trends underscore the growing interplay between graph generation and representation learning, enabling more adaptable and data-efficient graph foundation models.

## H   Limitations and Broader Impact

**Limitations.** While GRAVER demonstrates strong performance, several peripheral aspects remain to be explored. The vocabulary construction relies on class-aware supervision, which may limit applicability in unsupervised or open-world scenarios. Extending toward unsupervised or weakly supervised vocabulary discovery could enhance generality. In addition, the method is mainly evaluated on classifications, and its effectiveness on more complex reasoning settings, such as link prediction or anomaly detection, remains unclear. Finally, the reuse and compression of disentangled vocabularies across tasks have not been thoroughly examined, which may offer further gains in scalability.

**Broader Impact.** This work contributes to the growing field of GFMs by addressing a core challenge in robust and efficient few-shot adaptation. The proposed generative vocabulary framework has the potential to improve the stability and scalability of GFM deployment in real-world applications such as scientific discovery, bioinformatics, and recommendation systems, where labeled data is often scarce. By enabling more reliable cross-domain generalization, GRAVER may facilitate broader access to graph-based intelligence capabilities across under-resourced domains. We do not foresee any direct negative societal impacts, as the proposed method does not involve human subjects, sensitive data, or high-risk generative content. Instead, it aims to strengthen the foundation of trustworthy and generalizable graph learning.

