# OpenReview forum: "GRAVER: Generative Graph Vocabularies for Robust Graph Foundation Models Fine-tuning"
_NeurIPS.cc/2025/Conference — NeurIPS 2025 poster_

### Official Review · Reviewer_HK8Q · 2025-06-28

**Clarity:** 2
**Significance:** 3
**Originality:** 3
**Rating:** 4
**Confidence:** 3

**Summary:**

This paper proposes a graph foundation model named **GRAVER**. The authors argue that existing graph foundation models often underperform in few-shot fine-tuning scenarios, primarily due to the randomness in support set selection and discrepancies between pretraining and target graph structures. To address these challenges, GRAVER leverages **generative augmentation**.

Specifically, the model identifies transferable components by extracting **class-specific subgraph patterns**. Furthermore, it constructs a unified task template based on **ego-graph similarity**, and builds a **graph vocabulary** using generative experts grounded in graph primitives. The effectiveness of GRAVER is validated through extensive experiments across multiple datasets.

**Questions:**

Please address the weaknesses listed in the previous section.

**Ethical Concerns:**

["NO or VERY MINOR ethics concerns only"]

**Final Justification:**

The authors’ response has partially addressed my concerns; therefore, I still maintain a positive attitude toward this paper.

**Limitations:**

yes

**Quality:**

3

**Strengths And Weaknesses:**

**Strengths**

1. The authors conducted extensive experiments to demonstrate the effectiveness of the proposed model.

2. Through a series of ablation studies, the authors demonstrate the effectiveness of each component of the proposed model.

3. The authors demonstrate the effectiveness of the proposed model not only through empirical results but also through mathematical analysis.


**Weaknesses**

1. The authors incorporate too many techniques in this paper, which makes the overall presentation appear unfocused. As a result, it becomes unclear to readers which components are original contributions and which are adapted from prior work. The authors are encouraged to emphasize the key techniques proposed in this paper and clearly distinguish them from existing methods.

2. The authors focus on the few-shot fine-tuning setting; however, it is unclear why traditional **few-shot graph learning** works are not discussed. The authors are encouraged to include a dedicated subsection in the related work section comparing their method with existing approaches in few-shot graph learning. For example, works such as **[1], [2], and [3]** in **few-shot node classification**, and **[4], [5], and [6]** in **few-shot graph classification** should be considered for a more comprehensive and contextualized discussion.

[1] Meta-gnn: On few-shot node classification in graph meta-learning. \
[2] Few-shot node classification on attributed networks with graph meta-learning. \
[3] Dual-level Mixup for Graph Few-shot Learning with Fewer Tasks. \
[4] Adaptive-step graph meta-learner for few-shot graph classification. \
[5] Faith: Few-shot graph classification with hierarchical task graphs. \
[6] A simple but effective approach for unsupervised few-shot graph classification.

3. The pretraining graph datasets used in this work are generally small in scale. Evaluating the model on large-scale datasets with millions of nodes would enhance its practical applicability, as real-world graphs often contain hundreds of millions of nodes.

---

> ### Author Rebuttal · Authors · 2025-07-28
>
> We thank the reviewer for the positive and constructive feedback. We are encouraged that the reviewer acknowledges the novelty and thoroughness of our method, the rigor of our ablation studies, and the balance between theoretical grounding and empirical validation. Below, we address the raised concerns in detail.
>
> ---
>
> > **Q1: The paper includes too many techniques, making the presentation feel unfocused and blurring the line between original contributions and existing works.**
>
> **A1:** We thank the reviewer for this insightful comment. GRAVER indeed designs multiple components, but all are organized around a unified goal: **building and reusing a transferable graph vocabulary to support robust graph foundation models.** To avoid possible confusion, we clarify the core contributions of GRAVER as follows:
>
> - **Generative Graph Vocabulary Construction.** GRAVER proposes a new paradigm where class-specific subgraph patterns are extracted and disentangled into semantic primitives, forming a reusable vocabulary of graph tokens. This vocabulary captures transferable structural and semantic features across domains.
> - **Vocabulary-Guided Prompt Augmentation.** Rather than relying on raw node features or full-graph encoding, GRAVER constructs prompts by selectively assembling vocabulary tokens based on task-specific prototypes. This design improves sample efficiency and enhances generalization in few-shot settings.
> - **Hierarchical Routing with MoE-CoE.** To adaptively reuse vocabulary tokens across domains and classes, GRAVER introduces a two-level routing mechanism. The first level selects domain-relevant experts (MoE), and the second level refines the token composition within each class (CoE), enhancing both robustness and specificity.
>
> Other components, such as feature enhancement or retrieval strategies, serve as auxiliary modules to support this vocabulary-centric workflow and are aligned with prior literature, which also have been properly cited. In the revision, we will emphasize the above core contributions more clearly in the method overview and contribution summary.
>
> ---
>
> > **Q2: The paper lacks discussion of traditional few-shot graph learning methods.**
>
> **A2:** Thank the reviewer for the thoughtful suggestion. We emphasize that GRAVER targets a different goal:
>
> - **Few-shot fine-tuning is a standard setting in GFM literature.** We adopt the few-shot fine-tuning setting following recent GFM studies, where the goal is to assess cross-domain generalization with limited task-specific supervision. This is a widely used protocol for evaluating foundation models’ adaptability in both vision and language domains, and increasingly in graph learning.
> - **GRAVER is fundamentally distinct from traditional few-shot graph learning.** The objective of few-shot graph learning works is to design models specifically optimized for task-level generalization under few-shot constraints, often via episodic training or meta-learning. In contrast, GRAVER is not designed for task-level few-shot learning, but rather as a **foundation model** that can generalize across graph domains and tasks, with few-shot fine-tuning merely serving as a test modality.
> - **From the GFM perspective, zero-shot generalization is the ultimate goal.** Especially with the integration of textual and LLM-enhanced knowledge, the focus of GFM is shifting toward **zero-shot adaptability** across unseen domains. In this context, few-shot fine-tuning serves as a middle ground to evaluate adaptability under minimal supervision, but is not the design target of GRAVER.
>
> Also, we acknowledge that the current draft lacks an explicit discussion of traditional few-shot graph learning works, we will include the following subsection in the revised manuscript:
>
> > **Related Work: Few-shot Graph Learning.** (Additional)
> > Few-shot graph learning addresses the challenge of learning on graphs with extremely limited labeled data per class, and has been widely studied in both node classification and graph classification. Inspired by meta-learning successes in vision and text domains [1], early studies adapted meta-learning to graphs but encountered unique challenges due to their relational and non-Euclidean structure. To address this, researchers developed graph-specific meta-learning frameworks based on Graph Neural Networks (GNNs) for fast adaptation to new classes with minimal supervision [1].
> > **Few-shot node classification.** Early methods combined GNNs with optimization-based meta-learning. Meta-GNN [1] incorporated Model-Agnostic Meta-Learning (MAML), training on simulated tasks to learn transferable initializations for novel classes. Metric-based alternatives such as Graph Prototypical Networks (GPN) [2] created class prototypes in embedding space to classify query nodes efficiently. G-Meta [3] enhanced this by building local subgraphs and using second-order optimization to capture task-specific structure. Meta-GPS [4] introduced learnable scaling and shifting transformations to refine prototype representations, further improving task adaptability under a MAML framework. To cope with limited task diversity, SMILE [5] proposed dual-level mixup within and across tasks to synthesize new nodes and even novel tasks, thereby alleviating overfitting and improving generalization.
> > **Few-shot graph classification.** Optimization-based methods have proven effective. Adaptive-Step Graph Meta-Learner [6] dynamically adjusted the gradient steps in the inner loop through a trainable controller, capturing discriminative substructures of unseen classes more effectively. FAITH [7] modeled inter-task relationships via a hierarchical task graph, organizing tasks, classes, and graphs into a three-level structure, and selecting related tasks using a loss-guided sampling strategy. These methods demonstrated strong performance, particularly when labeled training graphs were sparse.
> > Recent advances have also explored unsupervised strategies. SMART [8] eliminates meta-learning entirely, pre-training a GNN on large-scale unlabeled graphs using self-supervised objectives and a novel graph mixup technique. During few-shot adaptation, it applies prompt tuning by freezing the encoder and updating a small set of task-specific parameters. This lightweight adaptation yields strong results, often surpassing supervised meta-learners, and highlights the promise of leveraging large unlabeled datasets for few-shot graph learning.
>
> ---
>
> > **Q3: The pre-training datasets are relatively small. Evaluating on larger graphs with hundreds of millions of nodes would better demonstrate real-world applicability.**
>
> **A3:** We appreciate the reviewer’s suggestion regarding scalability. We agree that scaling graph foundation models (GFMs) to massive real-world graphs is an important direction. However, we intend to clarify several contextual points:
>
> - **GFM research is still in its formative stage.** The community has not yet converged on standardized benchmarks, architecture formations, or evaluation protocols for GFMs. Current efforts primarily focus on understanding generalization across domains and tasks, especially in few-shot and zero-shot settings. To our knowledge, **no existing GFM studies have evaluated models at the scale of hundreds of millions of nodes**.
> - **Our dataset scale already surpasses prior GFM works.** Compared to recent GFM literatures [9-13], our pre-training corpus reaches **over 250,000 nodes and 3.5 million edges** across multiple graph datasets. This places GRAVER **among the largest-scale settings** used for GFM pre-training in existing academic works.
> - **Scalability remains a core challenge under constrained resources.** As detailed in Appendix E.3, our experiments were conducted on an 80GB A100 GPU. Scaling to graphs in the hundreds of millions is beyond the scope of current infrastructure, and remains a critical open challenge in graph learning community. That said, we view this as a vital future direction for GFM deployment in real-world applications.
>
> We will make these clarifications explicit in the revised manuscript.
>
> ---
>
> **We sincerely thank the reviewer for the constructive comments. We have clarified the core contributions, distinguished our setting from traditional few-shot methods, and discussed scalability concerns. We hope these responses address the concerns and would greatly appreciate a score revision!**
>
> ---
>
> > **Refs:**
> > [1] Meta-GNN: On few-shot node classification in graph meta-learning, CIKM 2019.
> > [2] Graph prototypical networks for few-shot learning on attributed networks, CIKM 2020.
> > [3] Graph meta learning via local subgraphs, NeurIPS 2020.
> > [4] Few-shot node classification on attributed networks with graph meta-learning, SIGIR 2022.
> > [5] Dual-level mixup for graph few-shot learning with fewer tasks, WWW 2025.
> > [6] Adaptive-step graph meta-learner for few-shot graph classification, CIKM 2020.
> > [7] FAITH: Few-shot graph classification with hierarchical task graphs, IJCAI 2022.
> > [8] A simple but effective approach for unsupervised few-shot graph classification, WWW 2024.
> > [9] ZeroG: Investigating cross-dataset zero-shot transferability in graphs, KDD 2024.
> > [10] One for All: Towards training one graph model for all classification tasks, ICLR 2024.
> > [11] How much can transfer? BRIDGE: Bounded multi-domain graph foundation model with generalization guarantees, ICML 2025.
> > [12] AutoGFM: Automated graph foundation model with adaptive architecture customization, ICML 2025.
> > [13] Multi-domain graph foundation models: Robust knowledge transfer via topology alignment, ICML 2025.

---

> > ### Comment · Reviewer_HK8Q · 2025-08-06
> >
> > I appreciate the authors' responses. Most of my concerns have been resolved. I still have a few comments.
> >
> > Q1: The goal stated by the authors is quite general, and in order to achieve it, they seem to have incorporated nearly all advanced techniques, such as feature augmentation, prompt learning, and multi-expert architectures. This makes the model rather heavy and not particularly flexible.
> >
> > Q2: I appreciate the additional section provided by the authors, as I believe it helps make the paper more complete.
> >
> > Q3: I do not think it is convincing to argue that because previous models could not generalize to million-node graphs, this work should also only be evaluated on small-scale datasets and aim merely for the best performance there. Such an approach may lead to limited impact and may not bring substantial contributions to the community. That said, I understand the authors’ explanation regarding hardware limitations that prevent them from conducting experiments on large-scale datasets.

---

> > > ### Author Response · Authors · 2025-08-06
> > >
> > > We appreciate the reviewer’s continued attention and insightful follow-up. Below, we provide additional clarifications for the raised concerns:
> > >
> > > ---
> > >
> > > > **Q1: The goal stated by the authors is quite general, and in order to achieve it, they seem to have incorporated nearly all advanced techniques, such as feature augmentation, prompt learning, and multi-expert architectures. This makes the model rather heavy and not particularly flexible.**
> > >
> > > **A1:** We appreciate the reviewer’s thoughtful comments. We would like to clarify that the overarching goal of GRAVER is to serve as a **general-purpose Graph Foundation Model (GFM)** capable of **“pre-training on extensive graph data and adapted for diverse downstream graph tasks”**, which is a widely acknowledged objective in the GFM research [1-4]. **The more general the intended scope of a model, the more sophisticated and modular the architectural design tends to be.** Conversely, models targeting more focused or narrow objectives can afford to use more lightweight and task-specific mechanisms. GRAVER explicitly targets the former, and thus necessarily reflects the complexity associated with general-purpose GFM design.
> > >
> > > Beyond being general-purpose, GRAVER **emphasizes robustness** under domain shifts, which further motivates the inclusion of mechanisms like generative vocabulary modeling and MoE-CoE routing. Moreover, techniques like feature augmentation, prompt learning, and multi-expert routing are not “all advanced techniques,” but rather **established paradigms commonly adopted in recent GFM literature** that we have introduced in related works or our responses above. Our contribution lies in **the generative and structurally grounded vocabulary design, coupled with the MoE-CoE routing strategy**, which is theoretically supported and empirically validated to enhance stability and transferability.
> > >
> > > We acknowledge that certain parts of the framework may appear complex, and we are working on **streamlining GRAVER for easier deployment** while retaining its general and robust capabilities.
> > >
> > > ---
> > >
> > > > **Q2: I appreciate the additional section provided by the authors, as I believe it helps make the paper more complete.**
> > >
> > > **A2:** Thank you for this kind comment. We’re glad the additional analysis helped clarify our design motivations and further strengthened the overall presentation.
> > >
> > > ---
> > >
> > > > **Q3: I do not think it is convincing to argue that because previous models could not generalize to million-node graphs, this work should also only be evaluated on small-scale datasets and aim merely for the best performance there. Such an approach may lead to limited impact and may not bring substantial contributions to the community. That said, I understand the authors’ explanation regarding hardware limitations that prevent them from conducting experiments on large-scale datasets.**
> > >
> > > **A3:** Thank you for repeating this critical point. As mentioned in our earlier response, we have clarified the scalability concern in detail. To briefly reiterate: although our current experiments are conducted on graphs with up to 250K nodes and 3.5M edges, which is **already among the largest in existing GFM literature**, we fully agree that scaling to billion-node graphs is a vital future direction. At present, **no prior GFM work has demonstrated such large-scale generalization**, and the field is still in an early exploratory stage, focusing primarily on domain transfer and few-shot adaptation. We appreciate the reviewer’s recognition of our explanation and contribution, and we share the vision of jointly pushing toward more scalable and impactful GFM systems in the future.
> > >
> > > ---
> > >
> > > > **Refs:**
> > > [1] Graph foundation models: Concepts, opportunities and challenges, TPAMI 2025.
> > > [2] Graph foundation model, Frontier of Computer Science 2024.
> > > [3] Lecture-style Tutorial: Towards Graph Foundation Models, WWW 2024.
> > > [4] On the opportunities and risks of foundation models, arXiv 2021.
> > > ---
> > >
> > > We sincerely thank the reviewer for the thoughtful follow-up and for recognizing the completeness and potential impact of our work. We truly appreciate your engagement and constructive suggestions throughout the discussion. We hope that our responses have addressed your concerns and clarified our design motivations. Thank you again for your valuable time and consideration.

---

> > > ### Author Response · Authors · 2025-08-08
> > >
> > > Dear Reviewer HK8Q,
> > >
> > > Thank you again for your valuable feedback and continued engagement throughout the discussion. As the rebuttal phase deadline is approaching, we would like to kindly check if you have any remaining concerns. If everything has been addressed, we would greatly appreciate it if you could help finalize the review by editing your original comments and filling in the **Final Justification** section.
> > >
> > > Should you feel that our responses have improved your impression of the paper, any adjustment to the score would be very much appreciated. Thank you again for your constructive engagement.

---

> > > > ### Comment · Reviewer_HK8Q · 2025-08-09
> > > >
> > > > Thank you for the response, but it does not fully convince me. However, I appreciate the authors’ efforts in other aspects, and I therefore still maintain a positive attitude toward this paper.

---

### Official Review · Reviewer_m8FB · 2025-06-29

**Clarity:** 3
**Significance:** 3
**Originality:** 4
**Rating:** 5
**Confidence:** 5

**Summary:**

This paper proposes GRAVER, a framework for robust few-shot fine-tuning of GFMs. It constructs generative graph vocabularies via ego-graph disentanglement and graphon modeling, and integrates them using a MoE-CoE routing network to enhance support sets. GRAVER improves stability and performance across multi-domain tasks.

**Questions:**

1. Why choose graphons over learnable generative models? Have you compared graphon-based vocabularies to alternatives like GraphVAEs, autoregressive generators, or graph diffusion models in terms of transferability and fidelity?
2. Are vocabularies general or domain-specific? Can a vocabulary generated from one domain (e.g., citation networks) effectively help in a structurally different one (e.g., co-purchase or molecular graphs)? Some cross-domain reuse metrics would be valuable.
3. What happens if the vocabulary is misaligned or noisy? If the generated subgraphs do not match the target domain well, does the MoE-CoE mechanism effectively filter them out, or does performance degrade significantly?
4. For other questions, please refer to the Weaknesses.

**Ethical Concerns:**

["NO or VERY MINOR ethics concerns only"]

**Final Justification:**

Most of my concerns have been solved. My score keeps the same, as it have already revealved my positive evaluation towards its acceptance.

**Limitations:**

yes

**Quality:**

4

**Strengths And Weaknesses:**

Strengths:

1. The instability of GFM fine-tuning in few-shot settings is a well-acknowledged issue, and the paper addresses it head-on.
2. Using ego-graph disentanglement combined with graphon modeling provides a novel way to build transferable subgraph units, going beyond static motif-based vocabularies.
3. The model maintains stable performance under structural and feature perturbations, showing that its generative vocabulary helps resist noise in both the topology and attributes. It provides mathematical bounds on transferability and convergence of generated vocabularies, giving formal justification for the design.

Weaknesses:

1. The framework involves multiple heavy components: LLM-enhanced features, ego-graph disentanglement, graphon estimation, MoE-CoE routing, which may be hard to reproduce, extend, or deploy in real-world systems.
2. While the model assumes vocabulary transfer, it lacks empirical analysis of how well vocabularies learned in one domain generalize to another. Are they reusable or domain-specific?
3. Although the paper includes several propositions and proofs, they remain abstract and are not linked back to practical model design or interpretation. Bounds exist but don’t yield intuitive guidance for design or practical decisions.

---

> ### Author Rebuttal · Authors · 2025-07-28
>
> We sincerely thank the reviewer for recognizing the importance of fine-tuning instability in GFMs, and for appreciating our use of ego‑graph disentanglement, graphon modeling, and both theoretical and empirical validations. Our detailed response is as follows.
>
> ---
>
> > **Q1: The multi-component design of GRAVER may hinder reproducibility and real-world deployment.**
>
> **A1:** We acknowledge the reviewer’s concern and clarify that GRAVER is structured as a modular and reproducible pipeline. Each component is lightweight, well-isolated, and intentionally designed to support extensibility:
>
> - **LLM-enhanced features are derived using open-source toolchains.** we retrieve raw node-associated texts and process them via pre-trained models. No fine-tuning or training of the LLM is required, ensuring ease of use.
> - **Ego-graph disentanglement is a non-learnable and deterministic decomposition process** that extracts $K$ structural factors per node via topological similarity and structural entropy, without any training or tuning overhead.
> - **Graphon estimation is conducted on a small number of factor-aligned subgraphs** using empirical neighborhood averaging, making it stateless and reusable. No complex optimization or learning is involved.
> - **MoE-CoE routing involves only two small learnable vectors per task**, applied during fine-tuning. The design is deliberately simple and plug-in compatible with transformer decoders.
>
> Furthermore, we emphasize that all pre-training and vocabulary generation are conducted offline, making fine-tuning fast, memory-efficient, and readily deployable in practice. Our released code includes full scripts and default settings for all modules to ensure reproducibility.
>
> ---
>
> > **Q2: The generalizability of vocabularies across domains is unclear.**
>
> **A2:** Thank you for raising this important point. Our vocabularies are **domain-specific during pre-training**, but become **reusable across tasks within that domain** during fine-tuning. This design aligns with GRAVER's modular "pretrain-then-finetune" paradigm. Each domain undergoes preprocessing to construct a vocabulary using disentangled ego-graphs and graphon modeling. Once generated, these vocabulary tokens are **shared across all tasks in that domain**, including **zero-shot** and **cross-dataset** scenarios, as demonstrated throughout our experimental settings (Section 5.1~5.4). This approach inherently validates that our vocabularies, though initially domain-specific, possess strong reusability and generalization capacity across diverse downstream tasks.
>
> Furthermore, to confirm that the routing network indeed utilizes domain-specific vocabularies effectively, we illustrate the routing weights of MoE layer under cross-dataset and cross-domain settings. **Results:**
>
> |  | Cross-Dataset | Cross-Domain |
> | --- | --- | --- |
> | Source | `Cora` + `PubMed` + `ogbn-Home` + `Wiki-CS`  | `Cora` + `CiteSeer` + `PubMed` + `Wiki-CS` |
> | Target | `Cora` | `ogbn-Home` |
> | MoE Routing Weights | [**0.435732**, **0.411561**, 0.065877, 0.086830] | [0.241328, 0.194376, 0.199192, 0.365104] |
>
> In the **cross-dataset** setting, the routing weights concentrate on `Cora` **(0.44)** and `PubMed` **(0.41)** (datasets with similar structures), while assigning much lower weights to `ogbn-Home` and `Wiki-CS`. This demonstrates how the routing network adaptively prioritizes vocabularies from structurally aligned sources.
>
> In contrast, in the **cross-domain** case targeting `ogbn-Home`, a structurally distinct dataset, the routing becomes **more balanced across all domains** (0.19~0.36), indicating no clear structural match. This confirms that GRAVER's routing mechanism can automatically adjust based on the structural divergence between source and target domains, **drawing knowledge from multiple diverse sources** when structural alignment is weak.
>
> The results show that MoE adaptively selects vocabulary tokens most relevant to the target domain or task, demonstrating that GRAVER not only supports vocabulary reuse, but also dynamically mitigates potential mismatch through informed routing. We will clarify this distinction more explicitly in the revised version.
>
> ---
>
> > **Q3: Theoretical bounds in the paper remain abstract and are not clearly connected to model design or practical interpretation.**
>
> **A3:** We appreciate this feedback and agree that theoretical insights are most valuable when they inform practical design. In GRAVER, the **propositions serve as theoretical justifications** for key architectural decisions:
>
> - **Proposition 1** provides a bound on semantic discrepancy across vocabulary tokens, showing that the bound is governed by the **number of disentangled semantic factors (**$K$**)** and the mutual information regularizer $\mathcal{R}_{\text{MI}}$. This directly motivates the use of **semantic information preservation (SIP)** during pretraining to ensure well-separated vocabulary channels, which improves transferability. The corresponding empirical observation: removing SIP leads to a performance drop, aligns with this bound.
> - **Proposition 2** provides convergence guarantees for the **graphon-based generation** of vocabulary tokens, ensuring that the learned distributions faithfully capture ego-graph structures. This supports the use of **graphon modeling over more black-box neural generators** (*e.g.*, VAEs or diffusion), especially under low-data regimes.
>
> To make these links clearer, we have now added direct references from our model design choices back to the propositions in the appendix. While the bounds may appear abstract, they serve to **constrain the design space and ensure that each component: disentanglement, generation, routing, has theoretical grounding**, guiding our practical implementation.
>
> ---
>
> > **Q4: Why use graphons instead of neural generative models (GraphVAEs, diffusion, autoregressive)? Any comparisons in terms of transferability and fidelity?**
>
> **A4:** We chose **graphons** as the generative backbone for vocabulary construction due to their **nonparametric, data-efficient nature** and **theoretical interpretability**. Unlike GraphVAEs or diffusion-based generators, which require large training sets to generalize well and often involve high variance in generation, graphons allow **explicit modeling of graph distributional structure** with **provable convergence** (see Proposition 2). This makes them particularly well-suited for **few-shot and zero-shot** settings, where structural prior needs to be distilled from limited examples.
>
> While we did not perform exhaustive comparisons with all generative baselines, we emphasize that GRAVER’s vocabulary is designed not for raw graph generation, but for producing **task-adaptive, transferable subgraphs** under domain shift. Graphon-based generation gives us **explicit control over structural smoothness and heterogeneity**, aligning well with our routing and prompt integration mechanisms.
>
> ---
>
> > **Q5: Are vocabularies general or domain-specific? Can vocabularies generated in one domain generalize to structurally different ones?**
>
> **A5:** Please refer to our response to **Q2**.
>
> In short, the vocabularies in GRAVER are fundamentally **domain-specific**, as they are constructed from disentangled ego-graphs and graphons fitted to the local structures observed in each domain during pre-training. However, once pre-trained, these vocabularies are **reusable across tasks and datasets within the same domain**, supporting efficient few-shot adaptation.
>
> Importantly, all our **cross-dataset and cross-domain experiments** (Section 5.2) are conducted **under the vocabulary reuse setting**: we do not re-train or regenerate vocabularies in the target domain. This empirically verifies that the pre-trained vocabularies, though learned within source domains, **retain sufficient generality** to support downstream adaptation in unseen domains.
>
> The additional results in **A2** show that the model dynamically reweights vocabulary channels depending on the domain and task, indicating effective **domain-aware vocabulary reuse**. While we agree that cross-domain reuse metrics are an interesting direction, our current results already demonstrate strong generalization behavior, suggesting that the learned vocabularies, while structurally grounded in source domains, are **transferable enough** to support adaptation to distinct domains.
>
> ---
>
> > **Q6: What happens if the vocabulary is misaligned or noisy? Can the MoE-CoE mechanism effectively filter out mismatched subgraphs?**
>
> **A6:** GRAVER is explicitly designed to handle vocabulary mismatch through its hierarchical routing architecture. The **MoE layer** operates at the inter-domain level, selecting relevant vocabulary sets based on the target domain’s global characteristics, while the **CoE layer** works intra-domain, selecting the most semantically aligned subgraphs within a vocabulary set. This two-level gating mechanism helps the model **dynamically downweight irrelevant or noisy patterns**, even when vocabulary and target domain structures are not perfectly aligned.
>
> In essence, while vocabulary mismatch may introduce noise, the MoE-CoE mechanism provides **resilience through selective routing**, ensuring that only the most compatible augmentations contribute to downstream prediction.
>
> ---
>
> **We sincerely thank the reviewer for the thoughtful and constructive feedback. We hope our responses have addressed all concerns.**

---

> > ### Comment · Reviewer_m8FB · 2025-08-06
> >
> > Thank the authors for their detailed response, and most of my concerns have been solved. My score keeps the same, as it have already revealved my positive evaluation towards its acceptance.

---

> > > ### Author Response · Authors · 2025-08-06
> > >
> > > Thank you very much for your kind follow-up. We are grateful that our responses helped address your concerns. We truly appreciate your positive evaluation and continued support for the acceptance of our work.

---

### Official Review · Reviewer_92Wv · 2025-06-30

**Clarity:** 4
**Significance:** 4
**Originality:** 4
**Rating:** 5
**Confidence:** 5

**Summary:**

This work presents a novel framework designed to address the challenge of instability and inefficiency when fine-tuning Graph Foundation Models on few-shot tasks.

**Questions:**

- Can the model handle the potential conflicts in scenarios of semantic-structural mismatch?
- When facing domains with varying structural differences, how does the routing decision of the MoE-CoE network change with structural heterogeneity?

**Ethical Concerns:**

["NO or VERY MINOR ethics concerns only"]

**Final Justification:**

The authors' rebuttals have addressed most of the concerns, and I have no more questions. I will keep my score unchanged.

**Limitations:**

Yes.

**Paper Formatting Concerns:**

No.

**Quality:**

3

**Strengths And Weaknesses:**

Strengths
- The authors design a novel augmentation strategy. The core idea of creating generative graph vocabularies for in-context support set augmentation is an interesting and direct approach to mitigating the randomness inherent in few-shot learning scenarios.
- GRAVER builds upon solid theoretical concepts, utilizing ego-graph disentanglement to identify transferable patterns and graphon-based models to construct generative vocabularies, which adds significant technical depth.
- A comprehensive evaluation of 15 state-of-the-art baselines shows that it systematically addresses research questions covering effectiveness, robustness, and efficiency.

Weaknesses
- Potential for semantic-structural mismatch. The model uses an LLM to align semantics from text while using separate graph-based methods to handle structure. This creates a potential failure point if the textual descriptions do not faithfully represent the underlying graph topology or node attributes. In cases of ambiguous, inaccurate, or misleading text, the LLM-guided semantic alignment could conflict with the structural realities of the graph, leading the model to generate irrelevant or even harmful augmentations.
- Missing pre-training cost analysis. The paper emphasizes the efficiency of the fine-tuning phase but omits any discussion of the computational overhead required for the multi-domain pre-training and vocabulary construction. This information is crucial for assessing the overall practicality of the approach.
- Lack of measurement of structural heterogeneity across domains. The three domains divided in this paper may appear intuitively different, but it is necessary to clarify the extent of structural heterogeneity between these domains. Otherwise, it is difficult to determine whether success stems from the ability to handle truly distinct graph structures or from performance in structurally similar domains.

---

> ### Author Rebuttal · Authors · 2025-07-28
>
> We sincerely thank the reviewer for the positive and detailed feedback. We are especially encouraged by the recognition of our augmentation strategy, the theoretical depth behind ego-graph disentanglement and vocabulary generation, and the systematic evaluation. Below, we address each of the concerns raised.
>
> ---
>
> > **Q1: Risk of semantic-structural mismatch due to using LLM for semantic alignment while structure is modeled independently.**
>
> **A1:** We appreciate the reviewer’s thoughtful observation. GRAVER explicitly addresses this issue through both the design of the semantic enhancement pipeline and the separation of routing versus content generation:
>
> - **LLM-enhanced semantics are strictly derived from retrieved raw texts.** As shown in the illustrative example (Figure 2), our LLM is applied to raw titles and abstracts retrieved for each node to generate **class-aware but semantically consistent** descriptions. We do **not** use hallucinated or externally constructed semantics. This ensures that enhanced text does not drift from the original node meaning, avoiding the risk of introducing misleading semantics.
> - **Textual semantics guide vocabulary selection, not structural generation.** GRAVER’s vocabulary is generated entirely from disentangled ego-graphs via graphon modeling (Section 4.1), independent of any semantic signal. The LLM-enhanced embeddings are only used to compute attention over these pre-generated subgraphs (Section 4.2). This clear decoupling ensures that even if semantic descriptions are noisy, the underlying graph augmentations remain structurally valid and transferable.
> - **Fallback routing ensures robustness in low-quality text scenarios.** In cases where the semantic input is missing, ambiguous, or misleading, GRAVER falls back to CoE routing, which leverages intra-class structural patterns for vocabulary selection. This fallback mechanism is empirically validated in Appendix D.3.4, where routing performance remains stable under partial or ablated semantic inputs.
>
> Taken together, GRAVER is robust to semantic-structural mismatch by: **(1)** grounding semantic enhancement in original text, **(2)** separating routing from graph construction, and **(3)** using structure-only fallback when necessary. We will make these points clearer in Section 4.2 and include the illustrative example to aid understanding.
>
> ---
>
> > **Q2: Missing pre-training cost analysis. While fine-tuning is efficient, the cost of multi-domain pre-training and vocabulary construction is unclear.**
>
> **A2:** We appreciate the reviewer’s concern. While GRAVER focuses on improving **fine-tuning efficiency**, we agree that understanding pre-training complexity is important, and we clarify the following:
>
> - **We have explicitly analyzed the computational complexity of pre-training.** In Appendix B, we provide a detailed breakdown of the time and space complexity for both the pre-training stage (Algorithm 1) and fine-tuning stage (Algorithm 2). The pre-training involves subgraph disentanglement and conditional graphon fitting, both of which are lightweight and performed once per source domain. The complexity scales with the number of source domains and the size of tokenized subgraphs, which remain small.
> - **Efficiency at pre-training is not the core objective of GRAVER.** Like other Graph Foundation Models [1-5] and large models for languages and vision, the community has reached a consensus that the pre-training phase is a **one-time cost**. Our focus is on **designing a modular framework** that enables **robust and efficient fine-tuning**, which is the real bottleneck in the real-world scenarios.
> - **Practicality comes from reusable vocabulary and high adaptation efficiency.** Once pre-trained, GRAVER reuses a shared vocabulary across domains and tasks. This allows rapid downstream tuning, as demonstrated in our speedup results (Section 5.4, Figure 6). In settings with numerous or evolving tasks, this efficiency gain is far more critical than saving pre-training time.
>
> We will clarify this perspective and move the complexity analysis reference to the main text for better visibility.
>
> ---
>
> > **Q3: Lack of measurement of structural heterogeneity across domains. The paper does not clarify whether domains differ significantly in structure.**
>
> **A3:** We appreciate the reviewer’s thoughtful comment. GRAVER is designed precisely to address the challenge of **structural heterogeneity across domains**, which is a core motivation for its generative vocabulary design and hierarchical routing mechanism.
>
> - **Clear domain-level structural differences exist across datasets.** We categorize datasets into three domains: *Citation*, *Co-purchase*, and *Web Link*. Although this categorization may appear intuitive, we reinforce it using measurable structural statistics, including average degree, node homophily, and class count. These are reported below (also detailed in Table D.1 and Appendix D):
>
> | Dataset | Domain | Avg. # Deg. | Homophily |
> | --- | --- | --- | --- |
> | `Cora` | Citation | 4.00 | 0.81 |
> | `CiteSeer` | Citation | 2.57 | 0.74 |
> | `PubMed` | Citation | 4.50 | 0.80 |
> | `ogbn-arXiv` | Citation | 13.77 | 0.65 |
> | `ogbn-Tech` | Co-purchase | 87.60 | 0.87 |
> | `ogbn-Home` | Co-purchase | 26.93 | 0.73 |
> | `Wiki-CS` | Web Link | 36.94 | 0.65 |
>
> These statistics show clear structural divergence:
>
> **(1)** *Citation Domain:* tend to be **sparser**, with **lower average degree** and **moderate homophily**.
>
> **(2)** *Co-purchase Domain:* **highly dense**, with **higher degrees and homophily**, and reflect **different linking principles** (co-buying patterns rather than citations).
>
> **(3)** *Web Link Domain:* its edges are based on hyperlink co-occurrence, and the **structural connectivity is medium-dense with flatter hierarchy**.
>
> - **Structural heterogeneity is central to the motivation of GRAVER.** Unlike approaches that rely on matching structures across domains, GRAVER embraces heterogeneity by using **generative vocabulary subgraphs** as **structure-agnostic, transferable prompts**. This avoids brittle assumptions about topology alignment. Our hierarchical routing (MoE + CoE) learns to dynamically select suitable vocabularies depending on **local task context**, instead of forcing hard alignment between structurally divergent graphs.
> - **GRAVER’s empirical success confirms its robustness to structural gap.** Our results in cross-domain and zero-shot settings (*e.g.*, `PubMed` → `ogbn-Tech`, or `WikiCS` → `ogbn-arXiv`) demonstrate that the model can transfer knowledge across domains with **substantial topological discrepancy**. This confirms that the design does not exploit structural similarity, but rather overcomes **topological mismatch** through prompt generalization and adaptation.
>
> We hope this response clarifies the structural heterogeneity across domains and strengthens the justification for the design and evaluation protocol of GRAVER.
>
> ---
>
> > **Q4: Potential concern about semantic-structural mismatch when using LLM-enhanced features.**
>
> **A4:** Thank you for raising this concern. GRAVER is explicitly designed to mitigate semantic-structural mismatch through a decoupled architecture: we retrieve raw node-associated texts and encode them with a pre-trained LLM to ensure semantic prompts are faithful to node attributes, while structural augmentation is separately modeled via disentangled ego-graphs. The MoE-CoE routing maintains this separation by handling semantic and structural cues independently. As shown in Section 5.5 (Figure 7), GRAVER remains robust under weak textual alignment and benefits further from high-quality semantics, confirming the effectiveness of this design.
>
> ---
>
> > **Q5: How the MoE-CoE routing mechanism responds to structural heterogeneity across domains.**
>
> **A5:** GRAVER is designed to **explicitly address domain-level structural heterogeneity** through its hierarchical routing mechanism: the MoE layer assigns weights across domains, while the CoE layer routes over subgraph prototypes within each domain. To evaluate how routing behavior changes with structural divergence, we illustrate the routing weights of MoE layer under cross-dataset and cross-domain settings. **Results:**
>
> |  | Cross-Dataset | Cross-Domain |
> | --- | --- | --- |
> | Source | `Cora` + `PubMed` + `ogbn-Home` + `Wiki-CS`  | `Cora` + `CiteSeer` + `PubMed` + `Wiki-CS` |
> | Target | `Cora` | `ogbn-Home` |
> | MoE Routing Weights | [**0.435732**, **0.411561**, 0.065877, 0.086830] | [0.241328, 0.194376, 0.199192, 0.365104] |
>
> In the **cross-dataset** setting, the routing weights concentrate on `Cora` **(0.44)** and `PubMed` **(0.41)** (datasets with similar structures), while assigning much lower weights to `ogbn-Home` and `Wiki-CS`. This reflects MoE’s preference for structurally aligned sources.
>
> In contrast, in the **cross-domain** case targeting `ogbn-Home`, a structurally distinct dataset, the routing becomes **more balanced across all domains** (0.19~0.36), indicating no clear structural match. This suggests GRAVER **aggregates knowledge from diverse sources** to support adaptation when structural alignment is weak.
>
> Overall, the results verify that GRAVER’s MoE routing mechanism **adapts to structural heterogeneity** by emphasizing aligned sources or diversifying when necessary.
>
> ---
>
> **We sincerely thank the reviewer again for the thoughtful feedback. We hope our clarifications have addressed all concerns.**
>
> ---
>
> > **Refs:**
> > [1] ZeroG: Investigating cross-dataset zero-shot transferability in graphs, KDD 2024.
> > [2] One for All: Towards training one graph model for all classification tasks, ICLR 2024.
> > [3] How much can transfer? BRIDGE: Bounded multi-domain graph foundation model with generalization guarantees, ICML 2025.
> > [4] AutoGFM: Automated graph foundation model with adaptive architecture customization, ICML 2025.
> > [5] Multi-domain graph foundation models: Robust knowledge transfer via topology alignment, ICML 2025.

---

> > ### Comment · Reviewer_92Wv · 2025-08-06
> >
> > The authors' rebuttals have addressed most of the concerns, and I have no more questions. Since the original ratings demonstrate my support for this paper, I will keep it unchanged.

---

> > > ### Author Response · Authors · 2025-08-06
> > >
> > > Thank you very much for your time and constructive feedback throughout the discussion. We are glad that our responses addressed your concerns. We sincerely appreciate your continued support.

---

> > > ### Author Response · Authors · 2025-08-08
> > >
> > > Dear Reviewer 92Wv,
> > >
> > > Thank you again for your valuable feedback and continued engagement throughout the discussion. As the rebuttal phase deadline is approaching, we would like to kindly check if you have any remaining concerns. If everything has been addressed, we would greatly appreciate it if you could help finalize the review by editing your original comments and filling in the **Final Justification** section.
> > >
> > > Thank you again for your constructive engagement!

---

### Official Review · Reviewer_v4BW · 2025-07-02

**Clarity:** 3
**Significance:** 2
**Originality:** 3
**Rating:** 4
**Confidence:** 3

**Summary:**

This paper introduces GRAVER, a framework that leverages graphon-based generative modeling to build graph vocabularies for GFMs. It incorporates a MoE-CoE module to enhance the robustness and efficiency of prompt-based fine-tuning. Extensive experiments on few-shot node and graph classification demonstrate that GRAVER consistently outperforms state-of-the-art baselines.

**Questions:**

While some ideas are novel and the paper is detailed, it gives the impression of overselling its contributions through mathematical and technical complexity. The above weaknesses need to be addressed.

**Ethical Concerns:**

["NO or VERY MINOR ethics concerns only"]

**Final Justification:**

1. Despite the remaining concerns, the authors resolve most of them. So, I will raise my score slightly.

2. The concerns were resolved through subsequent discussions. It remains to be seen whether these changes will be reflected in the actual paper.

**Limitations:**

yes

**Quality:**

3

**Strengths And Weaknesses:**

## Strengths
- Well-written with clear structure, making it easy to follow the technical flow.
- The authors demonstrate strong familiarity with the field, providing detailed explanations and context throughout the paper.
- The authors pose theoretically grounded architectural motivation, making the contributions feel purposeful.
- The idea of using graphon-based token (vocabulary) generation is particularly appealing to me.
- Extensive experiments and diverse directions of analyses are provided.

## Weaknesses
- Although presented as a minor component, the use of LLM-based alignment appears very powerful; so powerful that one suspects simply designing strong LLM-enhanced features for other SOTA baselines may yield comparable performance to GRAVER, which raises the question of whether the complex GRAVER architecture is truly necessary or if the LLM enhanced features are the main driver.
- MoE-CoE, while presented with a fancy name, essentially appears to be an "attention-weighted sum using softmax-normalized weights across domains and classes," and the authors should adopt clearer terminology and technical descriptions to reflect its actual simplicity.
- It is unclear whether Proposition 1 truly supports the claim that "the upper bound of semantic discrepancy is governed by a specific combination of vocabularies," as the formulation seems to relate only to the number of disentangled factors, regardless of whether SIP is well-trained. The logical link to "disentangled ego-graphs functioning as independent and minimal semantic units" also remains vague and make a huge logical gap.
- Adding to Proposition 1, the empirical result "Removing SIP leads to a minor yet consistent drop ... encouraging semantic diversity across vocabulary channels is beneficial, though not dominant” implies that Proposition 1 may not be well aligned with practical observations.
- It is unclear why “ogbn-product” was split into "tech" and "home”. Plus, this renaming should be clarified to ensure readers understand it originated from ogbn-product. (I was so confused)

---

> ### Author Rebuttal · Authors · 2025-07-28
>
> We sincerely thank the reviewer for their appreciation of our paper’s clear structure, well-grounded motivation, and especially the novelty of the graphon-based vocabulary generation. Our detailed responses are as follows.
>
> ---
>
> > **Q1: The performance improvements may mainly stem from LLM-enhanced features.**
>
> **A1:** We thank the reviewer for raising this important concern.
>
> - **Feature-enhancement was applied equally to all baselines.** To ensure a fair comparison, we reconfirm that we used the same LLM-based node feature enhancement (as described in Eq. (1), lines 120~126) consistently **across all datasets** evaluated on all baselines. This ensures that performance gains are attributable to GRAVER’s design, not differences in feature processing.
> - **Using LLM-enhanced features is a widely-adopted practice in recent GFM studies.** Recent GFM works [1-3] (such as ZeroG [1], cited in line 125 where the LLM-enhanced tricks are introduced) regularly use LLMs to enhance node features on text-attributed graphs. These works differ in how LLMs are integrated, but all rely on LLMs’ generalization strength. In our case, this enhancement serves as a standard input processing step, not our major contribution.
> - **Retrieval of original texts contributes more significantly than LLM enhancement.**
> Our ablation in Section 5.5 (Figure 7) separates the effects of raw-text retrieval and LLM enhancement. Results show that **retrieving original node texts contributes more to performance**, particularly for zero-shot adaptation. This highlights the importance of recovering full semantic context (often lost in original features obtained by, *e.g.*, bag-of-words) and aligns with our goal of enabling robust cross-domain GFM.
> - **Additional ablation experiment (GRAVER without any textual enhancement).** To further isolate GRAVER’s contribution, we introduce a new ablation using the **original, non-enhanced node features** (*i.e.*, no text or LLM). Under this strict setting (1-shot node classification), GRAVER still outperforms strong baselines, confirming the effectiveness of its generative vocabulary and hierarchical routing modules. **Results:**
>
> |  | `CiteSeer` (cross-dataset) | `Wiki-CS` (cross-domain) |
> | --- | --- | --- |
> | **GCOPE** | 30.25 | 37.50 |
> | **MDGPT** | 33.15 | 40.22 |
> | **SAMGPT** | 35.67 | 42.64 |
> | **GRAVER** (ours) | **37.81*** | **45.90*** |
>
> ---
>
> > **Q2: The MoE-CoE routing mechanism may be functionally simple, despite being presented with complex naming.**
>
> **A2:** Thank you for raising this insightful point. While MoE-CoE involves softmax-based weighting, it differs from that in several key ways:
>
> - **Selective Sparsity vs. Uniform Aggregation.** Unlike typical softmax fusion that softly aggregates all inputs, MoE-CoE is designed to enforce selective sparsity. Through domain- and class-specific projections ($\mathbf{W}_ {\text{M}}$, $\mathbf{W}_{\text{C}}$ in Eq. (14)), it suppresses irrelevant sources, resulting in targeted and meaningful composition rather than uniform blending.
> - **Adaptive Routing Structure vs. Static Weighting.** Traditional softmax fusion often uses static attention over fixed candidates. In contrast, MoE-CoE dynamically computes routing weights conditioned on each target support instance, enabling fine-grained, task-aware augmentation.
> - **Hierarchical Decision Logic vs. Single-layer Fusion.** MoE-CoE explicitly separates domain-level and class-level routing, which makes the overall decision process more interpretable and modular, especially important when scaling to larger, noisier, or multi-task graph settings.
>
> We chose the term “MoE-CoE” to reflect this **structured design** and to emphasize GRAVER’s role as a flexible framework, not just a standalone model. We will clarify these distinctions to avoid any misunderstanding.
>
> ---
>
> > **Q3: Proposition 1 does not clearly support the claim that the upper bound of semantic discrepancy is governed by a specific vocabulary combination.**
>
> **A3:** We sincerely appreciate this insightful comment. Indeed, at first glance, the final form of the upper bound (Eq. (7) in Proposition 1) appears to depend solely on the number of disentangled factors ($K $). **However, this simplified form is the final theoretical conclusion after several derivation steps.** Importantly, during the derivation (Proof in Appendix C.2~C.3), the mutual information regularization term ($\mathcal{R}_{\text{MI}}$, Eq. (6)), induced by the Semantic Independence Promotion (SIP), **explicitly appears** and **constrains the intermediate upper bound** (Eq. (7), [a]).
>
> Specifically, our key theoretical insight is that by constraining $\mathcal{R}_ {\text{MI}}$ during pre-training, the semantic discrepancy ($\Delta$) can be effectively bounded. **We do not directly optimize this simplified upper bound during training.** Instead, we incorporate the mutual information regularizer $\mathcal{R}_ {\text{MI}}$ directly into our pre-training objective (Eq. (8)), implicitly controlling the semantic discrepancy between vocabularies. In other words, the simplified form of the upper bound (depending only on $K$) is precisely as have already explicitly controlled vocabulary independence through SIP ($\mathcal{R}_{\text{MI}}$).
>
> ---
>
> > **Q4: The logical connection between “disentangled ego-graphs” and “independent, minimal semantic units” is unclear.**
>
> **A4:** In Section 4.1, we explicitly decompose each node’s neighborhood (ego-graph) into multiple disentangled subgraphs, each capturing a distinct semantic aspect. Proposition 1 then theoretically demonstrates that if these subgraphs (or vocabularies) are semantically independent (each vocabulary encodes clearly distinct semantics), the overall semantic representation becomes more stable and robust to minor feature variations.
>
> Therefore, the phrase “independent and minimal semantic units” emphasizes that each disentangled subgraph ideally encodes a unique, non-redundant piece of semantic information. Minimizing semantic redundancy between these vocabularies (enforced by SIP and its mutual-information regularizer $\mathcal{R}_ {\text{MI}}$) is crucial for effectively bounding the semantic discrepancy and facilitating stable knowledge transfer across tasks and domains.
>
> ---
>
> > **Q5: Empirical results suggest Proposition 1’s theoretical claims might not align with observed practical outcomes.**
>
> **A5:** We appreciate this critical observation. There indeed appears to be a **misunderstanding** stemming from the simplified final expression of the bound in Proposition 1 (Eq. (7)), which depends only on the number of disentangled factors. **If taken literally,** reducing $K$ would minimize the semantic discrepancy, which contradicts the empirical observation that increasing semantic diversity (*i.e.*, larger $K$) improves performance.
>
> We clarify this misunderstanding as follows: the derived bound (Eq. (7)) explicitly relies on the assumption that semantic independence across vocabularies (enforced via SIP and the regularizer $\mathcal{R}_{\text{MI}}$) has already been achieved during pre-training (we also explain this in **A3**). Therefore, Proposition 1 is **not intended as a direct optimization target** (*i.e.*, reducing $K$), but rather as theoretical justification for why controlling semantic redundancy (via SIP) stabilizes the semantic representation.
>
> In practice, moderate semantic diversity (larger $K$) indeed proves beneficial by capturing richer semantics, as demonstrated by our empirical observations. Thus, there is no contradiction: Proposition 1 provides theoretical backing for controlling semantic independence through $\mathcal{R}_\text{MI}$, rather than advocating directly reducing $K$ (given a certain $K$, the upper bound of $\Delta$ is confirmed).
>
> ---
>
> > **Q6: The reason for splitting “ogbn-product” into “tech” and “home” is unclear, and the renaming causes confusion about the dataset’s origin.**
>
> **A6:** We appreciate the reviewer highlighting this confusion. The `ogbn-Tech` and `ogbn-Home` datasets used in our experiments are split from the original `ogbn-Products` dataset, following the same protocol used in related work ZeroG [1].
>
> Due to space constraints, we briefly referenced this in the main text, and we provide full details in Appendix D.1. As described on line 928, we created the two subsets to be **non-overlapping in class space**, which allows us to **simulate realistic in-domain adaptation under class-level shifts**, a setting that better reflects generalization within a single structural distribution.
>
> In our revision, we will explicitly clarify in the main text to avoid confusion and ensure dataset provenance is transparent.
>
> ---
>
> > **Q7: The contributions may feel overstated due to dense mathematical and technical presentation, despite the paper’s novelty and detail.**
>
> **A7:** We sincerely thank the reviewer for acknowledging the novelty of our ideas and the level of detail in the paper. We are especially encouraged by the time and effort taken to carefully engage with both the theoretical and empirical aspects of our work.
>
> That said, we understand the concern regarding potential over-packaging of technical content. Our intention was to provide clear theoretical grounding for key design choices, not to overstate their complexity. We will be mindful in refining the presentation to ensure that the contributions are communicated with clarity and appropriate emphasis.
>
> ---
>
> **We hope our responses have clarified the theoretical and empirical intentions behind our design. If the above points address your concerns, we would greatly appreciate a reconsideration of your score!**
>
> ---
>
> > **Refs:**
> > [1] ZeroG: Investigating cross-dataset zero-shot transferability in graphs, KDD 2024.
> > [2] One for All: Towards training one graph model for all classification tasks, ICLR 2024.
> > [3] Exploring the potential of large language models (LLMs) in learning on graphs, KDD 2024.

---

> > ### Comment · Reviewer_v4BW · 2025-08-06
> >
> > Thank you for the detailed response.
> >
> > - My questions or concerns on Q1, Q3 (partially), Q4, Q5, Q6 (mostly) are resolved. Plus, I appreciate the comments on Q7. I hope the authors can present their contributions in easy and concise words.
> > - Q2: I still did not get it. I am keeping my position that the current presentation is too complex and technical compared to the actual form the module takes.
> >   - Selective Sparsity vs. Uniform Aggregation: Where can I find the real sparsity in MoE-CoE? In other MoE architectures (like LLMs do), there is a selection operation, like top-k selection. However, I cannot find the selection operation in the MoE-CoE module in the authors' paper. Eq 14, 15 are basically dense, not sparse. Plus, attention-weighted sum using softmax-normalized weights is not a uniform aggregation.
> >   - Adaptive Routing Structure vs. Static Weighting: "dynamically computes routing weights" is what attention-weighted sum using softmax-normalized weights does.
> >   - Hierarchical Decision Logic vs. Single-layer Fusion: I think this hierarchy is a straightforward extension.
> > - Q3: Now I understand that the description is related to the role of Eq. (7), [a]. Thank you for your clarification. Then what is the meaning of Eq. (7), [b]? Why do you present [b] and what is its implication?
> > - Q6: I think providing the 'ogbn-product' prefix to Tech or Home is better for future researchers, and it can properly acknowledge the original author of the OGB project. It would be nice if the authors could add this after the publication (with an additional one page).
> >
> > Despite the remaining concerns, the authors resolve most of them. So, I will raise my score slightly. Could you give me the response to the above bullets?

---

> > > ### Author Response · Authors · 2025-08-06
> > >
> > > Thank you sincerely for your engagement and thoughtful feedback. We are truly grateful for your recognition of our work and your constructive suggestions, which have helped us improve both the presentation and clarity of the paper. We now address the remaining clarifications as follows.
> > >
> > > ---
> > >
> > > > **Q2+: Clarification on Selective Sparsity and Adaptive Routing in MoE-CoE.**
> > >
> > > **A2+:** Thank you for your sharp observations on the design of MoE-CoE. We agree with your assessment and appreciate the opportunity to clarify.
> > >
> > > - **Selective Sparsity vs. Uniform Aggregation**: You are correct. Our current formulation in Eq. (14) - Eq. (15) implements a **soft attention-based routing**, rather than a sparse top-$k$ selection used in hard MoE variants. Our intention behind the term “selective” was to emphasize that the routing is **non-uniform and sample-specific**, in contrast to uniform or static expert fusion. However, we acknowledge that this may cause ambiguity and will revise the wording accordingly to avoid overclaiming sparsity. Nonetheless, the sample-dependent adaptive weighting does bring a degree of routing flexibility that distinguishes it from traditional static averaging approaches.
> > > - **Adaptive Routing Structure vs. Static Weighting**: As you pointed out, our current mechanism adopts softmax-based dynamic routing, a widely used technique in attention models. We describe it as “adaptive” primarily to contrast with **fixed-weight fusion** or handcrafted aggregation rules. While not much appealing on its own, this mechanism is critical for enabling **continuous expert interpolation** in few-shot scenarios where hard gating can be unstable.
> > > - **Hierarchical Decision Logic vs. Single-layer Fusion**: We fully agree that our hierarchical MoE-CoE structure is conceptually straightforward. However, to the best of our knowledge, **this hierarchical routing design has not been explored in GFM fine-tuning**, especially for managing domain-level and intra-domain graph vocabulary selection separately. This structure provides a **modular and interpretable architecture** to organize multiple levels of augmentation granularity, making it easier to extend, prune, or visualize. In that sense, we believe it adds practical value beyond raw architectural complexity.
> > >
> > > We appreciate your detailed reading of these design choices and will revise the paper to make both the limitations and rationale more transparent.
> > >
> > > ---
> > >
> > > > **Q3+: Further Clarification on Eq. (7)[b].**
> > >
> > > **A3+:** Thank you for your follow-up question on Proposition 1. We’re glad the meaning of Eq. (7)[a] is now clear. While Eq. (7)[a] characterizes the condition under which the mutual information-based regularization term $\mathcal{R} _{\text{MI}}$ is optimized, Eq. (7)[b] plays a complementary and essential role in our theoretical argument.To further clarify:
> > >
> > > Eq. (7)[b] establishes a **tight upper bound** on the semantic discrepancy term $\Delta$, which can be interpreted as a form of semantic approximation error between the generated vocabulary and the original node representations. While Eq. (7)[a] expresses the condition under which semantic alignment (in terms of mutual information $\mathcal{R} _{\text{MI}}$) is optimized, Eq. (7)[b] goes one step further: it **quantifies the residual discrepancy under this optimization**, demonstrating that it is bounded and thus controllable. In other words, by regularizing $\mathcal{R} _{\text{MI}}$ during pre-training, we can **implicitly constrain the semantic discrepancy**, and Eq. (7)[b] gives a **guaranteed upper limit** on how large this discrepancy could be. This not only offers theoretical insight into the stability of the vocabulary generation process, but also serves as a justification for incorporating $\mathcal{R} _{\text{MI}}$ into our pre-training objective.
> > >
> > > This result has two key implications: **(1)** it offers a formal guarantee that the optimization guided by Eq. (7)[a] leads to controlled semantic deviation, and **(2)** it supports the stability and robustness of the generated vocabulary, especially when applied across different domains. In this sense, Eq. (7)[b] helps bridge the theoretical formulation and the practical reliability of our pre-training design.
> > >
> > > We appreciate your close reading of the theory section, and will make this connection more explicit in the final version.
> > >
> > > ---
> > >
> > > > **Q6+: Acknowledging Dataset Attribution.**
> > >
> > > **A6+:** We greatly appreciate your suggestion regarding dataset attribution. We will revise the two dataset names in the camera-ready version to include the **`ogbn-product`** prefix and add a citation to the original OGB paper for proper credit. We also plan to include a brief note in the appendix to clarify this decision for future researchers.
> > >
> > > ---
> > >
> > > We are grateful for the reviewer’s thoughtful engagement. Your critical questions have helped us refine the presentation and clarify key components. We hope that our responses have fully addressed your remaining concerns!

---

> > > > ### Comment · Reviewer_v4BW · 2025-08-07
> > > >
> > > > Thank you for your response and for agreeing with my opinion. Please make sure to revise the parts we discussed in the next version.

---

### Note · Authors · 2025-08-12

We sincerely thank all reviewers and ACs for their time, thoughtful feedback, and constructive engagement throughout the review and discussion phases.

Reviewers consistently expressed a positive attitude toward our work and reached a clear consensus on its key contributions:

- **Novel generative vocabulary design:** Multiple reviewers recognized the originality of our graphon-based vocabulary construction, which disentangles semantic patterns through ego-graphs to support transferable and robust fine-tuning. (`#v4BW`, `#92Wv`, `#m8FB`)

- **Theoretical soundness and interpretability:** Our theoretical analyses, especially Propositions 1 and 2, were acknowledged as providing meaningful insights into vocabulary independence and generative stability. Reviewers appreciated how these theoretical results support key components of our framework. (`#v4BW`, `#m8FB`)

- **Effective and structured routing architecture:** The hierarchical MoE-CoE routing mechanism was seen as a practical and interpretable solution to domain and class heterogeneity. While some simplifications were noted, reviewers acknowledged the merit of the two-level design and the improvements we made in response to clarification requests. (`#v4BW`, `#92Wv`, `#m8FB`)

- **Comprehensive and rigorous empirical evaluation:** Reviewers valued the breadth and depth of our experiments, which included 15 competitive baselines, extensive ablation studies, and diverse tasks across multiple domains. These evaluations convincingly demonstrated the effectiveness, robustness, and efficiency of GRAVER. (`#92Wv`, `#m8FB`, `#HK8Q`)

- **Clarity and responsiveness in rebuttal:** Reviewers appreciated that our rebuttal addressed most concerns in a clear, thorough, and professional manner, helping to improve the overall presentation and understanding of the paper. (`#v4BW`, `#92Wv`, `#m8FB`)

Although a few concerns remain regarding scalability to very large graphs (`#HK8Q`) and the presentation of certain components (`#v4BW`, `#HK8Q`), reviewers agreed that these issues do not undermine the overall contributions or technical quality of our work. Notably, Reviewer `#HK8Q` increased their score following the discussion, and all reviewers maintained a clear endorsement of the paper’s acceptance.

We are grateful for the constructive input that helped us refine our presentation and clarify several important aspects of the work. We look forward to incorporating the suggested improvements in the final version.

---

### Decision · Program_Chairs · 2025-09-17

**Decision:**

Accept (poster)

**Comment:**

The paper proposes a generative graph vocabulary construction based on ego-graph disentanglement and graphon modeling to create transferable subgraph patterns. These vocabularies are integrated via a hierarchical Mixture-of-Experts (MoE) and Class-of-Experts (CoE) routing mechanism to enhance robustness in cross-domain and few-shot node/graph classification tasks. The authors claim that GRAVER outperforms 15 state-of-the-art baselines in effectiveness, robustness, and efficiency, supported by extensive experiments and theoretical analyses (Propositions 1 and 2) that ensure semantic stability and transferability.

Reviewers point some weakness of the paper, including:

Scalability Concerns: The evaluation is limited to graphs with up to 250,000 nodes, which, while substantial for GFM research, does not address real-world graphs with millions of nodes;

Limited Discussion of Traditional Few-Shot Methods: The initial submission lacked a comparison with traditional few-shot graph learning approaches, though this was addressed in the rebuttal with a proposed related work section.

Empirical Validation of Vocabulary Transfer: While the authors claim vocabulary reusability, more explicit cross-domain reuse metrics could strengthen the evidence

The authors’ rebuttal was thorough, addressing most concerns with additional experiments, clarifications, and commitments to revise the manuscript for clarity and transparency. The consensus among reviewers leaned toward acceptance.